# Representations Shape Weak-to-Strong Generalization:
# Theoretical Insights and Empirical Predictions

**Yihao Xue** [1]  **Jiping Li** [2]  **Baharan Mirzasoleiman** [1]

## Abstract

Weak-to-Strong Generalization (W2SG), where a weak model supervises a stronger one, serves as an important analogy for understanding how humans might guide superhuman intelligence in the future. Promising empirical results revealed that a strong model can surpass its weak supervisor. While recent work has offered theoretical insights into this phenomenon, a clear understanding of the interactions between weak and strong models that drive W2SG remains elusive. We investigate W2SG through a theoretical lens and show that it can be characterized using kernels derived from the principal components of weak and strong models' internal representations. These kernels can be used to define a space that, at a high level, captures what the weak model is unable to learn but is learnable by the strong model. The projection of labels onto this space quantifies how much the strong model falls short of its full potential due to weak supervision. This characterization also provides insights into how certain errors in weak supervision can be corrected by the strong model, regardless of overfitting. Our theory has significant practical implications, providing a representation-based metric that predicts W2SG performance trends without requiring labels, as shown in experiments on molecular predictions with transformers and 5 NLP tasks involving 52 LLMs.

## 1. Introduction

As AI systems become increasingly capable of performing complex tasks beyond human comprehension, humans will inevitably serve as "weak supervisors" in aligning advanced AI. To investigate this fundamental problem, Burns

[1]Department of Computer Science, University of California, Los Angeles [2]Department of Mathematics, University of California, Los Angeles. Correspondence to: Yihao Xue <yihaoxue@g.ucla.edu>.

*Proceedings of the 42nd International Conference on Machine Learning*, Vancouver, Canada. PMLR 267, 2025. Copyright 2025 by the author(s).

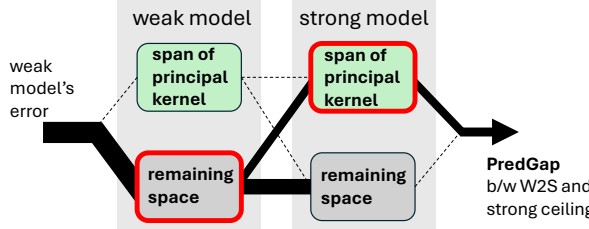

Figure 1: An illustration of our main result (Thm. 3.8). The path connecting the two highlighted regions represents the overlap b/w the complement of a scaled span of the weak model's *principal kernel* and the scaled span of the strong model's *principal kernel*, determining the contribution of the weak model's errors to **PredGap**.

et al. (2023) propose an analogy that can be empirically explored today: can a weak model effectively supervise a stronger one? This framework, known as Weak-to-Strong Generalization (W2SG), involves leveraging a weak model, finetuned on a specific task, to supervise the finetuning of a stronger model. In this analogy, the finetuning task represents concepts tied to human values or skills, the finetuned weak model represents humans—limited in capability but aligned with human values, and the strong model represents superhuman intelligence–powerful but initially unaligned. Promising results from (Burns et al., 2023) show that the strong model can significantly outperform its weak supervisor. For instance, a GPT-4 model supervised by a fine-tuned GPT-2-level model achieves nearly 20% better performance than the weak supervisor on NLP tasks.

At first glance, this phenomenon seems counterintuitive. After all, the strong model is explicitly trained to fit the weak supervision. Yet, it goes beyond mere imitation and generalizes better. It is important to understand which intrinsic properties of the weak and strong models enable W2SG.

Efforts have been made toward a theoretical understanding of W2SG. Charikar et al. (2024) demonstrates that the disagreement between finetuned weak and strong models correlates with performance gains in W2SG. However, their analysis assumes high-quality representations in the strong model and does not address the role of the weak model's representations. The analysis of (Lang et al., 2024; Shin et al., 2024) assumes a generalized version of an adversarially robust strong model, where W2SG arises solely from underfitting weak supervision. This framework excludes im-

portant scenarios such as benign overfitting, where W2SG occurs despite overfitting. Wu & Sahai (2024) particularly studied benign overfitting and examined the impact of number of weakly labeled data points. However, we still lack an overarching explanation that captures the interaction between weak and strong models in enabling W2SG, as well as how it determines which weak supervision errors are corrected in general scenarios. The challenge lies in characterizing the abstract concepts including the knowledge embedded in the weak and strong models, their utilization, and their respective roles in W2SG. Striving for results that are general enough to capture a spectrum of behaviors without overly strict assumptions further adds to the complexity.

To address this, we adopt a representation-based perspective, analyzing finetuning as a process of learning a function on fixed representations to uncover how the internal structures of weak and strong models influence W2SG. Under a very general assumption about the representations, we demonstrate (illustrated in Fig. 1) that the key quantifiable property governing W2SG is the overlap between two spaces: one representing what the weak model's *principal representations* (capturing key knowledge gained during pretraining) do not cover, and the other representing what the strong model's *principal representations* do cover. Errors in weak supervision that fall within this overlap hinder the strong model from reaching its full potential, leading to a prediction gap between the strong model finetuned with weak supervision and that finetuned with ground truth labels. A smaller overlap implies that fewer of the weak model's mistakes are replicated, resulting in better W2SG performance.

We then demonstrate an important use case of our main result: explaining *benign overfitting*, where the W2S model overfits the weak model's mistakes on finetuning data yet paradoxically generalizes better on the test set. Using our theoretical framework, we establish a general condition for benign overfitting and apply it to a toy example to concretely illustrate the role of representations in error replication: errors that do not align with the kernel defined by the strong model's principal representations are not replicated by the W2S model, regardless of the extent of overfitting.

Our theory offers a metric that predicts trends in W2SG performance in practice *without having the finetuning task labels*. This metric, which measures the overlap between the two highlighted regions in Fig. 1, shows a strong correlation with W2SG performance across various settings. The extensive experiments across 8 datasets, involving 150 small transformers and 52 LLMs, not only validate our theoretical insights but also suggest their potential applications in managing W2SG, providing a deeper understanding of LLM behavior through their internal representation structures.

## 2. Related Work

There have been many recent works that theoretically explore W2SG. Somerstep et al. (2024) adopt a transfer learning perspective, focusing on improving W2SG through in-context learning rather than explaining how W2SG emerges. Lang et al. (2024); Shin et al. (2024) analyze W2SG by considering a generalized version of adversarially robust models, showing that certain errors in weak supervision can be corrected by leveraging the good neighborhood structure in the data. However, their argument attributes error correction solely to underfitting—i.e., avoiding fitting mislabeled finetuning data. This overlooks an important scenario recently discussed in (Wu & Sahai, 2024), known as benign overfitting, where the strong model overfits mislabeled finetuning data but still achieves accurate test-time predictions. Benign overfitting is particularly relevant in practice, as large neural networks often have the capacity to overfit while still generalizing effectively (Zhang et al., 2021). Closer to our setting, Charikar et al. (2024) formalized W2SG using a representation-based perspective. Their work demonstrates that performance gain in W2SG correlates with the disagreement between the finetuned weak and strong models, assuming high-quality representations for the strong model. While insightful, it does not characterize the role of the weak model's representations, leaving the exact conditions for effective W2SG unclear.

Compared to (Lang et al., 2024), we analyze W2SG in a more realistic setting where error correction can result from either underfitting or overfitting, allowing for a full spectrum of behaviors. While benign overfitting is not our primary focus, we discuss it as a special case in Sec. 4 due to its importance and offer new insights. Compared to (Charikar et al., 2024), we explicitly links W2SG performance to the interaction between the weak and strong models' representations, providing a more comprehensive view of how the intrinsic properties of the two models jointly determine W2SG.

## 3. W2SG from a Representation Perspective

We first formalize finetuning from a representation-based perspective, then introduce the properties of the representations considered, and finally present our main theory.

### 3.1. A representation-based perspective

The knowledge a model acquires through pretraining enables it to interpret inputs, extract relevant information, and organize it into meaningful intermediate states. This can be formalized as a "representation function", $h$, which transforms data into structured representations. Finetuning leverages this knowledge to produce the desired output, which we formalize as learning a new function $f$ on the fixed $h$. The entire model is thus represented as the composition $f \circ h$. For simplicity, we consider the outputs

of $h$ as vectors, and focus on the case where $f$ is a linear functions. This is practically relevant because: (1) Training a linear task head on fixed representations is common with large foundation models, e.g., using embedding LLMs (Muennighoff et al., 2022), linear probing on intermediate activations (Zou et al., 2023; Nanda et al., 2023; Marks & Tegmark, 2023). (2) fine-tuning of LLMs largely operates in the NTK regime (Jacot et al., 2018), where training dynamics are captured by a linear model on representations derived from model gradients (Malladi et al., 2023). (3) Our experiments in Sec. 5 show that insights from analyzing linear functions generalize to the complex non-linear setting of finetuning entire LLMs from pretrained weights.

## 3.2. Preliminaries

**Notations.** We sometimes abbreviate a matrix $\boldsymbol{A} \in \mathbb{R}^{l \times m}$ as $[A_{i,j}]_{1 \le i \le l, 1 \le j \le m}$ when each element $A_{i,j}$ can be expressed as a generic term in terms of its indices. $\lambda_{\min, \ne 0}(\boldsymbol{A})$ denotes the smallest nonzero eigenvalue of matrix $\boldsymbol{A}$.

**Data.** Let $\mathcal{D}$ denote the distribution of the finetuning task's data, defined over the input-label pairs $(\boldsymbol{x}, y) \in \mathcal{X} \times \mathcal{Y}$, where $\mathcal{Y} = \mathbb{R}$. In W2SG, we have two splits of data sampled from $\mathcal{D}$. The first subset, $\tilde{\mathcal{D}} = \{(\tilde{\boldsymbol{x}}_i, \tilde{y}_i)\}_{i=1}^{\tilde{n}}$, consists of $\tilde{n}$ i.i.d. samples and is used for finetuning the weak model. The second subset, $\hat{\mathcal{D}} = \{(\hat{\boldsymbol{x}}_i, \hat{y}_i)\}_{i=1}^{\hat{n}}$ with $\hat{n}$ i.i.d. samples is used for finetuning the strong model. Note that the weak model's outputs will be used as labels in place of the actual $\hat{y}_i$'s. In our notation, quantities associated with the two splits are marked by the diacritical symbols, ˜ and ˆ, respectively.

**Models.** We denote the weak and strong models' representation functions as $h_{\mathrm{w}}$ and $h_{\mathrm{s}}$, respectively. The finetuned weak model is represented as $f_{\mathrm{w}} \circ h_{\mathrm{w}}$, with

$$f_{\mathrm{w}} = \arg\min_{f \in \mathcal{F}_{\mathrm{w}}} \left( \frac{1}{\tilde{n}} \sum_{i=1}^{\tilde{n}} (f(h_{\mathrm{w}}(\tilde{\boldsymbol{x}}_i)) - \tilde{y}_i)^2 + \beta_{\mathrm{w}} R(f) \right).$$

where $R(\cdot)$ represents $\ell_2$ regularization.

The *W2S model*, which refers to the strong model finetuned with weak supervision, is represented as $f_{\mathrm{w2s}} \circ h_{\mathrm{s}}$, with

$$f_{\mathrm{w2s}} = \arg\min_{f \in \mathcal{F}_{\mathrm{s}}} \left( \frac{1}{\hat{n}} \sum_{i=1}^{\hat{n}} (f(h_{\mathrm{s}}(\hat{\boldsymbol{x}}_i)) - f_{\mathrm{w}}(h_{\mathrm{w}}(\hat{\boldsymbol{x}}_i)))^2 + \beta_{\mathrm{s}} R(f) \right).$$

Additionally, as a reference, we define the *strong ceiling model* as the strong model finetuned with the ground truth labels. It is represented as $f_{\mathrm{sc}} \circ h_{\mathrm{s}}$ with

$$f_{\mathrm{sc}} = \arg\min_{f \in \mathcal{F}_{\mathrm{s}}} \left( \frac{1}{\hat{n}} \sum_{i=1}^{\hat{n}} (f(h_{\mathrm{s}}(\hat{\boldsymbol{x}}_i)) - \hat{y}_i)^2 + R_{\mathrm{s}}(f) \right).$$

**Evaluation.** At test time, given any labeling function $g : \mathcal{X} \to \mathcal{Y}$, we define its test error as the loss on the population: $\mathrm{Err}(g) = \mathbb{E}_{(\boldsymbol{x}, y) \sim \mathcal{D}}[(g(\boldsymbol{x}) - y)^2]$. We then introduce

the shorthand notations: the weak model's test error $\mathrm{Err}_{\mathrm{w}} = \mathrm{Err}(f_{\mathrm{w}} \circ h_{\mathrm{w}})$, the W2S model's test error $\mathrm{Err}_{\mathrm{w2s}} = \mathrm{Err}(f_{\mathrm{w2s}} \circ h_{\mathrm{s}})$, and the strong ceiling model's test error $\mathrm{Err}_{\mathrm{sc}} = \mathrm{Err}(f_{\mathrm{sc}} \circ h_{\mathrm{s}})$. $\mathrm{Err}_{\mathrm{w2s}}$ measures the performance achieved through W2SG, while $\mathrm{Err}_{\mathrm{sc}}$ serves as the upper limit.

We also introduce **PredGap**, the squared difference between the predictions of the W2S and strong ceiling models:

$$\mathbf{PredGap} = \mathbb{E}_{(\boldsymbol{x}, y) \sim \mathcal{D}}[(f_{\mathrm{w2s}}(h_{\mathrm{s}}(\boldsymbol{x})) - f_{\mathrm{sc}}(h_{\mathrm{s}}(\boldsymbol{x})))^2].$$

It captures how much the strong model falls short of its full potential due to weak supervision. It is also indicative of $\mathrm{Err}_{\mathrm{w2s}}$, the direct measure of W2SG performance, through these connections: (1) If the strong ceiling model is nearly perfect, it follows that **PredGap** $\approx \mathrm{Err}_{\mathrm{w2s}}$ as the strong ceiling's predictions are almost identical to the ground truth. This is not unlikely, since the ultimate goal of W2SG is to operate in cases where the strong model is a superhuman-level AI (Burns et al., 2023), plausibly capable of achieving perfect results if provided with ground truth labels. (2) With small regularization and well-conditioned representations, $\mathrm{Err}_{\mathrm{w2s}} \approx \mathbf{PredGap} + \mathrm{Err}_{\mathrm{sc}}$ (Thm. B.3), analogous to the Pythagorean theorem. Then, **PredGap** directly determines $\mathrm{Err}_{\mathrm{w2s}}$ for fixed $\mathrm{Err}_{\mathrm{sc}}$. (3) For general cases, the upper bound $\sqrt{\mathrm{Err}_{\mathrm{w2s}}} \le \sqrt{\mathbf{PredGap}} + \sqrt{\mathrm{Err}_{\mathrm{sc}}}$ follows from the triangle inequality. Furthermore, the result obtained from analyzing **PredGap** helps predict $\mathrm{Err}_{\mathrm{w2s}}$ in our experiments (Sec. 5). Thus, our main analysis focuses on **PredGap**.

## 3.3. Setting: representations with a well-concentrated principal part and a manageable non-principal part

We first define two basic concepts, kernel and covariance, before introducing a general assumption on representations.

**Definition 3.1** (Kernel Matrix). Given $h : \mathcal{X} \to \mathbb{R}^d$, we define the kernel matrix on the finetuning dataset $\hat{\mathcal{D}}$ as $\hat{\boldsymbol{K}}(h) = [h(\hat{\boldsymbol{x}}_i)^\top h(\hat{\boldsymbol{x}}_j)]_{1 \le i,j \le \hat{n}}$, a $\hat{n} \times \hat{n}$ matrix where each element represents the inner product between a pair of representations. $\tilde{\boldsymbol{K}}(h)$ is defined on $\tilde{\mathcal{D}}$ in the same manner.

**Definition 3.2** (Population/Empirical Covariance Matrices). Given $h : \mathcal{X} \to \mathbb{R}^d$, we define the population covariance over distribution $\mathcal{D}$ as $\boldsymbol{\Sigma}(h) := \mathbb{E}_{\mathcal{D}_{\boldsymbol{x}}}[h(\boldsymbol{x})h(\boldsymbol{x})^\top]$. The empirical version on $\hat{\mathcal{D}}$ is defined as $\hat{\boldsymbol{\Sigma}}(h) := \frac{1}{\hat{n}} \sum_{i=1}^{\hat{n}} h(\hat{\boldsymbol{x}}_i)h(\hat{\boldsymbol{x}}_i)^\top$. $\tilde{\boldsymbol{\Sigma}}(h)$ is defined on $\tilde{\mathcal{D}}$ in the same manner.

Given a representation function and a reasonable sample size, certain components in the representations should *concentrate well*, meaning they adequately reflect the population distribution. These components are pivotal to the model's generalization. In our analysis, we focus on cases where the remainder—the less-well-concentrated components—satisfies certain conditions, ensuring their impact remains theoretically tractable. The decomposition of representations into these two parts is formalized as follows.

**Definition 3.3** $((\delta, \hat{\gamma}, \tilde{\gamma})$-decomposability). Given $\mathcal{D}$, $\tilde{\mathcal{D}}$, $\hat{\mathcal{D}}$, and a representation function $h : \mathcal{X} \to \mathcal{R}$, we say that the representations of $h$ are $(\delta, \hat{\gamma}, \tilde{\gamma})$-*decomposable w.r.t. a subspace* $\mathcal{V}$ (of $\mathcal{R}$), for some $\delta = O(1)$, $\hat{\gamma} = O(1)$, and $\tilde{\gamma} = O(1)$, if there exists a subset of eigenvectors of $\boldsymbol{\Sigma}(h)$ corresponding to non-zero eigenvalues such that the following holds. Let $\mathcal{V}$ denote the span of these eigenvectors, and let $\mathcal{V}^{\perp}$ denote its orthogonal complement. Let $\boldsymbol{\Pi}_{\mathcal{V}}$ and $\boldsymbol{\Pi}_{\mathcal{V}^{\perp}}$ denote the orthogonal projections onto $\mathcal{V}$ and $\mathcal{V}^{\perp}$, respectively. Define $\rho = \lambda_{\min, \neq 0}(\boldsymbol{\Sigma}(\boldsymbol{\Pi}_{\mathcal{V}}h))$ and $\gamma = \min(\hat{\gamma}, \tilde{\gamma})$. With high probability of $1 - o(1)$:

(a) **Boundedness**. A basic condition that ensures reasonable magnitudes of representations and labels: $\|\boldsymbol{\Sigma}(h)\|_{\mathrm{op}} = O(1)$, $\|\hat{\boldsymbol{\Sigma}}(h)\|_{\mathrm{op}} = O(1)$ $\|\tilde{\boldsymbol{\Sigma}}(h)\|_{\mathrm{op}} = O(1)$, $\mathbb{E}[y^2] = O(1)$, $\frac{1}{\hat{n}} \sum_{i=1}^{\hat{n}} \hat{y}_i^2 = O(1)$ and $\frac{1}{\tilde{n}} \sum_{i=1}^{\tilde{n}} \tilde{y}_i^2 = O(1)$.

(b) **Concentration on $\mathcal{V}$**. Representations are well-concentrated in the subspace $\mathcal{V}$, both in terms of their covariance and their correlation with labels: $\|\hat{\boldsymbol{\Sigma}}(\boldsymbol{\Pi}_{\mathcal{V}}h) - \boldsymbol{\Sigma}(\boldsymbol{\Pi}_{\mathcal{V}}h)\|_{\mathrm{op}} = o(\gamma^2 + \delta^2 + \rho^2)$, $\|\tilde{\boldsymbol{\Sigma}}(\boldsymbol{\Pi}_{\mathcal{V}}h) - \boldsymbol{\Sigma}(\boldsymbol{\Pi}_{\mathcal{V}}h)\|_{\mathrm{op}} = o(\gamma^2 + \delta^2 + \rho^2)$, $\|\frac{1}{\hat{n}} \sum_{i=1}^{\hat{n}} \boldsymbol{\Pi}_{\mathcal{V}}h(\hat{\boldsymbol{x}}_i)\hat{y}_i - \mathbb{E}[\boldsymbol{\Pi}_{\mathcal{V}}h(\boldsymbol{x})y]\| = o(\gamma + \delta + \rho)$ and $\|\frac{1}{\tilde{n}} \sum_{i=1}^{\tilde{n}} \boldsymbol{\Pi}_{\mathcal{V}}h(\tilde{\boldsymbol{x}}_i)\tilde{y}_i - \mathbb{E}[\boldsymbol{\Pi}_{\mathcal{V}}h(\boldsymbol{x})y]\| = o(\gamma + \delta + \rho)$.

(c) **Kernel-wise $\delta$-isotropy on $\mathcal{V}^{\perp}$**. The kernels constructed using only the components in $\mathcal{V}^{\perp}$ exhibit certain uniformity in all orientations, with the extent of uniformity controlled by $\delta$: $\|\frac{1}{\hat{n}}\hat{\boldsymbol{K}}(\boldsymbol{\Pi}_{\mathcal{V}^{\perp}}h) - \hat{\gamma}\boldsymbol{I}\|_{\mathrm{op}} = o(\gamma^2 + \delta^2)$, and $\|\frac{1}{\tilde{n}}\tilde{\boldsymbol{K}}(\boldsymbol{\Pi}_{\mathcal{V}^{\perp}}h) - \tilde{\gamma}\boldsymbol{I}\|_{\mathrm{op}} = o(\gamma^2 + \delta^2)$.

(d) **Small cross-sample inner-product on $\mathcal{V}^{\perp}$**. $\|\frac{1}{\sqrt{\hat{n}\tilde{n}}}[(\boldsymbol{\Pi}_{\mathcal{V}^{\perp}}h(\hat{\boldsymbol{x}}_i))^{\top}\boldsymbol{\Pi}_{\mathcal{V}^{\perp}}h(\tilde{\boldsymbol{x}}_j)]_{1 \leq i \leq \hat{n}, 1 \leq j \leq \tilde{n}}\|_{\mathrm{op}} = o(\gamma + \delta)$, which holds when representations on $\mathcal{V}^{\perp}$ are nearly orthogonal across samples or have small magnitudes.

(e) **Diminishing population covariance on $\mathcal{V}^{\perp}$**. The representations on $\mathcal{V}^{\perp}$ have small magnitude in the population: $\|\boldsymbol{\Sigma}(\boldsymbol{\Pi}_{\mathcal{V}^{\perp}}h)\|_{\mathrm{op}} = o(\gamma + \delta)$.

**Additional explanation for Kernel-wise $\delta$-isotropy on $\mathcal{V}^{\perp}$**. To provide a clearer understanding of this condition, consider the following: If $\delta$ is very small (e.g., $\delta = 0$), the kernel on $\hat{\mathcal{D}}$ is nearly identical to $\hat{\gamma}\boldsymbol{I}$, meaning it does not exhibit any specific patterns that differentiate between data points. In contrast, with a larger $\delta$ (e.g., $\delta \gg \hat{\gamma}$), this requirement is much more relaxed—the kernel no longer needs to closely resemble $\hat{\gamma}\boldsymbol{I}$ but instead must simply have its magnitude bounded by $o(\delta)$. Thus, it accommodates scenarios where the kernel is highly isotropic, very small in scale, or anywhere in between. This is key to our analysis, as it ensures the effect of the less well-concentrated part of the representations remains tractable. We note that this condition is not only analytically convenient but also practically relevant in real-world scenarios. For example, high-dimensional sub-Gaussian noise satisfies this condition with a small $\delta$—a situation highly relevant to deep neural networks with large internal dimensions, where vectors tend

to be approximately orthogonal in the high-dimensional limit. More concrete instances will be presented in Examples 3.4 and 3.5, as well as in Theorem 3.6, along with discussions of their significance and relevance.

**Additional explanation for Diminishing population covariance on $\mathcal{V}^{\perp}$**. We note that this condition does not imply negligible impact of representations on $\mathcal{V}^{\perp}$. For example, when $\delta$ is small, the model can in fact leverage the components in $\mathcal{V}^{\perp}$ to interpolate the training data, even when such interpolation cannot be achieved by the components in $\mathcal{V}$ (see Example 4.2).

We refer to $\boldsymbol{\Pi}_{\mathcal{V}}h(\boldsymbol{x})$, the well-concentrated part of the representation, as the *principal representation*, and the remainder, $\boldsymbol{\Pi}_{\mathcal{V}^{\perp}}h(\boldsymbol{x})$, as the *non-principal representation*.

**Examples of Def. 3.3.** Def. 3.3 is highly general, covering various representation distributions and dimensionalities. One simple case is when all components are well-concentrated, i.e., the entire representation is principal. This occurs when the representations exhibit a certain low-rank structure, which is common in deep neural networks (Huh et al., 2021). Below is a concrete example.

**Example 3.4** (Arbitrarily parameterized; bounded representations with low intrinsic dimension). Given $h : \mathcal{X} \to \mathbb{R}^d$, for any $(\boldsymbol{x}, y)$, $\|h(\boldsymbol{x})\|^2 \leq B$ and $y^2 \leq C$, where $C = \Theta(1)$. Additionally, $\|\boldsymbol{\Sigma}(h)\|_{\mathrm{op}} = \Theta(1)$. The intrinsic dimension of $\boldsymbol{\Sigma}(h)$ is defined as $\mathbf{intdim}(\boldsymbol{\Sigma}(h)) = \frac{\mathrm{Tr}(\boldsymbol{\Sigma})}{\|\boldsymbol{\Sigma}\|_{\mathrm{op}}}$, denoted by $q$. Let $n = \min(\hat{n}, \tilde{n})$ and assume $n^{1-c} = \omega(B\log(q))$ for some constant $c < 1$. Then, the representations are $(n^{-0.1c}, 0, 0)$-decomposable w.r.t. $\mathbb{R}^d$.

*Remark.* The conditions imply a low intrinsic dimension relative to the sample size: $q\log q = o(n^{1-c})$ (App. C.1), but without restricting the actual dimension $d$, allowing both under- $(d < n)$ and over-parameterized $(d \geq n)$ settings.

The next example is related to the spiked covariance model originating from PCA and widely used in recent theoretical studies across various domains (e.g., (Muthukumar et al., 2021; Nakada et al., 2023)). It is also related to the sparse coding model, which has its roots in computer vision (Olshausen & Field, 1997), and has been applied to language modeling (Arora et al., 2018) and deep learning theory (e.g., (Allen-Zhu & Li, 2020)). More references are in App. C.2. We consider representations that follow a sub-Gaussian, which is a very general class of distributions, including, e.g., any bounded random variables and Gaussian.

**Example 3.5** (Heavily overparameterized; sub-Gaussian with spiked covariance). Given $h : \mathcal{X} \to \mathbb{R}^d$ and randomly drawn $\boldsymbol{x}$, $h(\boldsymbol{x})$ has independent zero-mean sub-Gaussian entries. The first $k$ entries have a (sub-Gaussian) parameter of $\Theta(1)$ and variance 1, while the remaining $d - k$ entries have a parameter of $\Theta(\frac{\sigma^2}{d-k})$ and variance $\frac{\sigma^2}{d-k}$. The scalings satisfy: $\tilde{n} = \Theta(\hat{n})$, $\sigma^2 = O(\hat{n})$, $\hat{n} = \omega(k^2)$, and

$d = \omega(\hat{n}^2)$. The labels have bounded moment, $\mathbb{E}[y^2] = O(1)$. Then, the representations are $(0, \frac{\sigma^2}{\hat{n}}, \frac{\sigma^2}{\hat{n}})$-decomposable w.r.t. the subspace corresponding to the first $k$ coordinates.

*Remark.* Compared to Example 3.4, this example accommodates cases with high intrinsic dimensions. For instance, if we set $\sigma^2 = \Theta(\hat{n})$, then **intdim**$(\boldsymbol{\Sigma}(h)) = \Theta(n)$.

More complex examples can be constructed from the fact that adding high-dimensional sub-Gaussian to $(\delta, 0, 0)$-decomposable representations preserves decomposability:

**Theorem 3.6.** *Given a representation function $h$ whose representations $h(\boldsymbol{x}) \in \mathbb{R}^d$ are $(\delta, 0, 0)$-decomposable w.r.t. $\mathbb{R}^d$, we construct new representations with $\alpha(\boldsymbol{x}) = \boldsymbol{M}h(\boldsymbol{x}) + \boldsymbol{M}^\perp \xi(\boldsymbol{x})$, where $\boldsymbol{M} \in \mathbb{R}^{(d+m) \times d}$ and $\boldsymbol{M}^\perp \in \mathbb{R}^{(d+m) \times m}$ both have orthonormal columns, and their column spaces are orthogonal to each others. If elements in $\xi(\boldsymbol{x}) \in \mathbb{R}^m$ are independent zero-mean sub-Gaussian with parameter $\Theta(\frac{\sigma^2}{m})$ and variance $\frac{\sigma^2}{m}$, assuming $\tilde{n} = \Theta(\hat{n})$, $m = \omega(\hat{n}^2)$, and $\sigma^2 = O(\hat{n})$, then $\alpha$'s representations are $(\delta, \frac{\sigma^2}{\hat{n}}, \frac{\sigma^2}{\hat{n}})$-decomposable w.r.t. the span of $\boldsymbol{M}$'s columns.*

*Remark.* For instance, one could take $h$ from Example 3.4.

We assume both models' representations satisfy Def. 3.3:

**Assumption 3.7.** $h_\mathrm{w}$'s representations are $(\delta_\mathrm{w}, \hat{\gamma}_\mathrm{w}, \tilde{\gamma}_\mathrm{w})$-decomposable w.r.t. $\mathcal{V}_\mathrm{w}$, and $h_\mathrm{s}$'s representations are $(\delta_\mathrm{s}, \hat{\gamma}_\mathrm{s}, \tilde{\gamma}_\mathrm{s})$-decomposable w.r.t. $\mathcal{V}_\mathrm{s}$ .

### 3.4. Principal representations shape PredGap

**Intuition.** One implication of Def. 3.3 is that only what is learned through the principal representations will be reflected at test time. Thus, the weak model's mistakes primarily stem from its inability to generate certain outputs using its principal representations. For the same reason, among these mistakes, only those expressible through the strong model's principal representations will affect its test performance. Therefore, a key concept affecting W2SG performance is **"what the weak model is unable to learn but is learnable by the strong model using their respective principal representations"**, which we seek to quantify.

**Formalization.** To formalize the above idea, we leverage $\hat{\boldsymbol{K}}(\boldsymbol{\Pi}_{\mathcal{V}_\mathrm{w}} h_\mathrm{w})$ and $\hat{\boldsymbol{K}}(\boldsymbol{\Pi}_{\mathcal{V}_\mathrm{s}} h_\mathrm{s})$–kernels computed using only the weak and strong models' principal representations, referred to as *principal kernels*. We define the following

$$\boldsymbol{P}_\mathrm{w} := \frac{1}{\hat{n}} \hat{\boldsymbol{K}}(\boldsymbol{\Pi}_{\mathcal{V}_\mathrm{w}} h_\mathrm{w}) \left( \frac{1}{\hat{n}} \hat{\boldsymbol{K}}(\boldsymbol{\Pi}_{\mathcal{V}_\mathrm{w}} h_\mathrm{w}) + (\beta_\mathrm{w} + \tilde{\gamma}_\mathrm{w})\boldsymbol{I} \right)^{-1},$$

$$\boldsymbol{P}_\mathrm{s} := \frac{1}{\hat{n}} \hat{\boldsymbol{K}}(\boldsymbol{\Pi}_{\mathcal{V}_\mathrm{s}} h_\mathrm{s}) \left( \frac{1}{\hat{n}} \hat{\boldsymbol{K}}(\boldsymbol{\Pi}_{\mathcal{V}_\mathrm{s}} h_\mathrm{s}) + (\beta_\mathrm{s} + \hat{\gamma}_\mathrm{s})\boldsymbol{I} \right)^{-1}.$$

$\boldsymbol{P}_\mathrm{w}$ and $\boldsymbol{P}_\mathrm{s}$ represent scaled projections onto the spans of the principal kernels. Each captures the space of output patterns that its respective model can express through its principal representations (with regularization taken into account). Then, the earlier intuition can be characterized as follows.

**Theorem 3.8** (Main result). *Under Assump. 3.7, and assuming reasonable regularization: $\delta_w \leq \beta_w = O(1)$ and $\delta_s \leq \beta_s = O(1)$, let $\hat{\boldsymbol{y}} = [\hat{y}_1 \; \hat{y}_2 \; \ldots \; \hat{y}_{\hat{n}}]^\top$. Then, w.h.p., we have*

$$\boldsymbol{PredGap} = \|\boldsymbol{P}_s(\boldsymbol{I} - \boldsymbol{P}_w)\frac{1}{\sqrt{\hat{n}}}\hat{\boldsymbol{y}}\|^2 \pm o(1) \qquad (1)$$

$\boldsymbol{P}_\mathrm{s}(\boldsymbol{I} - \boldsymbol{P}_\mathrm{w})$ **captures "what the weak model is unable to learn but is learnable by the strong model using their respective principal representations".** Therefore, it determines the mistakes that will be learned by the strong model, as discussed in the intuition. A more powerful weak model has a $\boldsymbol{P}_\mathrm{w}$ that covers more space, shrinking $\boldsymbol{P}_\mathrm{s}(\boldsymbol{I} - \boldsymbol{P}_\mathrm{w})$ and potentially leading to a smaller **PredGap**.

**Propagation of Errors.** The earlier intuition is reflected in the proof (App. A.7). Given the labeling $\hat{\boldsymbol{y}}$, its projection $(\boldsymbol{I} - \boldsymbol{P}_\mathrm{w})\hat{\boldsymbol{y}}$ is orthogonal to the scaled weak model's principal kernel and thus cannot be effectively learned, contributing to the weak model's error (Lem. A.12). The projection of this error onto the scaled strong model's principal kernel, $\boldsymbol{P}_\mathrm{s}(\boldsymbol{I} - \boldsymbol{P}_\mathrm{w})\hat{\boldsymbol{y}}$, is learned by the strong model and contributes to **PredGap** (Lem. A.13).

## 4. A Case Study on Benign Overfitting

Our theory can be applied to study and provide new insights into *benign overfitting*, an intriguing special case of W2SG, where the W2S model appears to mimic the weak supervision during finetuning, yet generalizes better at test time.

### 4.1. A general condition

Benign overfitting has been studied in the general machine learning context to understand deep neural networks' generalization (Bartlett et al., 2020; Wang et al., 2021; Frei et al., 2022; Mallinar et al., 2022). Recently, (Wu & Sahai, 2024) theoretically characterized benign overfitting in W2SG for a specific data distribution. Here, we aim to derive broader insights from a representation perspective. We consider the scenario where the strong model's representations are highly expressive, enabling near-perfect overfitting of arbitrary labelings on the finetuning data, mirroring the behavior of very large neural networks in practice (Zhang et al., 2021). This occurs when $\delta_\mathrm{s} = o(\hat{\gamma}_\mathrm{s})$ (Lem. B.4), yielding a highly isotropic non-principal kernel. Meanwhile, since generalization depends solely on the principal representations by Thm. 3.8, a small $\|\boldsymbol{P}_\mathrm{s}(\boldsymbol{I} - \boldsymbol{P}_\mathrm{w})\frac{1}{\sqrt{\hat{n}}}\hat{\boldsymbol{y}}\|^2$ suffices for good W2SG performance, regardless of the extent of overfitting. In this way, we connect benign overfitting to the general relationship between the weak and strong models' representations:

**Theorem 4.1** (A general condition for benign overfitting [1] ). *In addition to Assumption 3.7, suppose that (1) $\delta_s = o(\hat{\gamma}_s)$*

---

[1] Thm 4.1 can be extended to cases where the strong ceiling is not perfect, but we omit this for brevity.

*and* $\delta_s \le \beta_s = o(\hat{\gamma}_s)$*, (2) w.h.p., the strong ceiling model achieves nearly perfect performance, i.e.,* $\mathrm{Err}_{sc} = o(1)$*, (3) w.h.p.,* $\|\boldsymbol{P}_s(\boldsymbol{I} - \boldsymbol{P}_w)\frac{1}{\sqrt{\hat{n}}}\hat{\boldsymbol{y}}\|^2 = \mathrm{Err}_w - \Delta$ *with* $\Delta = \Theta(1)$*. Then, w.h.p., the W2S model achieves an almost zero* $(o(1))$ *training error on* $\hat{\mathcal{D}}$*, but generalizes better than the weak model:* $\mathrm{Err}_{w2s} \le \mathrm{Err}_w - \Delta + o(1)$*. See proof in App. B.3.1.*

*Remark.* Compared to (Wu & Sahai, 2024), which focuses on demonstrating that benign overfitting can occur under specific assumptions—such as a bi-level ensemble structure and labels depending 1-sparsely on representations—we extract more general insights into when and how benign overfitting arises. Specifically, we identify a single key quantity driving benign overfitting in W2SG: $\|\boldsymbol{P}_s(\boldsymbol{I} - \boldsymbol{P}_w)\frac{1}{\sqrt{\hat{n}}}\hat{\boldsymbol{y}}\|$. When this quantity is small, the strong model can avoid repeating the weak model's mistakes—regardless of the extent of overfitting—thereby achieving error mitigation. This precise mechanism was not revealed in prior work.

### 4.2. Instantiation of Theorem 4.1 on a toy example

We present a concrete example of the scenario in Theorem 4.1 to demonstrate the realizability of the conditions. While more complex examples could be constructed, we focus on a simple one to succinctly illustrate the core ideas.

**Example 4.2.** The label is a Gaussian: $y \sim \mathcal{N}(0, 1)$. Given $(\boldsymbol{x}, y)$, the weak model's representation is $h_w(\boldsymbol{x}) = [(\sqrt{\eta}\, y + \sqrt{1-\eta}\, \zeta) \quad \boldsymbol{\xi}_w^\top]^\top$, where $\eta \in (0, 1)$ is some constant, $\zeta \sim \mathcal{N}(0, 1)$ and $\boldsymbol{\xi}_w \sim \mathcal{N}(0, \frac{\sigma^2}{d-1}\boldsymbol{I})$ are both independently drawn. The strong model's representation is $h_s(\boldsymbol{x}) = [y \quad \boldsymbol{\xi}_s^\top]^\top$, where $\boldsymbol{\xi}_s \sim \mathcal{N}(0, \frac{\sigma^2}{d-1}\boldsymbol{I})$ independently. The scalings satisfy $\tilde{n} = \Theta(\hat{n}) = \omega(1)$, $d = \omega(\hat{n}^2)$, and $\sigma^2 = o(\hat{n})$ but $\ne 0$. Additionally, $\beta_s = o(\frac{\sigma^2}{\hat{n}})$ and $\beta_w = o(\frac{\sigma^2}{\hat{n}})$.

Here, the weak model's first coordinate carries a signal about the label $y$, but corrupted by noise $\zeta$, with $\eta$ controlling the signal strength (i.e., with SNR $\frac{\eta}{1-\eta}$). The strong model's first coordinate carries a perfect signal about $y$. The remaining coordinates in both models are high-dimensional random noise. Both models' representations are special cases of Example 3.5 and are therefore $(0, \frac{\sigma^2}{\hat{n}}, \frac{\sigma^2}{\hat{n}})$ decomposable.

**Corollary 4.3.** *Benign overfitting occurs in Example 4.2. Specifically, w.h.p., (1) The weak model's errors on both* $\hat{\mathcal{D}}$ *and the population are* $(1-\eta)\pm o(1)$*. (2) The W2S model overfits the weak model's outputs on* $\hat{\mathcal{D}}$*, achieving a training loss of* $o(1)$*. (3) However, compared to the weak model, the W2S model achieves a smaller test error:* $\mathrm{Err}_{w2s} = (1-\eta)^2 \pm o(1)$*.*

For instance, if $\eta = 0.6$, then $\mathrm{Err}_w \approx 0.4$, while $\mathrm{Err}_{w2s} \approx 0.16$, despite nearly perfect overfitting on $\hat{\mathcal{D}}$.

### 4.3. A closer look at error propagation

We provide a rough derivation of the W2S error (with details in App. B.3.2), illustrating which errors are replicated and

which are corrected (overfitted but benignly) by the W2S model, and how representations determine this.

The principal representations for both models are simply at their first coordinates. Thus, the spans of their principal kernels are one-dimensional. Let $\hat{\boldsymbol{\zeta}} \in \mathbb{R}^{\hat{n}}$ denote the vector collecting the $\zeta$ values on $\hat{\mathcal{D}}$, i.e., $\hat{\boldsymbol{\zeta}} = [\hat{\zeta}_1, \ldots, \hat{\zeta}_{\hat{n}}]^\top$. Similarly, define $\hat{\boldsymbol{y}} = [\hat{y}_1, \ldots, \hat{y}_{\hat{n}}]^\top$. We can approximate the projection matrices as: $\boldsymbol{P}_w \approx \frac{1}{\hat{n}}\hat{\boldsymbol{q}}\hat{\boldsymbol{q}}^\top$ and $\boldsymbol{P}_s \approx \frac{1}{\hat{n}}\hat{\boldsymbol{y}}\hat{\boldsymbol{y}}^\top$, where $\hat{\boldsymbol{q}} = \sqrt{\eta}\hat{\boldsymbol{y}} + \sqrt{1-\eta}\hat{\boldsymbol{\zeta}}$. Note that vectors $\frac{1}{\sqrt{\hat{n}}}\hat{\boldsymbol{y}}$ and $\frac{1}{\sqrt{\hat{n}}}\hat{\boldsymbol{\zeta}}$ are almost orthogonal as the corresponding random variables are uncorrelated: $\frac{1}{\sqrt{\hat{n}}}\hat{\boldsymbol{y}}^\top \frac{1}{\sqrt{\hat{n}}}\hat{\boldsymbol{\zeta}} = \frac{1}{\hat{n}}\sum_i \hat{y}_i\hat{\zeta}_i \approx \mathbb{E}[y\zeta] = 0$. Let $\boldsymbol{\epsilon}_w$ be the vector whose $i$-th element is the weak model's error on data point $(\hat{\boldsymbol{x}}_i, \hat{y}_i)$. By Lemma A.12, we can approximate $\boldsymbol{\epsilon}_w$ as:

$$\boldsymbol{\epsilon}_w \approx (\boldsymbol{I} - \boldsymbol{P}_w)\hat{\boldsymbol{y}} \approx (1-\eta)\frac{1}{\sqrt{\hat{n}}}\hat{\boldsymbol{y}} - \sqrt{\eta(1-\eta)}\frac{1}{\sqrt{\hat{n}}}\hat{\boldsymbol{\zeta}}$$

The strong ceiling model's error $\mathrm{Err}_{sc} \approx 0$ as its representations directly encode $y$ in the first coordinate. Thus, $\mathrm{Err}_{w2s} \approx \textbf{PredGap}$. By Thm 3.8, $\textbf{PredGap} \approx \boldsymbol{P}_s\boldsymbol{\epsilon}_w$. Then,

$$\mathrm{Err}_{w2s} \approx \underbrace{\frac{1}{\hat{n}}\hat{\boldsymbol{y}}\hat{\boldsymbol{y}}^\top(1-\eta)\frac{1}{\sqrt{\hat{n}}}\hat{\boldsymbol{y}}}_{\textbf{replicated}} \underbrace{-\frac{1}{\hat{n}}\hat{\boldsymbol{y}}\hat{\boldsymbol{y}}^\top\sqrt{\eta(1-\eta)}\frac{1}{\sqrt{\hat{n}}}\hat{\boldsymbol{\zeta}}}_{\textbf{avoided}; \approx 0 \text{ since } \hat{\boldsymbol{\zeta}} \perp \hat{\boldsymbol{y}} \text{ almost}}$$

The first term of the weak model's error, $(1-\eta)\frac{1}{\sqrt{\hat{n}}}\hat{\boldsymbol{y}}$, aligns with $\boldsymbol{P}_s$ which spans the strong model's principal kernel, and is therefore replicated by the W2S model. The second term, $-\sqrt{\eta(1-\eta)}\frac{1}{\sqrt{\hat{n}}}\hat{\boldsymbol{\zeta}}$, is orthogonal to $\boldsymbol{P}_s$ and thus mitigated. Notably, $-\sqrt{\eta(1-\eta)}\frac{1}{\sqrt{\hat{n}}}\hat{\boldsymbol{\zeta}}$ aligns with the strong model's non-principal kernel, which is highly isotropic ($\gamma_s = \omega(\delta_s)$), causing the corresponding errors to appear mimicked by the W2S model during finetuning. However, they do not manifest at test time. In other words, only errors within the span of the strong model's principal kernel are overfitted harmfully, while overfitting elsewhere remains benign.

## 5. Predicting W2SG Without Labels

Leveraging Thm. 3.8, we derive a representation-based metric that can predict W2SG performance without labels in experiments across various settings. Notably, this metric strongly correlates with W2SG performance even when we finetune entire LLMs—a scenario significantly more complex than what we analyze in theory.

### 5.1. A label-agnostic metric for W2SG

We start with upper-bounding the RHS of Thm. 3.8.

**Corollary 5.1** (Upper Bound 1)**.** *Define* $C = \frac{1}{\hat{n}}\sum_{i=1}^{\hat{n}} \hat{y}_i^2$*. Following Theorem 3.8, directly applying the submultiplicative property of the norm yields the following upper bound:*

$$\textbf{PredGap} \le C\|\boldsymbol{P}_s(\boldsymbol{I} - \boldsymbol{P}_w)\|_{\mathrm{op}}^2 + o(1),$$

Table 1: An overview of the three setups considered in our experiments.

| EXP ID | Task | Strong model | Weak models | Finetuning |
|--------|------|--------------|-------------|------------|
| I | molecular tasks | MolBERT | 150 transformers pretrained on GuacaMol | task head |
| II | NLP tasks | nvidia/NV-Embed-v2 | 22 other embedding models | task head |
| III | NLP tasks | Qwen/Qwen-7B | 28 smaller LLMs | full model |

**Corollary 5.2** (Upper Bound 2). *Following Theorem 3.8, we can also obtain an upper bound that involves $Err_{sc}$ as long as $|\mathbb{E}[y^2] - \frac{1}{\hat{n}} \sum_{i=1}^{\hat{n}} \hat{y}_i^2| = o(1)$ (see proof in Appendix B.4) :*

$$\textbf{PredGap} \leq \left( \sqrt{C} \|\boldsymbol{P}_s(\boldsymbol{I} - \boldsymbol{P}_w)\boldsymbol{P}_s\|_{\text{op}} + \sqrt{Err_{sc}} \right)^2 + o(1).$$

In both upper bounds, $C$ represents the variance of the labels on $\hat{\mathcal{D}}$, which can be treated as a constant given a fixed dataset. Therefore, **PredGap** is governed by the norm $\|\boldsymbol{P}_s(\boldsymbol{I} - \boldsymbol{P}_w)\|_{\text{op}}$ or $\|\boldsymbol{P}_s(\boldsymbol{I} - \boldsymbol{P}_w)\boldsymbol{P}_s\|_{\text{op}}$. Comparing the two bounds, the one in Corollary 5.2 is tighter particularly when $Err_{sc}$ is small [2]. This follows from $\|\boldsymbol{P}_s(\boldsymbol{I}-\boldsymbol{P}_w)\boldsymbol{P}_s\|_{\text{op}} \leq \|\boldsymbol{P}_s(\boldsymbol{I}-\boldsymbol{P}_w)\|_{\text{op}}$. However, in our experiments, both are similarly indicative of W2SG performance.

Now that **PredGap** can be bounded in terms of the above label-agnostic metrics, and **PredGap** is indicative of the error $Err_{w2s}$ as discussed at the end of Sec. 3.2, we turn our focus to examining the following relationship in real models

$$Err_{w2s} \overset{?}{\sim} \|\boldsymbol{P}_s(\boldsymbol{I} - \boldsymbol{P}_w)\|_{\text{op}} \text{ (or } \|\boldsymbol{P}_s(\boldsymbol{I} - \boldsymbol{P}_w)\boldsymbol{P}_s\|_{\text{op}})$$

to evaluate whether the metrics offer practical insights. Specifically, we consider the three setups summarized in Table 1, with their details discussed in the corresponding subsections. In each setup, we fix the strong model and vary the weak model to obtain different $Err_{w2s}$ and $\|\boldsymbol{P}_s(\boldsymbol{I} - \boldsymbol{P}_w)\|_{\text{op}}$ (or $\|\boldsymbol{P}_s(\boldsymbol{I} - \boldsymbol{P}_w)\boldsymbol{P}_s\|_{\text{op}}$) pairs and study their relationship.

### 5.2. Empirical measure of $\boldsymbol{P}_w$ and $\boldsymbol{P}_s$

Before proceeding, let's address an important question: how can we compute $\boldsymbol{P}_w$ and $\boldsymbol{P}_s$ for real models? In some cases, representations are not fixed during fine-tuning, making $h$ difficult to define. Additionally, determining the principal representation, $\boldsymbol{\Pi}_{\mathcal{V}} h$, is challenging because the exact $\mathcal{V}$ depends on the population, which is unknown in practice. To tackle this, we design heuristics to approximate $\boldsymbol{P}$ as follows

$$\frac{1}{\hat{n}} \hat{\boldsymbol{K}}(\boldsymbol{\Pi}_\alpha h)(\frac{1}{\hat{n}} \hat{\boldsymbol{K}}(\boldsymbol{\Pi}_\alpha h) + \beta_{\text{eff}} \boldsymbol{I})^{-1} \qquad (2)$$

We explain the key components below.

$h$**: extracting representations.** We consider two ways of defining the representations, depending on the setup. **(1) Last layer embeddings.** In Exps. I and II, the definition of representation is self-evident, as finetuning is simply

training a task head on the embeddings produced by the base model [3]. **(2) Activation maps.** [4] In Exp. III, we finetune the entire LLM from pretrained weights, so we don't have fixed representations as in the theoretical setting. To address this, we adopt a simple heuristic: we treat the layer-wise normalized vectorized activation maps of the pre-trained LLM, which encode information about how inputs are represented within the model, as the representations for computing $h(\boldsymbol{x})$. This heuristic serves primarily as a proof of concept, demonstrating that even straightforward approach like this can yield meaningful results. More principled definitions of representations, e.g., those based on NTK (Malladi et al., 2023) or representation engineering (Zou et al., 2023), could be explored in future work. See further discussion in Appx. E.

$\boldsymbol{\Pi}_\alpha$**: approximating principal representations.** We consider two versions of $\boldsymbol{\Pi}_\alpha$, the operation that extracts the principal part from the representations, based on the intuition that principal representations tend to have larger magnitudes (e.g., Example 3.5). (1) In Exps. I and II, we apply PCA by projecting the representations onto the eigenvectors of the covariance $\hat{\boldsymbol{\Sigma}}(h)$ with eigenvalues $\geq \alpha \times$ (the largest eigenvalue). (2) In Exp. III, we select the top coordinates with variance exceeding $\alpha \times$ (the largest coordinate-wise variance), a cheaper alternative to PCA for high-dimensional activation maps, as it avoids the expensive eigendecomposition. In both cases $\alpha$ is a hyperparameter.

$\beta_{\text{eff}}$**: effective regularization.** In Thm. 3.8, $(\beta + \hat{\gamma})$ is the effective regularization, capturing both the explicit ($\beta$) and implicit ($\hat{\gamma}$) (Jacot et al., 2020) regularization. In practice, regularization can also stem from factors like early stopping, training algorithms, etc. We summarize these effects using $\beta_{\text{eff}}$ in Eq. 2 and treat $\beta_{\text{eff}}$ as a hyperparameter.

For each model, computing $\boldsymbol{P}$ introduces two hyperparameters, $\alpha$ and $\beta$. If every model is assigned unique hyperparameters, the total number of hyperparameters would be twice the number of models. To simplify this, we let all weak models share the same two hyperparameters, $\alpha_w$ and $\beta_w$. For the strong model (only one in each setting), it is treated separately with its own hyperparameters, $\alpha_s$ and $\beta_s$. Thus, we only have four parameters in total. More details are in App. D.2.

---

[3] In the analysis, the linear model does not include a bias term, but it does in our experiments. This is addressed by appending a constant 1 to the representation when computing the metrics.

[4] We observed worse results with last-layer embeddings in Exp. III, likely due to complex cross-layer dynamics during finetuning.

---

[2] One can also observe this in Example 4.2, where the equality in Corollary 5.2 holds, whereas that in Corollary 5.1 does not.

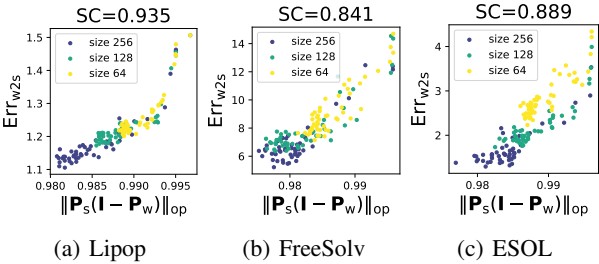

Figure 2: Results of Exp. I: our metric strongly correlates with $\text{Err}_{\text{w2s}}$ and serves as a more fine-grained indicator than model size.

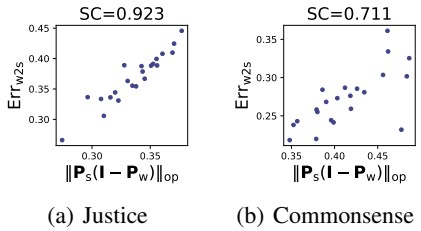

Figure 3: A strong correlation between $\|\boldsymbol{P}_{\text{s}}(\boldsymbol{I} - \boldsymbol{P}_{\text{w}})\|_{\text{op}}$ and $\text{Err}_{\text{w2s}}$ is observed in Exp. II where we finetune embedding models.

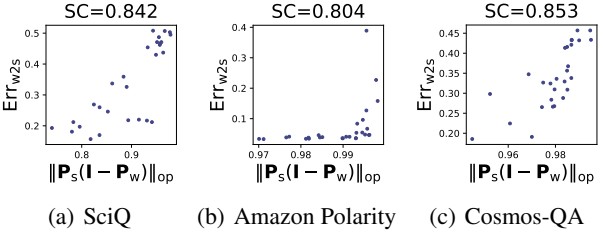

Figure 4: A strong correlation between $\|\boldsymbol{P}_{\text{s}}(\boldsymbol{I} - \boldsymbol{P}_{\text{w}})\|_{\text{op}}$ and $\text{Err}_{\text{w2s}}$ is observed in Exp. III involving general-purpose LLMs.

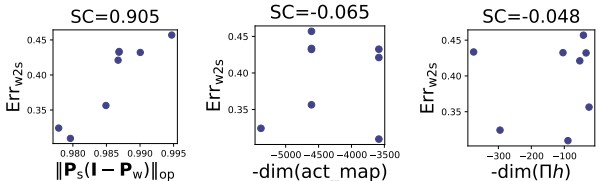

Figure 5: In Exp. III, for models with activation map dimensions $\leq 8000$, both the activation map dimension (middle) and the dimension of approximated principal representations (right) correlate poorly with $\text{Err}_{\text{w2s}}$. However, $\|\boldsymbol{P}_{\text{s}}(\boldsymbol{I} - \boldsymbol{P}_{\text{w}})\|_{\text{op}}$ remains strongly correlated with $\text{Err}_{\text{w2s}}$ (left). We only show the results for Cosmos QA and defer those for other datasets to App. D.4.

## 5.3. Experimental setups

**Exp. I: Molecular prediction.** Our first setting follows (Charikar et al., 2024). We use the GuacaMol (Brown et al., 2019) dataset for pretraining both the strong and weak models. For finetuning, we consider three regression datasets—ESOL, FreeSolv, and Lipop—from the MoleculeNet (Wu et al., 2018) benchmark, curated by ChemBench (Charleshen, 2020), which involve predicting molecular physical properties. The strong model is MolBERT (Fabian et al., 2020), a BERT (Devlin, 2018) pretrained for 100 epochs on GuacaMol. We use smaller transformers pretrained on GuacaMol as weak models. These weak models have 2 layers and 2 attention heads. We vary the hidden size across $64, 128, 256$, and vary the number of pretraining epochs from 1 to 50, resulting in 150 weak models. During finetuning, we extract last-layer embeddings and perform linear regression. MSE loss is used for both training and measuring $\text{Err}_{\text{w2s}}$ as the task is regression. Additional details are in App.D.1.

**Exp. II: NLP tasks with embedding models.** We use the "Justice" and "Commonsense" datasets from ETHICS (Hendrycks et al., 2020), which involve binary classification based on basic moral concepts. We consider embedding models—pretrained LLMs that convert text inputs into vector-based embeddings, with `nvidia/NV-Embed-v2` (Lee et al., 2024) (currently ranked first on the MTEB leaderboard (Muennighoff et al., 2022)) as the strong model, and 22 other models as weak models (details in Appx. D.1). For finetuning, we train a linear classifier on the embeddings with CE loss. $\text{Err}_{\text{w2s}}$ is measured as classification error.

**Exp. III: NLP tasks with end-to-end finetuned LLMs.** We replicate a setup from (Burns et al., 2023) on three datasets: (1) SciQ (Welbl et al., 2017), containing crowd-sourced science exam questions; (2) Amazon Polarity (Zhang et al., 2015), consisting of Amazon reviews; and (3) Cosmos QA (Huang et al., 2019), involving commonsense-based reading comprehension. Both data preprocessing and finetuning strictly follow (Burns et al., 2023). The entire model is finetuned with the unembedding layer replaced with a linear head, using CE loss. We use `Qwen/Qwen-7B` (Bai et al., 2023) as the strong model and 28 smaller LLMs as weak models (details in Appx. D.1). $\text{Err}_{\text{w2s}}$ is measured in terms of classification error.

## 5.4. Results

**Strong correlation between $\text{Err}_{\text{w2s}}$ and $\|\boldsymbol{P}_{\text{s}}(\boldsymbol{I} - \boldsymbol{P}_{\text{w}})\|_{\text{op}}$ across various settings.** For each of the weak models, we perform the W2SG procedure to obtain the resulting W2S model. We then measure $\text{Err}_{\text{w2s}}$ and $\|\boldsymbol{P}_{\text{s}}(\boldsymbol{I} - \boldsymbol{P}_{\text{w}})\|_{\text{op}}$ and plot the results in Figures 2, 3 and 4. Across all the setups, we observe a strong correlation between the two quantities, with high Spearman's correlation values displayed at

the top of the figures. The results are highly similar for $\|\boldsymbol{P}_s(\boldsymbol{I} - \boldsymbol{P}_w)\boldsymbol{P}_s\|_{op}$, as shown in Appx. D.3. Therefore, we only focus on discussing $\|\boldsymbol{P}_s(\boldsymbol{I} - \boldsymbol{P}_w)\|_{op}$ in the main paper. Notably, the correlation between $\mathrm{Err}_{w2s}$ and $\|\boldsymbol{P}_s(\boldsymbol{I} - \boldsymbol{P}_w)\|_{op}$ extends beyond the theoretical setting, covering the following variations: *(1) Loss function and evaluation metric.* While Thm. 3.8 is based on linear regression with MSE loss, Exps. II and III demonstrate that the correlation also holds for classification tasks using CE finetuning loss, with $\mathrm{Err}_{w2s}$ measured as classification error. *(2) The form of finetuning.* Thm. 3.8 assumes that finetuning involves training a function on fixed representations. However, in Exp. III, the entire LLM is finetuned. Despite the complex training dynamics in this scenario, a strong correlation between $\mathrm{Err}_{w2s}$ and $\|\boldsymbol{P}_s(\boldsymbol{I} - \boldsymbol{P}_w)\|_{op}$ is still observed when activation maps are heuristically used as representations. These results underscore the broad applicability of our conclusion.

**Capturing W2SG beyond model size.** Smaller weak models can sometimes achieve better $\mathrm{Err}_{w2s}$ than larger ones. For example, in Exp. I, the leftmost yellow point (size 64) outperforms the rightmost teal point (size 128) in Fig. 2, likely because these smaller models were pretrained for more epochs (recall that we have 150 models span different combinations of sizes and pretraining epochs), resulting in better representations. Similarly, in Exp. III, the middle column of Fig. 5 shows a poor correlation between $\mathrm{Err}_{w2s}$ and size for models with dimension $\leq 8000$. Testing another dimension-based metric—the dimension of approximated principal representations—also reveals weak correlation with $\mathrm{Err}_{w2s}$ (last column of Fig. 5). This underscore the complexity of predicting W2SG performance, as larger models or higher representation dimensions do not guarantee better results. Factors such as the pretraining recipe, the quality and relevance of the pretraining data, etc., all contribute to the final outcome. However, even in these cases, $\|\boldsymbol{P}_s(\boldsymbol{I} - \boldsymbol{P}_w)\|_{op}$ consistently captures the trend in $\mathrm{Err}_{w2s}$ (Fig. 2 and the first column of Fig. 5), demonstrating its robustness as a metric that surpasses simple dimensional measures and provides meaningful insights for W2SG.

## 6. Conclusion

In this work, we show that W2SG can be characterized using kernels derived from the principal components of weak and strong models' representations. The theory is applicable to a wide range of representation distributions, provides insights into how models' internal structures influence error correction and the conditions for benign overfitting. Additionally, it offers a label-free metric for predicting W2SG performance, validated through experiments on diverse datasets and LLMs.

## Impact Statement

We see positive societal impacts in our work as it advances the understanding of Weak-to-Strong Generalization, a crucial problem for aligning superhuman AI in the future. Our results could enhance transparency in AI systems' behavior through analysis of their internal structures and contribute to the broader goal of improving AI safety and reliability.

## Acknowledgement

This research was partially supported by the National Science Foundation CAREER Award 2146492 and an OpenAI SuperAlignment Grant.

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

# A. Main Analysis

In this section, we provide a thorough analysis of the errors associated with the weak model, the W2S model, and the strong ceiling model. Some of these results are used to prove our main conclusion, Theorem 3.8, while others are applied in subsequent analyses.

## A.1. Notations and additional notes

**Symbol definitions.** We introduce the following notations. The symbol $r$ represents a representation, i.e., $r = h(x)$. For the samples in the splits $\tilde{\mathcal{D}}$ and $\hat{\mathcal{D}}$, we denote their representations as $\tilde{r}_1, \ldots, \tilde{r}_{\tilde{n}}$ and $\hat{r}_1, \ldots, \hat{r}_{\hat{n}}$, respectively. We define the sample representation matrices, where each column corresponds to a representation:

$$\tilde{R} := [\tilde{r}_1 \ \tilde{r}_2 \ \ldots \tilde{r}_{\tilde{n}}] \quad \text{and} \quad \hat{R} := [\hat{r}_1 \ \hat{r}_2 \ \ldots \hat{r}_{\hat{n}}].$$

We also define $y$ which collects the labels of the samples:

$$\tilde{y} = \begin{bmatrix} \tilde{y}_1 \\ \tilde{y}_2 \\ \vdots \\ \tilde{y}_{\tilde{n}} \end{bmatrix} \quad \text{and} \quad \hat{y} = \begin{bmatrix} \hat{y}_1 \\ \hat{y}_2 \\ \vdots \\ \hat{y}_{\hat{n}} \end{bmatrix}.$$

For the covariance matrices, we use the following shorthand notations to avoid clutter:

$$\Sigma = \Sigma(h), \ \hat{\Sigma} = \hat{\Sigma}(h), \ \tilde{\Sigma} = \tilde{\Sigma}(h),$$
$$\Sigma' = \Sigma(\Pi_{\mathcal{V}} h), \ \hat{\Sigma}' = \hat{\Sigma}(\Pi_{\mathcal{V}} h), \ \tilde{\Sigma}'' = \tilde{\Sigma}(\Pi_{\mathcal{V}} h), \ \Sigma'' = \Sigma(\Pi_{\mathcal{V}^\perp} h), \ \hat{\Sigma}'' = \hat{\Sigma}(\Pi_{\mathcal{V}^\perp} h), \ \tilde{\Sigma}'' = \tilde{\Sigma}(\Pi_{\mathcal{V}^\perp} h).$$

**Use of subscripts.** Additionally, we use subscripts 'w' and 's' to indicate the model associated with a given quantity. For example, $\tilde{R}_w$ and $\hat{R}_w$ denote the sample representation matrices generated by the weak model, while $\tilde{R}_s$ and $\hat{R}_s$ denote those generated by the strong model. Similarly, this convention applies to covariance matrices; for instance, $\hat{\Sigma}'_s = \hat{\Sigma}(\Pi_{\mathcal{V}_s} h_s)$.

**Mathematical notations.** For convenience, whenever we say $A = B + o(1)$, where $A$ and $B$ are matrices or vectors, we mean that $\|A - B\|_{\text{op}} = o(1)$. We let $\lambda_i(A)$, $\lambda_{\min}(A)$, $\lambda_{\min, \neq 0}(A)$, and $\lambda_{\max}(A)$ represent the $i$-th, smallest, smallest nonzero, and largest eigenvalues of the matrix $A$, respectively. The expression $A \preccurlyeq B$ means that the matrix $B - A$ is positive semidefinite, and $A \succcurlyeq B$ means that $A - B$ is positive semidefinite.

**Implied proof techniques.** Sometimes, in the proof, we use the triangle inequality and the sub-multiplicativity of norms without explicitly stating them when they are straightforward, as mentioning them would make the text unnecessarily verbose.

## A.2. Restatement of Definition 3.3

Here, we restate Definition 3.3 with simplified notations for convenience and clarity in the proof.

**Definition A.1** $((\delta, \hat{\gamma}, \tilde{\gamma})$-decomposability (restated) ). Given $\mathcal{D}, \tilde{\mathcal{D}}, \hat{\mathcal{D}}$, and a representation function $h$, we say that the representations of $h$ are $(\delta, \hat{\gamma}, \tilde{\gamma})$-*decomposable with respect to a subspace* $\mathcal{V}$ (of the representation space), for some $\delta = O(1)$, $\hat{\gamma} = O(1)$, and $\tilde{\gamma} = O(1)$, if the following holds. Let $U\Lambda U^\top$ be the singular value decomposition (SVD) of $\Sigma$. There exists a matrix $U'$ consisting of a subset of columns of $U$, corresponding to the nonzero eigenvalues, such that the following conditions are satisfied. Let $U''$ denote the matrix that collects the remaining columns of $U$. Define diagonal matrices $\Lambda'$ and $\Lambda''$ to collect the eigenvalues corresponding to $U'$ and $U''$, respectively. Additionally, define: $\Sigma' = U'\Lambda'U'^\top$ and $\Sigma'' = U''\Lambda''U''^\top$. Let $\gamma = \min(\hat{\gamma}, \tilde{\gamma})$, and let $\mathcal{V}$ be the span of the columns of $U'$. Now, leveraging the fact that the projection $\Pi_{\mathcal{V}}$ can be written as $U'U'^\top$, and noting that $\lambda_{\min, \neq 0}(\Sigma') = \lambda_{\min}(\Lambda')$, we can reformulate the original Definition 3.3 in terms of $U'$: with high probability $1 - o(1)$,

    a. **Boundedness**. $\|\Sigma\|_{\text{op}} = O(1)$, $\|\hat{\Sigma}\|_{\text{op}} = O(1)$ and $\|\tilde{\Sigma}\|_{\text{op}} = O(1)$. Additionally, $\mathbb{E}[y^2] = O(1)$, $\frac{1}{\hat{n}}\|\hat{y}\|^2 = O(1)$ and $\frac{1}{\tilde{n}}\|\tilde{y}\|^2 = O(1)$.

b. **Concentration on $\mathcal{V}$.** The original statement is $\|\hat{\boldsymbol{\Sigma}}' - \boldsymbol{\Sigma}'\|_{\mathrm{op}} = o(1)$ and $\|\tilde{\boldsymbol{\Sigma}}' - \boldsymbol{\Sigma}'\|_{\mathrm{op}} = o(1)$. However, since:

$$\|\boldsymbol{U}'^{\top}\hat{\boldsymbol{\Sigma}}\boldsymbol{U}' - \boldsymbol{\Lambda}'\|_{\mathrm{op}} = \|\frac{1}{\hat{n}}\boldsymbol{U}'^{\top}\hat{\boldsymbol{R}}\hat{\boldsymbol{R}}^{\top}\boldsymbol{U}' - \boldsymbol{\Lambda}'\|_{\mathrm{op}}$$

$$= \|\frac{1}{\hat{n}}\boldsymbol{U}\boldsymbol{U}'^{\top}\hat{\boldsymbol{R}}\hat{\boldsymbol{R}}^{\top}\boldsymbol{U}'\boldsymbol{U}^{\top} - \boldsymbol{U}\boldsymbol{U}'^{\top}\boldsymbol{\Lambda}\boldsymbol{U}'\boldsymbol{U}^{\top}\|_{\mathrm{op}}$$

$$= \|\hat{\boldsymbol{\Sigma}}' - \boldsymbol{\Sigma}'\|_{\mathrm{op}},$$

and similarly for $\tilde{\boldsymbol{\Sigma}}'$, we can restate it as: $\|\boldsymbol{U}'^{\top}\hat{\boldsymbol{\Sigma}}\boldsymbol{U}' - \boldsymbol{\Lambda}'\|_{\mathrm{op}} = o(\gamma^2 + \delta^2 + \lambda_{\min}(\boldsymbol{\Lambda}')^2)$ and $\|\boldsymbol{U}'^{\top}\tilde{\boldsymbol{\Sigma}}\boldsymbol{U}' - \boldsymbol{\Lambda}'\|_{\mathrm{op}} = o(\gamma^2 + \delta^2 + \lambda_{\min}(\boldsymbol{\Lambda}')^2)$. Similarly, by noting that the operator norm is invariant under left multiplication by $\boldsymbol{U}'$, we can restate the statement regarding $y$ as: $\|\boldsymbol{U}'^{\top}\frac{1}{\sqrt{\tilde{n}}}\tilde{\boldsymbol{R}}\tilde{\boldsymbol{y}} - \boldsymbol{U}'^{\top}\mathbb{E}[ry]\| = o(\gamma + \delta + \lambda_{\min}(\boldsymbol{\Lambda}'))$ and $\|\boldsymbol{U}'^{\top}\frac{1}{\sqrt{\hat{n}}}\hat{\boldsymbol{R}}\hat{\boldsymbol{y}} - \boldsymbol{U}'^{\top}\mathbb{E}[ry]\| = o(\gamma + \delta + \lambda_{\min}(\boldsymbol{\Lambda}'))$.

c. **Kernel-wise $\delta$-isotropy on $\mathcal{V}^{\perp}$.** $\|\frac{1}{\hat{n}}\hat{\boldsymbol{R}}^{\top}\boldsymbol{U}''\boldsymbol{U}''^{\top}\hat{\boldsymbol{R}} - \hat{\gamma}\boldsymbol{I}\|_{\mathrm{op}} = o(\gamma^2 + \delta^2)$ and $\|\frac{1}{\tilde{n}}\tilde{\boldsymbol{R}}^{\top}\boldsymbol{U}''\boldsymbol{U}''^{\top}\tilde{\boldsymbol{R}} - \tilde{\gamma}\boldsymbol{I}\|_{\mathrm{op}} = o(\gamma^2 + \delta^2)$.

d. **Small cross-sample inner-product on $\mathcal{V}^{\perp}$.** $\|\frac{1}{\sqrt{\hat{n}}}\hat{\boldsymbol{R}}^{\top}\boldsymbol{U}''\boldsymbol{U}''^{\top}\frac{1}{\sqrt{\tilde{n}}}\tilde{\boldsymbol{R}}\|_{\mathrm{op}} = o(\gamma + \delta)$.

e. **Diminishing population covariance on $\mathcal{V}^{\perp}$.** $\|\boldsymbol{\Sigma}''\|_{\mathrm{op}} = o(\gamma + \delta)$.

**Use of subscripts.** Since in Assumption 3.7 we assume that the representations of both the weak and strong models satisfy Definition A.1, all the notations in Definition A.1 have corresponding versions for the weak model's representations and the strong model's representations. We follow the previously mentioned convention and use the subscripts w' and s' to distinguish between them. For example, notations such as $\boldsymbol{U}'_{\mathrm{w}}$ and $\boldsymbol{U}'_{\mathrm{s}}$, $\boldsymbol{\Lambda}'_{\mathrm{w}}$ and $\boldsymbol{\Lambda}'_{\mathrm{s}}$, will be used. The meaning of such notations should be clear from the context in which they appear.

### A.3. Lemmas

Below, we introduce some basic lemmas and prove properties that will be used in the later analysis.

**Lemma A.2** (Push-through identity)**.** *For any matrices $\boldsymbol{A}, \boldsymbol{B}$, and any scalar $a$, the identity $(a\boldsymbol{I} + \boldsymbol{A}\boldsymbol{B})^{-1}\boldsymbol{A} = \boldsymbol{A}(a\boldsymbol{I} + \boldsymbol{B}\boldsymbol{A})^{-1}$ holds as long as $(a\boldsymbol{I} + \boldsymbol{A}\boldsymbol{B})^{-1}$ and $(a\boldsymbol{I} + \boldsymbol{B}\boldsymbol{A})^{-1}$ are invertible.*

**Lemma A.3.** *A classical result on the effect of perturbations on the inverse of a square matrix states that $\|(\boldsymbol{A} + \Delta)^{-1} - \boldsymbol{A}^{-1}\|_{\mathrm{op}} \leq \|\boldsymbol{A}^{-1}\|_{\mathrm{op}}^2\|\Delta\|_{\mathrm{op}}$, where $\boldsymbol{A}$ is an invertible square matrix. This result can be found, for example, in (Demmel, 1992) or Equation 1.1 of (El Ghaoui, 2002).*

**Lemma A.4.** *If condition **Kernel-wise $\delta$-isotropy on $\mathcal{V}^{\perp}$** holds, we have that $\|\frac{1}{\tilde{n}}\tilde{\boldsymbol{R}}^{\top}\tilde{\boldsymbol{R}} - \left(\frac{1}{\tilde{n}}\tilde{\boldsymbol{R}}^{\top}\boldsymbol{U}'\boldsymbol{U}'^{\top}\tilde{\boldsymbol{R}} + \tilde{\gamma}\boldsymbol{I}\right)\|_{\mathrm{op}} = o(\gamma^2 + \delta^2)$, and a similar conclusion holds for $\hat{\boldsymbol{R}}$ as well.*

*Proof.* By **Kernel-wise $\delta$-isotropy on $\mathcal{V}^{\perp}$** ,

$$\|\frac{1}{\tilde{n}}\tilde{\boldsymbol{R}}^{\top}\tilde{\boldsymbol{R}} - \left(\frac{1}{\tilde{n}}\tilde{\boldsymbol{R}}^{\top}\boldsymbol{U}'\boldsymbol{U}'^{\top}\tilde{\boldsymbol{R}} + \tilde{\gamma}\boldsymbol{I}\right)\|_{\mathrm{op}}$$

$$= \|\frac{1}{\tilde{n}}\tilde{\boldsymbol{R}}^{\top}(\boldsymbol{U}'\boldsymbol{U}'^{\top} + \boldsymbol{U}''\boldsymbol{U}''^{\top})\tilde{\boldsymbol{R}} - \left(\frac{1}{\tilde{n}}\tilde{\boldsymbol{R}}^{\top}\boldsymbol{U}'\boldsymbol{U}'^{\top}\tilde{\boldsymbol{R}} + \tilde{\gamma}\boldsymbol{I}\right)\|_{\mathrm{op}}$$

$$= \|\frac{1}{\tilde{n}}\tilde{\boldsymbol{R}}^{\top}\boldsymbol{U}''\boldsymbol{U}''^{\top}\tilde{\boldsymbol{R}} - \tilde{\gamma}\boldsymbol{I}\|_{\mathrm{op}}$$

$$= o(\gamma^2 + \delta^2).$$

$\square$

**Lemma A.5.** *If condition **Kernel-wise $\delta$-isotropy on $\mathcal{V}^{\perp}$** holds, then for any $\beta = O(1)$ s.t. $\beta \geq \delta$, we have that $\|(\frac{1}{\tilde{n}}\tilde{\boldsymbol{R}}^{\top}\tilde{\boldsymbol{R}} + \beta\boldsymbol{I})^{-1} - (\frac{1}{\tilde{n}}\tilde{\boldsymbol{R}}^{\top}\boldsymbol{U}'\boldsymbol{U}'^{\top}\tilde{\boldsymbol{R}} + (\tilde{\gamma} + \beta)\boldsymbol{I})^{-1}\|_{\mathrm{op}} = o(1)$, and a similar conclusion holds for $\hat{\boldsymbol{R}}$ as well.*

*Proof.* By **Kernel-wise $\delta$-isotropy on $\mathcal{V}^{\perp}$** ,

$$\|\frac{1}{\tilde{n}}\tilde{\boldsymbol{R}}^{\top}\tilde{\boldsymbol{R}} + \beta\boldsymbol{I} - \left(\frac{1}{\tilde{n}}\tilde{\boldsymbol{R}}^{\top}\boldsymbol{U}'\boldsymbol{U}'^{\top}\tilde{\boldsymbol{R}} + (\tilde{\gamma} + \beta)\boldsymbol{I}\right)\|_{\text{op}}$$

$$=\|\frac{1}{\tilde{n}}\tilde{\boldsymbol{R}}^{\top}(\boldsymbol{U}'\boldsymbol{U}'^{\top} + \boldsymbol{U}''\boldsymbol{U}''^{\top})\tilde{\boldsymbol{R}} + \beta\boldsymbol{I} - \left(\frac{1}{\tilde{n}}\tilde{\boldsymbol{R}}^{\top}\boldsymbol{U}'\boldsymbol{U}'^{\top}\tilde{\boldsymbol{R}} + (\tilde{\gamma} + \beta)\boldsymbol{I}\right)\|_{\text{op}}$$

$$=\|\frac{1}{\tilde{n}}\tilde{\boldsymbol{R}}^{\top}\boldsymbol{U}''\boldsymbol{U}''^{\top}\tilde{\boldsymbol{R}} - \tilde{\gamma}\boldsymbol{I}\|_{\text{op}}$$

$$=o(\gamma^2 + \delta^2).$$

Then, by Lemma A.3, we have

$$\|(\frac{1}{\tilde{n}}\tilde{\boldsymbol{R}}^{\top}\tilde{\boldsymbol{R}} + \beta\boldsymbol{I})^{-1} - (\frac{1}{\tilde{n}}\tilde{\boldsymbol{R}}^{\top}\boldsymbol{U}'\boldsymbol{U}'^{\top}\tilde{\boldsymbol{R}} + (\tilde{\gamma} + \beta)\boldsymbol{I})^{-1}\|_{\text{op}} \leq o(\gamma^2 + \delta^2) \; \|(\frac{1}{\tilde{n}}\tilde{\boldsymbol{R}}^{\top}\boldsymbol{U}'\boldsymbol{U}'^{\top}\tilde{\boldsymbol{R}} + (\tilde{\gamma} + \beta)\boldsymbol{I})^{-1}\|_{\text{op}}^2$$

$$=o(\frac{\gamma^2 + \delta^2}{(\tilde{\gamma} + \beta)^2})$$

$$=o(1).$$

$\square$

**Lemma A.6.** *If condition **Concentration on $\mathcal{V}$** holds, then for any $\beta = O(1)$ s.t. $\beta \geq \delta$, and $\gamma_0 \in \{\hat{\gamma}, \tilde{\gamma}\}$ we have*

$$\|(\boldsymbol{U}'^{\top}\tilde{\boldsymbol{\Sigma}}\boldsymbol{U}' + (\gamma_0 + \beta)\boldsymbol{I})^{-1} - (\boldsymbol{\Lambda}' + (\gamma_0 + \beta)\boldsymbol{I})^{-1}\|_{\text{op}} = o(1),$$

*and a similar conclusion holds for $\hat{\boldsymbol{\Sigma}}$ as well.*

*Proof.* By condition **Concentration on $\mathcal{V}$**, we have

$$\|\boldsymbol{U}'^{\top}\tilde{\boldsymbol{\Sigma}}\boldsymbol{U}' - \boldsymbol{\Lambda}'\|_{\text{op}} = o(\gamma^2 + \delta^2 + \lambda_{\min}(\boldsymbol{\Lambda}')^2).$$

Then, by Lemma A.3, we have

$$\|(\boldsymbol{U}'^{\top}\tilde{\boldsymbol{\Sigma}}\boldsymbol{U}' + (\gamma_0 + \beta)\boldsymbol{I})^{-1} - (\boldsymbol{\Lambda}' + (\gamma_0 + \beta)\boldsymbol{I})^{-1}\|_{\text{op}} = \leq o(\gamma^2 + \delta^2 + \lambda_{\min}(\boldsymbol{\Lambda}')^2) \; \|(\boldsymbol{\Lambda}' + (\gamma_0 + \beta)\boldsymbol{I})^{-1}\|_{\text{op}}^2$$

$$=o(\frac{\gamma^2 + \delta^2 + \lambda_{\min}(\boldsymbol{\Lambda}')^2}{(\gamma_0 + \beta + \lambda_{\min}(\boldsymbol{\Lambda}'))^2})$$

$$=o(1).$$

$\square$

**Lemma A.7.** *If conditions **Boundedness** and **Concentration on $\mathcal{V}$** hold, then $|\lambda_{\min}(\boldsymbol{\Lambda}')^2 - \lambda_{\min}(\boldsymbol{U}'^{\top}\hat{\boldsymbol{\Sigma}}\boldsymbol{U}')^2| = o(\gamma^2 + \delta + \lambda_{\min}(\boldsymbol{\Lambda}')^2)$. It still holds if we replace $\hat{}$ with $\tilde{}$.*

*Proof.* Define $t = \lambda_{\min}(\boldsymbol{\Lambda}') - \lambda_{\min}(\boldsymbol{U}'^{\top}\hat{\boldsymbol{\Sigma}}'\boldsymbol{U}')$. By condition **Concentration on $\mathcal{V}$** and Weyl's theorem, we have $|t| = o(\gamma^2 + \delta^2 + \lambda_{\min}(\boldsymbol{\Lambda}')^2)$. Then, we compute:

$$\lambda_{\min}(\boldsymbol{U}'^{\top}\hat{\boldsymbol{\Sigma}}'\boldsymbol{U}')^2$$

$$=\lambda_{\min}(\boldsymbol{\Lambda}')^2 + t^2 - 2t\lambda_{\min}(\boldsymbol{\Lambda}')$$

$$=\lambda_{\min}(\boldsymbol{\Lambda}')^2 \pm o(\gamma^2 + \delta^2 + \lambda_{\min}(\boldsymbol{\Lambda}')^2),$$

where the last step follows because $\lambda_{\min}(\boldsymbol{\Lambda}') = O(1)$ (via condition **Boundedness**) and $|t| = o(\gamma^2 + \delta^2 + \lambda_{\min}(\boldsymbol{\Lambda}')^2)$. $\square$

**Corollary A.8.** *Lemma A.7 further implies that $\frac{\gamma^2 + \delta^2 + \lambda_{\min}(\boldsymbol{\Lambda}')^2}{\hat{\gamma}^2 + \delta^2 + \lambda_{\min}(\boldsymbol{U}'^{\top}\hat{\boldsymbol{\Sigma}}\boldsymbol{U}')^2} = O(1)$ when conditions **Boundedness** and **Concentration on $\mathcal{V}$** hold. It still holds if we replace $\hat{}$ with $\tilde{}$.*

*Proof.*

$$\frac{\gamma^2 + \delta^2 + \lambda_{\min}(\boldsymbol{U'}^\top \hat{\boldsymbol{\Sigma}} \boldsymbol{U'})^2}{\hat{\gamma}^2 + \delta^2 + \lambda_{\min}(\boldsymbol{\Lambda'})^2} = \frac{\gamma^2 + \delta^2 + \lambda_{\min}(\boldsymbol{\Lambda'})^2}{\hat{\gamma}^2 + \delta^2 + \lambda_{\min}(\boldsymbol{\Lambda'})^2} - \frac{\lambda_{\min}(\boldsymbol{\Lambda'})^2 - \lambda_{\min}(\boldsymbol{U'}^\top \hat{\boldsymbol{\Sigma}} \boldsymbol{U'})^2}{\hat{\gamma}^2 + \delta^2 + \lambda_{\min}(\boldsymbol{\Lambda'})^2}$$

$$\leq 1 \pm \frac{o(\gamma^2 + \delta^2 + \lambda_{\min}(\boldsymbol{\Lambda'})^2)}{\hat{\gamma}^2 + \delta^2 + \lambda_{\min}(\boldsymbol{\Lambda'})^2}$$

$$\leq 1 + o(1).$$

Therefore, $\frac{\gamma^2 + \delta^2 + \lambda_{\min}(\boldsymbol{\Lambda'})^2}{\hat{\gamma}^2 + \delta^2 + \lambda_{\min}(\boldsymbol{U'}^\top \hat{\boldsymbol{\Sigma}} \boldsymbol{U'})^2} = O(1)$. $\qquad\square$

**Corollary A.9.** *If conditions **Boundedness** and **Concentration on** $\mathcal{V}$ hold, then for any $\boldsymbol{q}$ with $\|\boldsymbol{q}\| = O(1)$, we have* $\|\boldsymbol{U'}\sqrt{\boldsymbol{\Lambda'}}(\boldsymbol{U'}^\top\boldsymbol{\Sigma'}\boldsymbol{U'} + (\hat{\gamma}+\beta)\boldsymbol{I})^{-1}\boldsymbol{q}\|^2 = \|\frac{1}{\sqrt{\hat{n}}}\hat{\boldsymbol{R}}^\top\boldsymbol{U'}(\frac{1}{\hat{n}}\boldsymbol{U'}^\top\hat{\boldsymbol{R}}\hat{\boldsymbol{R}}^\top\boldsymbol{U'} + (\hat{\gamma}+\beta)\boldsymbol{I})^{-1}\boldsymbol{q}\|^2 \pm o(1)$. *It still holds if we replace* $\hat{}$ *with* $\tilde{}$.

*Proof.*

$$\|\boldsymbol{U'}\sqrt{\boldsymbol{\Lambda'}}(\boldsymbol{U'}^\top\boldsymbol{\Sigma'}\boldsymbol{U'} + (\hat{\gamma}+\beta)\boldsymbol{I})^{-1}\boldsymbol{q}\|^2$$

$$=\boldsymbol{q}^\top(\boldsymbol{U'}^\top\boldsymbol{\Sigma'}\boldsymbol{U'} + (\hat{\gamma}+\beta)\boldsymbol{I})^{-1}\boldsymbol{\Lambda'}(\boldsymbol{U'}^\top\boldsymbol{\Sigma'}\boldsymbol{U'} + (\hat{\gamma}+\beta)\boldsymbol{I})^{-1}\boldsymbol{q}$$

$$=\boldsymbol{q}^\top(\boldsymbol{U'}^\top\boldsymbol{\Sigma'}\boldsymbol{U'} + (\hat{\gamma}+\beta)\boldsymbol{I})^{-1}\boldsymbol{U'}^\top\hat{\boldsymbol{\Sigma}}\boldsymbol{U'}(\boldsymbol{U'}^\top\boldsymbol{\Sigma'}\boldsymbol{U'} + (\hat{\gamma}+\beta)\boldsymbol{I})^{-1}\boldsymbol{q}$$

$$\pm o\left((\gamma^2 + \delta^2 + \lambda_{\min}(\boldsymbol{\Lambda'})^2)\|(\boldsymbol{U'}^\top\boldsymbol{\Sigma'}\boldsymbol{U'} + (\hat{\gamma}+\beta)\boldsymbol{I})^{-1}\|_{\mathrm{op}}^2\right) \qquad \text{by \textbf{Concentration on} } \mathcal{V} \text{ and } \|\boldsymbol{q}\| = O(1)$$

$$=\boldsymbol{q}^\top(\boldsymbol{U'}^\top\boldsymbol{\Sigma'}\boldsymbol{U'} + (\hat{\gamma}+\beta)\boldsymbol{I})^{-1}\boldsymbol{U'}^\top\hat{\boldsymbol{\Sigma}}\boldsymbol{U'}(\boldsymbol{U'}^\top\boldsymbol{\Sigma'}\boldsymbol{U'} + (\hat{\gamma}+\beta)\boldsymbol{I})^{-1}\boldsymbol{q} \pm o\left(\frac{\gamma^2 + \delta^2 + \lambda_{\min}(\boldsymbol{\Lambda'})^2}{(\lambda_{\min}(\boldsymbol{U'}^\top\boldsymbol{\Sigma'}\boldsymbol{U'}) + \hat{\gamma}+\beta)^2}\right)$$

$$=\boldsymbol{q}^\top(\boldsymbol{U'}^\top\boldsymbol{\Sigma'}\boldsymbol{U'} + (\hat{\gamma}+\beta)\boldsymbol{I})^{-1}\boldsymbol{U'}^\top\hat{\boldsymbol{\Sigma}}\boldsymbol{U'}(\boldsymbol{U'}^\top\boldsymbol{\Sigma'}\boldsymbol{U'} + (\hat{\gamma}+\beta)\boldsymbol{I})^{-1}\boldsymbol{q} \pm o\left(\frac{\gamma^2 + \delta^2 + \lambda_{\min}(\boldsymbol{\Lambda'})^2}{\hat{\gamma}^2 + \beta^2 + \lambda_{\min}(\boldsymbol{U'}^\top\boldsymbol{\Sigma'}\boldsymbol{U'})^2}\right)$$

$$=\boldsymbol{q}^\top(\boldsymbol{U'}^\top\boldsymbol{\Sigma'}\boldsymbol{U'} + (\hat{\gamma}+\beta)\boldsymbol{I})^{-1}\boldsymbol{U'}^\top\hat{\boldsymbol{\Sigma}}\boldsymbol{U'}(\boldsymbol{U'}^\top\boldsymbol{\Sigma'}\boldsymbol{U'} + (\hat{\gamma}+\beta)\boldsymbol{I})^{-1}\boldsymbol{q} \pm o\left(\frac{\gamma^2 + \delta^2 + \lambda_{\min}(\boldsymbol{\Lambda'})^2}{\hat{\gamma}^2 + \delta^2 + \lambda_{\min}(\boldsymbol{U'}^\top\boldsymbol{\Sigma'}\boldsymbol{U'})^2}\right)$$

$$=\boldsymbol{q}^\top(\boldsymbol{U'}^\top\boldsymbol{\Sigma'}\boldsymbol{U'} + (\hat{\gamma}+\beta)\boldsymbol{I})^{-1}\boldsymbol{U'}^\top\hat{\boldsymbol{\Sigma}}\boldsymbol{U'}(\boldsymbol{U'}^\top\boldsymbol{\Sigma'}\boldsymbol{U'} + (\hat{\gamma}+\beta)\boldsymbol{I})^{-1}\boldsymbol{q} \pm o(1) \qquad \text{by Corollary A.8}$$

$$=\|\frac{1}{\sqrt{\hat{n}}}\hat{\boldsymbol{R}}^\top\boldsymbol{U'}(\frac{1}{\hat{n}}\boldsymbol{U'}^\top\hat{\boldsymbol{R}}\hat{\boldsymbol{R}}^\top\boldsymbol{U'} + (\hat{\gamma}+\beta)\boldsymbol{I})^{-1}\boldsymbol{q}\|^2 \pm o(1)$$

$\qquad\square$

**Corollary A.10.** *If conditions **Boundedness** and **Concentration on** $\mathcal{V}$ hold, then for any $\boldsymbol{\psi}$ with $\|\boldsymbol{\psi}\| = O(1)$, we have* $\|\boldsymbol{U'}\sqrt{\boldsymbol{\Lambda'}}\boldsymbol{U'}^\top\frac{1}{\sqrt{\hat{n}}}\hat{\boldsymbol{R}}(\frac{1}{\hat{n}}\hat{\boldsymbol{R}}^\top\boldsymbol{U'}\boldsymbol{U'}^\top\hat{\boldsymbol{R}} + (\hat{\gamma}+\beta)\boldsymbol{I})^{-1}\boldsymbol{\psi}\|^2 = \|\frac{1}{\hat{n}}\hat{\boldsymbol{R}}^\top\boldsymbol{U'}\boldsymbol{U'}^\top\hat{\boldsymbol{R}}(\frac{1}{\hat{n}}\hat{\boldsymbol{R}}^\top\boldsymbol{U'}\boldsymbol{U'}^\top\hat{\boldsymbol{R}} + (\hat{\gamma}+\beta)\boldsymbol{I})^{-1}\boldsymbol{\psi}\|^2 \pm o(1)$, *and* $\|\boldsymbol{U'}\sqrt{\boldsymbol{\Lambda'}}\boldsymbol{U'}^\top\frac{1}{\sqrt{\hat{n}}}\hat{\boldsymbol{R}}(\frac{1}{\hat{n}}\hat{\boldsymbol{R}}^\top\boldsymbol{U'}\boldsymbol{U'}^\top\hat{\boldsymbol{R}} + (\hat{\gamma}+\beta)\boldsymbol{I})^{-1}\boldsymbol{\psi}\| = O(1)$. *It still holds if we replace* $\hat{}$ *with* $\tilde{}$.

*Proof.* First, we have

$$\|\boldsymbol{U'}\sqrt{\boldsymbol{\Lambda'}}\boldsymbol{U'}^\top\frac{1}{\sqrt{\hat{n}}}\hat{\boldsymbol{R}}(\frac{1}{\hat{n}}\hat{\boldsymbol{R}}^\top\boldsymbol{U'}\boldsymbol{U'}^\top\hat{\boldsymbol{R}} + (\hat{\gamma}+\beta)\boldsymbol{I})^{-1}\boldsymbol{\psi}\|^2$$

$$=\|\boldsymbol{U'}\sqrt{\boldsymbol{\Lambda'}}(\frac{1}{\hat{n}}\boldsymbol{U'}^\top\hat{\boldsymbol{R}}\hat{\boldsymbol{R}}^\top\boldsymbol{U'} + (\hat{\gamma}+\beta)\boldsymbol{I})^{-1}\boldsymbol{U'}^\top\frac{1}{\sqrt{\hat{n}}}\hat{\boldsymbol{R}}\boldsymbol{\psi}\|^2 \qquad \text{by Lemma A.2}$$

$$=\|\frac{1}{\sqrt{\hat{n}}}\hat{\boldsymbol{R}}^\top\boldsymbol{U'}(\frac{1}{\hat{n}}\boldsymbol{U'}^\top\hat{\boldsymbol{R}}\hat{\boldsymbol{R}}^\top\boldsymbol{U'} + (\hat{\gamma}+\beta)\boldsymbol{I})^{-1}\boldsymbol{U'}^\top\frac{1}{\sqrt{\hat{n}}}\hat{\boldsymbol{R}}\boldsymbol{\psi}\|^2 \pm o(1)$$

$$\qquad \text{by the fact that } \|\boldsymbol{U'}^\top\frac{1}{\sqrt{\hat{n}}}\hat{\boldsymbol{R}}\boldsymbol{\psi}\| = O(1) \text{ (via \textbf{Boundedness}) and invoking Corollary A.9}$$

$$=\|\frac{1}{\hat{n}}\hat{\boldsymbol{R}}^\top\boldsymbol{U'}\boldsymbol{U'}^\top\hat{\boldsymbol{R}}(\frac{1}{\hat{n}}\hat{\boldsymbol{R}}^\top\boldsymbol{U'}\boldsymbol{U'}^\top\hat{\boldsymbol{R}} + (\hat{\gamma}+\beta)\boldsymbol{I})^{-1}\boldsymbol{\psi}\|^2 \pm o(1) \qquad \text{by Lemma A.2.}$$

Additionally, since $\|\frac{1}{\hat{n}}\hat{\boldsymbol{R}}^\top\boldsymbol{U'}\boldsymbol{U'}^\top\hat{\boldsymbol{R}}(\frac{1}{\hat{n}}\hat{\boldsymbol{R}}^\top\boldsymbol{U'}\boldsymbol{U'}^\top\hat{\boldsymbol{R}} + (\hat{\gamma}+\beta)\boldsymbol{I})^{-1}\|_{\mathrm{op}} = \frac{\|\frac{1}{\hat{n}}\hat{\boldsymbol{R}}^\top\boldsymbol{U'}\boldsymbol{U'}^\top\hat{\boldsymbol{R}}\|_{\mathrm{op}}}{\|\frac{1}{\hat{n}}\hat{\boldsymbol{R}}^\top\boldsymbol{U'}\boldsymbol{U'}^\top\hat{\boldsymbol{R}}\|_{\mathrm{op}}+\hat{\gamma}+\beta} \leq 1$, we also have the bound $\|\boldsymbol{U'}\sqrt{\boldsymbol{\Lambda'}}\boldsymbol{U'}^\top\frac{1}{\sqrt{\hat{n}}}\hat{\boldsymbol{R}}(\frac{1}{\hat{n}}\hat{\boldsymbol{R}}^\top\boldsymbol{U'}\boldsymbol{U'}^\top\hat{\boldsymbol{R}} + (\hat{\gamma}+\beta)\boldsymbol{I})^{-1}\boldsymbol{\psi}\| = O(1)$. $\qquad\square$

**Lemma A.11.** *If condition **Kernel-wise** $\delta$-**isotropy on** $\mathcal{V}^\perp$ holds, then* $\|\boldsymbol{U}''^\top \frac{1}{\sqrt{\hat{n}}}\hat{\boldsymbol{R}}\|_{\mathrm{op}} \le \sqrt{o(\gamma^2 + \delta^2) + \hat{\gamma}}$. *Similarly,* $\|\boldsymbol{U}''^\top \frac{1}{\sqrt{\tilde{n}}}\tilde{\boldsymbol{R}}\|_{\mathrm{op}} \le \sqrt{o(\gamma^2 + \delta^2) + \tilde{\gamma}}$.

*Proof.* By condition **Kernel-wise** $\delta$-**isotropy on** $\mathcal{V}^\perp$ and triangle inequality, we have

$$\|\frac{1}{\hat{n}}\hat{\boldsymbol{R}}^\top \boldsymbol{U}''\boldsymbol{U}''^\top \hat{\boldsymbol{R}}\|_{\mathrm{op}} \le o(\gamma^2 + \delta^2) + \hat{\gamma}$$

Then,

$$\|\boldsymbol{U}''^\top \frac{1}{\sqrt{\hat{n}}}\hat{\boldsymbol{R}}\|_{\mathrm{op}} = \sqrt{\|\frac{1}{\hat{n}}\hat{\boldsymbol{R}}^\top \boldsymbol{U}''\boldsymbol{U}''^\top \hat{\boldsymbol{R}}\|_{\mathrm{op}}} \le \sqrt{o(\gamma^2 + \delta^2) + \hat{\gamma}}.$$

$\square$

### A.4. Basic expressions for the model weights and errors

Let $\boldsymbol{w}_{\mathrm{w}} \in \mathbb{R}^{d_{\mathrm{w}}}$, $\boldsymbol{w}_{\mathrm{w2s}} \in \mathbb{R}^{d_{\mathrm{s}}}$, and $\boldsymbol{w}_{\mathrm{s}} \in \mathbb{R}^{d_{\mathrm{s}}}$ represent the weights of the linear models $f_{\mathrm{w}}$, $f_{\mathrm{w2s}}$, and $f_{\mathrm{s}}$, respectively. Using the well-known closed-form solution for the minimizer of the MSE loss with $\ell_2$ regularization, we derive their formulas:

$$\begin{aligned}
\boldsymbol{w}_{\mathrm{w}} &= \frac{1}{\sqrt{\tilde{n}}}\tilde{\boldsymbol{R}}_{\mathrm{w}}(\frac{1}{\tilde{n}}\tilde{\boldsymbol{R}}_{\mathrm{w}}^\top \tilde{\boldsymbol{R}}_{\mathrm{w}} + \beta_{\mathrm{w}}\boldsymbol{I})^{-1}\frac{1}{\sqrt{\tilde{n}}}\tilde{\boldsymbol{y}} \\
\boldsymbol{w}_{\mathrm{w2s}} &= \frac{1}{\sqrt{\hat{n}}}\hat{\boldsymbol{R}}_{\mathrm{s}}(\frac{1}{\hat{n}}\hat{\boldsymbol{R}}_{\mathrm{s}}^\top \hat{\boldsymbol{R}}_{\mathrm{s}} + \beta_{\mathrm{s}}\boldsymbol{I})^{-1}\frac{1}{\sqrt{\hat{n}}}(\hat{\boldsymbol{R}}_{\mathrm{w}}^\top \boldsymbol{w}_{\mathrm{w}}) \\
\boldsymbol{w}_{\mathrm{s}} &= \frac{1}{\sqrt{\hat{n}}}\hat{\boldsymbol{R}}_{\mathrm{s}}(\frac{1}{\hat{n}}\hat{\boldsymbol{R}}_{\mathrm{s}}^\top \hat{\boldsymbol{R}}_{\mathrm{s}} + \beta_{\mathrm{s}}\boldsymbol{I})^{-1}\frac{1}{\sqrt{\hat{n}}}\hat{\boldsymbol{y}}.
\end{aligned} \tag{3}$$

Then, we derive the expression of **PredGap**

$$\begin{aligned}
\textbf{PredGap} &= \mathbb{E}_{\boldsymbol{r}_{\mathrm{s}}}[(\boldsymbol{r}_{\mathrm{s}}^\top \boldsymbol{w}_{\mathrm{s}} - \boldsymbol{r}_{\mathrm{s}}^\top \boldsymbol{w}_{\mathrm{w2s}})^2] \\
&= \mathbb{E}_{\boldsymbol{r}_{\mathrm{s}}}[(\boldsymbol{r}_{\mathrm{s}}^\top (\boldsymbol{w}_{\mathrm{s}} - \boldsymbol{w}_{\mathrm{w2s}}))^2] \\
&= \mathbb{E}_{\boldsymbol{r}_{\mathrm{s}}}[(\boldsymbol{w}_{\mathrm{s}} - \boldsymbol{w}_{\mathrm{w2s}})^\top \boldsymbol{r}_{\mathrm{s}}\boldsymbol{r}_{\mathrm{s}}^\top (\boldsymbol{w}_{\mathrm{s}} - \boldsymbol{w}_{\mathrm{w2s}})] \\
&= (\boldsymbol{w}_{\mathrm{s}} - \boldsymbol{w}_{\mathrm{w2s}})^\top \mathbb{E}_{\boldsymbol{r}_{\mathrm{s}}}[\boldsymbol{r}_{\mathrm{s}}\boldsymbol{r}_{\mathrm{s}}^\top](\boldsymbol{w}_{\mathrm{s}} - \boldsymbol{w}_{\mathrm{w2s}}) \\
&= (\boldsymbol{w}_{\mathrm{s}} - \boldsymbol{w}_{\mathrm{w2s}})^\top \boldsymbol{\Sigma}_{\mathrm{s}}(\boldsymbol{w}_{\mathrm{s}} - \boldsymbol{w}_{\mathrm{w2s}}) \\
&= \|\sqrt{\boldsymbol{\Sigma}_{\mathrm{s}}}(\boldsymbol{w}_{\mathrm{s}} - \boldsymbol{w}_{\mathrm{w2s}})\|^2 \\
&= \| \underbrace{\sqrt{\boldsymbol{\Sigma}_{\mathrm{s}}}\frac{1}{\sqrt{\hat{n}}}\hat{\boldsymbol{R}}_{\mathrm{s}}(\frac{1}{\hat{n}}\hat{\boldsymbol{R}}_{\mathrm{s}}^\top \hat{\boldsymbol{R}}_{\mathrm{s}} + \beta_{\mathrm{s}}\boldsymbol{I})^{-1}}_{\text{a transformation determined by the strong model's representations}} \underbrace{\left(\frac{1}{\sqrt{\hat{n}}}\hat{\boldsymbol{y}} - \frac{1}{\sqrt{\hat{n}}}\hat{\boldsymbol{R}}_{\mathrm{w}}^\top \boldsymbol{w}_{\mathrm{w}}\right)}_{\text{weak model's normalized error vector on } \hat{\mathcal{D}}} \|
\end{aligned} \tag{4}$$

$$= \| \underbrace{\sqrt{\boldsymbol{\Sigma}_{\mathrm{s}}}\frac{1}{\sqrt{\hat{n}}}\hat{\boldsymbol{R}}_{\mathrm{s}}(\frac{1}{\hat{n}}\hat{\boldsymbol{R}}_{\mathrm{s}}^\top \hat{\boldsymbol{R}}_{\mathrm{s}} + \beta_{\mathrm{s}}\boldsymbol{I})^{-1}}_{\text{a transformation determined by the strong model's representations}} \underbrace{\left(\frac{1}{\sqrt{\hat{n}}}\hat{\boldsymbol{y}} - \frac{1}{\sqrt{\hat{n}}}\hat{\boldsymbol{R}}_{\mathrm{w}}^\top \frac{1}{\sqrt{\tilde{n}}}\tilde{\boldsymbol{R}}_{\mathrm{w}}(\frac{1}{\tilde{n}}\tilde{\boldsymbol{R}}_{\mathrm{w}}^\top \tilde{\boldsymbol{R}}_{\mathrm{w}} + \beta_{\mathrm{w}}\boldsymbol{I})^{-1}\frac{1}{\sqrt{\tilde{n}}}\tilde{\boldsymbol{y}}\right)}_{\text{weak model's normalized error vector on } \hat{\mathcal{D}}} \|.$$

$$(5)$$

From the above, we see that **PredGap** can be broken into two parts: the weak model's normalized error vector on $\hat{\mathcal{D}}$, and a transformation applied to this error vector which captures how the weak model's errors propagate to the strong model. In Sections A.5 and A.6, we will analyze each part individually.

### A.5. The weak model's error

**Lemma A.12** (The weak model's error on $\hat{\mathcal{D}}$ ). *The weak model's error vector on $\hat{\mathcal{D}}$ can be approximated as follows*

$$\|\left(\frac{1}{\sqrt{\hat{n}}}\hat{\boldsymbol{y}} - \frac{1}{\sqrt{\hat{n}}}\hat{\boldsymbol{R}}_{\mathrm{w}}^\top \frac{1}{\sqrt{\tilde{n}}}\tilde{\boldsymbol{R}}_{\mathrm{w}}(\frac{1}{\tilde{n}}\tilde{\boldsymbol{R}}_{\mathrm{w}}^\top \tilde{\boldsymbol{R}}_{\mathrm{w}} + \beta_{\mathrm{w}}\boldsymbol{I})^{-1}\frac{1}{\sqrt{\tilde{n}}}\tilde{\boldsymbol{y}}\right) - (\boldsymbol{I} - \boldsymbol{P}_{\mathrm{w}})\frac{1}{\sqrt{\hat{n}}}\hat{\boldsymbol{y}}\| = o(1),$$

*where* $\boldsymbol{P}_w = \frac{1}{\hat{n}}\hat{\boldsymbol{R}}_w^\top \boldsymbol{U}_w' \boldsymbol{U}_w'^\top \hat{\boldsymbol{R}}_w \left(\frac{1}{\hat{n}}\hat{\boldsymbol{R}}_w^\top \boldsymbol{U}_w' \boldsymbol{U}_w'^\top \hat{\boldsymbol{R}}_w + (\tilde{\gamma}_w + \beta_w)\boldsymbol{I}\right)^{-1}$.

*Proof.* By condition **Boundedness** and Lemma A.5, we have

$$\frac{1}{\sqrt{\hat{n}}}\hat{\boldsymbol{R}}_{\mathrm{w}}^{\top}\frac{1}{\sqrt{\tilde{n}}}\tilde{\boldsymbol{R}}_{\mathrm{w}}(\frac{1}{\tilde{n}}\tilde{\boldsymbol{R}}_{\mathrm{w}}^{\top}\tilde{\boldsymbol{R}}_{\mathrm{w}}+\beta_{\mathrm{w}}\boldsymbol{I})^{-1}\frac{1}{\sqrt{\tilde{n}}}\tilde{\boldsymbol{y}}$$

$$=\frac{1}{\sqrt{\hat{n}}}\hat{\boldsymbol{R}}_{\mathrm{w}}^{\top}\frac{1}{\sqrt{\tilde{n}}}\tilde{\boldsymbol{R}}_{\mathrm{w}}\left(\frac{1}{\tilde{n}}\tilde{\boldsymbol{R}}_{\mathrm{w}}^{\top}\boldsymbol{U}_{\mathrm{w}}'\boldsymbol{U}_{\mathrm{w}}'^{\top}\tilde{\boldsymbol{R}}_{\mathrm{w}}+(\tilde{\gamma}_{\mathrm{w}}+\beta_{\mathrm{w}})\boldsymbol{I}\right)^{-1}\frac{1}{\sqrt{\tilde{n}}}\tilde{\boldsymbol{y}}+o(1)$$

$$=\left(\frac{1}{\sqrt{\hat{n}}}\hat{\boldsymbol{R}}_{\mathrm{w}}^{\top}\boldsymbol{U}_{\mathrm{w}}'\boldsymbol{U}_{\mathrm{w}}'^{\top}\frac{1}{\sqrt{\tilde{n}}}\tilde{\boldsymbol{R}}_{\mathrm{w}}+\frac{1}{\sqrt{\hat{n}}}\hat{\boldsymbol{R}}_{\mathrm{w}}^{\top}\boldsymbol{U}_{\mathrm{w}}''\boldsymbol{U}_{\mathrm{w}}''^{\top}\frac{1}{\sqrt{\tilde{n}}}\tilde{\boldsymbol{R}}_{\mathrm{w}}\right)\left(\frac{1}{\tilde{n}}\tilde{\boldsymbol{R}}_{\mathrm{w}}^{\top}\boldsymbol{U}_{\mathrm{w}}'\boldsymbol{U}_{\mathrm{w}}'^{\top}\tilde{\boldsymbol{R}}_{\mathrm{w}}+(\tilde{\gamma}_{\mathrm{w}}+\beta_{\mathrm{w}})\boldsymbol{I}\right)^{-1}\frac{1}{\sqrt{\tilde{n}}}\tilde{\boldsymbol{y}}+o(1)$$

By conditions **Small cross-sample inner-product on** $\mathcal{V}^{\perp}$ and **Boundedness**, and noting that $\|(\frac{1}{\tilde{n}}\tilde{\boldsymbol{R}}_{\mathrm{w}}^{\top}\boldsymbol{U}_{\mathrm{w}}'\boldsymbol{U}_{\mathrm{w}}'^{\top}\tilde{\boldsymbol{R}}_{\mathrm{w}}+(\tilde{\gamma}_{\mathrm{w}}+\beta_{\mathrm{w}})\boldsymbol{I})^{-1}\|_{\mathrm{op}}\leq\frac{1}{\tilde{\gamma}_{\mathrm{w}}+\beta_{w}}$, the preceding can be further bounded as

$$\frac{1}{\sqrt{\hat{n}}}\hat{\boldsymbol{R}}_{\mathrm{w}}^{\top}\frac{1}{\sqrt{\tilde{n}}}\tilde{\boldsymbol{R}}_{\mathrm{w}}(\frac{1}{\tilde{n}}\tilde{\boldsymbol{R}}_{\mathrm{w}}^{\top}\tilde{\boldsymbol{R}}_{\mathrm{w}}+\beta_{\mathrm{w}}\boldsymbol{I})^{-1}\frac{1}{\sqrt{\tilde{n}}}\tilde{\boldsymbol{y}}$$

$$=\frac{1}{\sqrt{\hat{n}}}\hat{\boldsymbol{R}}_{\mathrm{w}}^{\top}\boldsymbol{U}_{\mathrm{w}}'\boldsymbol{U}_{\mathrm{w}}'^{\top}\frac{1}{\sqrt{\tilde{n}}}\tilde{\boldsymbol{R}}_{\mathrm{w}}\left(\frac{1}{\tilde{n}}\tilde{\boldsymbol{R}}_{\mathrm{w}}^{\top}\boldsymbol{U}_{\mathrm{w}}'\boldsymbol{U}_{\mathrm{w}}'^{\top}\tilde{\boldsymbol{R}}_{\mathrm{w}}+(\tilde{\gamma}_{\mathrm{w}}+\beta_{\mathrm{w}})\boldsymbol{I}\right)^{-1}\frac{1}{\sqrt{\tilde{n}}}\tilde{\boldsymbol{y}}+o(1)$$

$$=\frac{1}{\sqrt{\hat{n}}}\hat{\boldsymbol{R}}_{\mathrm{w}}^{\top}\boldsymbol{U}_{\mathrm{w}}'\left(\frac{1}{\tilde{n}}\boldsymbol{U}_{\mathrm{w}}'^{\top}\tilde{\boldsymbol{R}}_{\mathrm{w}}\tilde{\boldsymbol{R}}_{\mathrm{w}}^{\top}\boldsymbol{U}_{\mathrm{w}}'+(\tilde{\gamma}_{\mathrm{w}}+\beta_{\mathrm{w}})\boldsymbol{I}\right)^{-1}\boldsymbol{U}_{\mathrm{w}}'^{\top}\frac{1}{\tilde{n}}\tilde{\boldsymbol{R}}_{\mathrm{w}}\tilde{\boldsymbol{y}}+o(1)\qquad\text{by Lemma A.2.}$$

By Lemma A.6 and condition **Boundedness**, the above further leads to

$$\frac{1}{\sqrt{\hat{n}}}\hat{\boldsymbol{R}}_{\mathrm{w}}^{\top}\frac{1}{\sqrt{\tilde{n}}}\tilde{\boldsymbol{R}}_{\mathrm{w}}(\frac{1}{\tilde{n}}\tilde{\boldsymbol{R}}_{\mathrm{w}}^{\top}\tilde{\boldsymbol{R}}_{\mathrm{w}}+\beta_{\mathrm{w}}\boldsymbol{I})^{-1}\frac{1}{\sqrt{\tilde{n}}}\tilde{\boldsymbol{y}}$$

$$=\frac{1}{\sqrt{\hat{n}}}\hat{\boldsymbol{R}}_{\mathrm{w}}^{\top}\boldsymbol{U}_{\mathrm{w}}'\left(\boldsymbol{\Lambda}_{\mathrm{w}}'+(\tilde{\gamma}_{\mathrm{w}}+\beta_{\mathrm{w}})\boldsymbol{I}\right)^{-1}\boldsymbol{U}_{\mathrm{w}}'^{\top}\frac{1}{\tilde{n}}\tilde{\boldsymbol{R}}_{\mathrm{w}}\tilde{\boldsymbol{y}}+o(1).$$

Condition **Concentration on** $\mathcal{V}$ implies that $\|\boldsymbol{U}_{\mathrm{w}}'^{\top}\frac{1}{\hat{n}}\hat{\boldsymbol{R}}_{\mathrm{w}}\hat{\boldsymbol{y}}-\boldsymbol{U}_{\mathrm{w}}'^{\top}\frac{1}{\tilde{n}}\hat{\boldsymbol{R}}_{\mathrm{w}}\hat{\boldsymbol{y}}\|_{\mathrm{op}}=o(\lambda_{\min}(\boldsymbol{\Lambda}_{\mathrm{w}}')+\gamma_{\mathrm{w}}+\beta_{\mathrm{w}})$ via the triangle inequality. Then, by condition **Boundedness** and that $\|(\boldsymbol{\Lambda}_{\mathrm{w}}'+(\tilde{\gamma}_{\mathrm{w}}+\beta_{\mathrm{w}})\boldsymbol{I})^{-1}\|_{\mathrm{op}}=\frac{1}{\lambda_{\min}(\boldsymbol{\Lambda}_{\mathrm{w}}')+\tilde{\gamma}_{\mathrm{w}}+\beta_{\mathrm{w}}}$, we further have

$$\frac{1}{\sqrt{\hat{n}}}\hat{\boldsymbol{R}}_{\mathrm{w}}^{\top}\frac{1}{\sqrt{\tilde{n}}}\tilde{\boldsymbol{R}}_{\mathrm{w}}(\frac{1}{\tilde{n}}\tilde{\boldsymbol{R}}_{\mathrm{w}}^{\top}\tilde{\boldsymbol{R}}_{\mathrm{w}}+\beta_{\mathrm{w}}\boldsymbol{I})^{-1}\frac{1}{\sqrt{\tilde{n}}}\tilde{\boldsymbol{y}}$$

$$=\frac{1}{\sqrt{\hat{n}}}\hat{\boldsymbol{R}}_{\mathrm{w}}^{\top}\boldsymbol{U}_{\mathrm{w}}'\left(\boldsymbol{\Lambda}_{\mathrm{w}}'+(\tilde{\gamma}_{\mathrm{w}}+\beta_{\mathrm{w}})\boldsymbol{I}\right)^{-1}\boldsymbol{U}_{\mathrm{w}}'^{\top}\frac{1}{\hat{n}}\hat{\boldsymbol{R}}_{\mathrm{w}}\hat{\boldsymbol{y}}+o(1)$$

$$=\frac{1}{\sqrt{\hat{n}}}\hat{\boldsymbol{R}}_{\mathrm{w}}^{\top}\boldsymbol{U}_{\mathrm{w}}'\left(\frac{1}{\hat{n}}\boldsymbol{U}_{\mathrm{w}}'^{\top}\hat{\boldsymbol{R}}_{\mathrm{w}}\hat{\boldsymbol{R}}_{\mathrm{w}}^{\top}\boldsymbol{U}_{\mathrm{w}}'+(\tilde{\gamma}_{\mathrm{w}}+\beta_{\mathrm{w}})\boldsymbol{I}\right)^{-1}\boldsymbol{U}_{\mathrm{w}}'^{\top}\frac{1}{\hat{n}}\hat{\boldsymbol{R}}_{\mathrm{w}}\hat{\boldsymbol{y}}+o(1)\qquad\text{by Lemma A.6 and condition }\textbf{Boundedness}$$

$$=\frac{1}{\hat{n}}\hat{\boldsymbol{R}}_{\mathrm{w}}^{\top}\boldsymbol{U}_{\mathrm{w}}'\boldsymbol{U}_{\mathrm{w}}'^{\top}\hat{\boldsymbol{R}}_{\mathrm{w}}\left(\frac{1}{\hat{n}}\hat{\boldsymbol{R}}_{\mathrm{w}}^{\top}\boldsymbol{U}_{\mathrm{w}}'\boldsymbol{U}_{\mathrm{w}}'^{\top}\hat{\boldsymbol{R}}_{\mathrm{w}}+(\tilde{\gamma}_{\mathrm{w}}+\beta_{\mathrm{w}})\boldsymbol{I}\right)^{-1}\frac{1}{\sqrt{\hat{n}}}\hat{\boldsymbol{y}}+o(1)\qquad\text{by Lemma A.2.}$$

Let us define the shorthand $\boldsymbol{P}_{\mathrm{w}}=\frac{1}{\hat{n}}\hat{\boldsymbol{R}}_{\mathrm{w}}^{\top}\boldsymbol{U}_{\mathrm{w}}'\boldsymbol{U}_{\mathrm{w}}'^{\top}\hat{\boldsymbol{R}}_{\mathrm{w}}\left(\frac{1}{\hat{n}}\hat{\boldsymbol{R}}_{\mathrm{w}}^{\top}\boldsymbol{U}_{\mathrm{w}}'\boldsymbol{U}_{\mathrm{w}}'^{\top}\hat{\boldsymbol{R}}_{\mathrm{w}}+(\tilde{\gamma}_{\mathrm{w}}+\beta_{\mathrm{w}})\boldsymbol{I}\right)^{-1}$. Then, we conclude that

$$\frac{1}{\sqrt{\hat{n}}}\hat{\boldsymbol{y}}-\frac{1}{\sqrt{\hat{n}}}\hat{\boldsymbol{R}}_{\mathrm{w}}^{\top}\frac{1}{\sqrt{\tilde{n}}}\tilde{\boldsymbol{R}}_{\mathrm{w}}(\frac{1}{\tilde{n}}\tilde{\boldsymbol{R}}_{\mathrm{w}}^{\top}\tilde{\boldsymbol{R}}_{\mathrm{w}}+\beta_{\mathrm{w}}\boldsymbol{I})^{-1}\frac{1}{\sqrt{\tilde{n}}}\tilde{\boldsymbol{y}}=(\boldsymbol{I}-\boldsymbol{P}_{\mathrm{w}})\frac{1}{\sqrt{\hat{n}}}\hat{\boldsymbol{y}}+o(1).$$

$\square$

## A.6. Propagation of the error to the strong model

**Lemma A.13.** *For any $\psi$ with $\|\psi\|=O(1)$, we have $\|\sqrt{\boldsymbol{\Sigma}_s}\frac{1}{\sqrt{\hat{n}}}\hat{\boldsymbol{R}}_s(\frac{1}{\hat{n}}\hat{\boldsymbol{R}}_s^{\top}\hat{\boldsymbol{R}}_s+\beta_s\boldsymbol{I})^{-1}\psi\|^2=\|\boldsymbol{P}_s\psi\|^2\pm o(1)$, where*

$$\boldsymbol{P}_s=\frac{1}{\hat{n}}\hat{\boldsymbol{R}}_s^{\top}\boldsymbol{U}_s'\boldsymbol{U}_s'^{\top}\hat{\boldsymbol{R}}_s(\frac{1}{\hat{n}}\hat{\boldsymbol{R}}_s^{\top}\boldsymbol{U}_s'\boldsymbol{U}'^{\top}\hat{\boldsymbol{R}}_s+(\hat{\gamma}_s+\beta_s)\boldsymbol{I})^{-1}.$$

*Proof.* We first decompose $\sqrt{\Sigma_{\mathrm{s}}}\frac{1}{\sqrt{\hat{n}}}\hat{R}_{\mathrm{s}}(\frac{1}{\hat{n}}\hat{R}_{\mathrm{s}}^\top\hat{R}_{\mathrm{s}}+\beta_{\mathrm{s}}I)^{-1}$ as follows

$$\sqrt{\Sigma_{\mathrm{s}}}\frac{1}{\sqrt{\hat{n}}}\hat{R}_{\mathrm{s}}(\frac{1}{\hat{n}}\hat{R}_{\mathrm{s}}^\top\hat{R}_{\mathrm{s}}+\beta_{\mathrm{s}}I)^{-1}$$

$$=\sqrt{\Sigma_{\mathrm{s}}}\frac{1}{\sqrt{\hat{n}}}\hat{R}_{\mathrm{s}}(\frac{1}{\hat{n}}\hat{R}_{\mathrm{s}}^\top U_{\mathrm{s}}'U_{\mathrm{s}}'^\top\hat{R}_{\mathrm{s}}+(\hat{\gamma}_{\mathrm{s}}+\beta_{\mathrm{s}})I)^{-1}+o(1) \qquad \text{by Lemma A.5}$$

$$=(U_{\mathrm{s}}'\sqrt{\Lambda_{\mathrm{s}}'}U_{\mathrm{s}}'^\top+U_{\mathrm{s}}''\sqrt{\Lambda_{\mathrm{s}}''}U_{\mathrm{s}}''^\top)\frac{1}{\sqrt{\hat{n}}}\hat{R}_{\mathrm{s}}(\frac{1}{\hat{n}}\hat{R}_{\mathrm{s}}^\top U_{\mathrm{s}}'U_{\mathrm{s}}'^\top\hat{R}_{\mathrm{s}}+(\hat{\gamma}_{\mathrm{s}}+\beta_{\mathrm{s}})I)^{-1}+o(1)$$

$$=U_{\mathrm{s}}'\sqrt{\Lambda_{\mathrm{s}}'}U_{\mathrm{s}}'^\top\frac{1}{\sqrt{\hat{n}}}\hat{R}_{\mathrm{s}}(\frac{1}{\hat{n}}\hat{R}_{\mathrm{s}}^\top U_{\mathrm{s}}'U_{\mathrm{s}}'^\top\hat{R}_{\mathrm{s}}+(\hat{\gamma}_{\mathrm{s}}+\beta_{\mathrm{s}})I)^{-1}$$

$$+U_{\mathrm{s}}''\sqrt{\Lambda_{\mathrm{s}}''}U_{\mathrm{s}}''^\top\frac{1}{\sqrt{\hat{n}}}\hat{R}_{\mathrm{s}}(\frac{1}{\hat{n}}\hat{R}_{\mathrm{s}}^\top U_{\mathrm{s}}'U_{\mathrm{s}}'^\top\hat{R}_{\mathrm{s}}+(\hat{\gamma}_{\mathrm{s}}+\beta_{\mathrm{s}})I)^{-1}+o(1) \tag{6}$$

The second term above can be bounded:

$$\|U_{\mathrm{s}}''\sqrt{\Lambda_{\mathrm{s}}''}U_{\mathrm{s}}''^\top\frac{1}{\sqrt{\hat{n}}}\hat{R}_{\mathrm{s}}(\frac{1}{\hat{n}}\hat{R}_{\mathrm{s}}^\top U_{\mathrm{s}}'U_{\mathrm{s}}'^\top\hat{R}_{\mathrm{s}}+(\hat{\gamma}_{\mathrm{s}}+\beta_{\mathrm{s}})I)^{-1}\|_{\mathrm{op}}$$

$$\leq\sqrt{\lambda_{\max}(\Lambda_{\mathrm{s}}'')}\frac{\sqrt{o(\gamma_{\mathrm{s}}^2+\delta_{\mathrm{s}}^2)+\hat{\gamma}_{\mathrm{s}}}}{\hat{\gamma}_{\mathrm{s}}+\beta_{\mathrm{s}}} \qquad \text{by \textbf{Boundedness} and Lemma A.11}$$

$$\leq\sqrt{\|\Sigma_{\mathrm{s}}''\|_{\mathrm{op}}}\frac{\sqrt{o(\gamma_{\mathrm{s}}^2+\delta^2)+\hat{\gamma}_{\mathrm{s}}}}{\hat{\gamma}_{\mathrm{s}}+\delta_{\mathrm{s}}}$$

$$=o\left(\sqrt{\frac{(\gamma_{\mathrm{s}}+\delta_{\mathrm{s}})o(\gamma_{\mathrm{s}}^2+\delta_{\mathrm{s}}^2)+\hat{\gamma}_{\mathrm{s}}(\gamma_{\mathrm{s}}+\delta_{\mathrm{s}})}{(\hat{\gamma}_{\mathrm{s}}+\delta_{\mathrm{s}})^2}}\right) \qquad \text{by \textbf{Diminishing population covariance on} } \mathcal{V}^\perp$$

$$\leq o\left(\sqrt{\frac{o(\gamma_{\mathrm{s}}^2+\delta_{\mathrm{s}}^2)}{\hat{\gamma}_{\mathrm{s}}+\delta_{\mathrm{s}}}+\frac{\hat{\gamma}_{\mathrm{s}}}{\hat{\gamma}_{\mathrm{s}}+\delta_{\mathrm{s}}}}\right)=o(1). \tag{7}$$

Combining Equations 6 and 7 yields

$$\sqrt{\Sigma_{\mathrm{s}}}\frac{1}{\sqrt{\hat{n}}}\hat{R}_{\mathrm{s}}(\frac{1}{\hat{n}}\hat{R}_{\mathrm{s}}^\top\hat{R}_{\mathrm{s}}+\beta_{\mathrm{s}}I)^{-1}\psi=U_{\mathrm{s}}'\sqrt{\Lambda_{\mathrm{s}}'}U_{\mathrm{s}}'^\top\frac{1}{\sqrt{\hat{n}}}\hat{R}_{\mathrm{s}}(\frac{1}{\hat{n}}\hat{R}_{\mathrm{s}}^\top U_{\mathrm{s}}'U_{\mathrm{s}}'^\top\hat{R}_{\mathrm{s}}+(\hat{\gamma}_{\mathrm{s}}+\beta_{\mathrm{s}})I)^{-1}\psi+o(1).$$

Finally, we consider the squared norm:

$$\|\sqrt{\Sigma_{\mathrm{s}}}\frac{1}{\sqrt{\hat{n}}}\hat{R}_{\mathrm{s}}(\frac{1}{\hat{n}}\hat{R}_{\mathrm{s}}^\top\hat{R}_{\mathrm{s}}+\beta_{\mathrm{s}}I)^{-1}\psi\|^2$$

$$=\|U_{\mathrm{s}}'\sqrt{\Lambda_{\mathrm{s}}'}U_{\mathrm{s}}'^\top\frac{1}{\sqrt{\hat{n}}}\hat{R}_{\mathrm{s}}(\frac{1}{\hat{n}}\hat{R}_{\mathrm{s}}^\top U_{\mathrm{s}}'U_{\mathrm{s}}'^\top\hat{R}_{\mathrm{s}}+(\hat{\gamma}_{\mathrm{s}}+\beta_{\mathrm{s}})I)^{-1}\psi\|^2$$

$$\pm o\left(\|U_{\mathrm{s}}'\sqrt{\Lambda_{\mathrm{s}}'}U_{\mathrm{s}}'^\top\frac{1}{\sqrt{\hat{n}}}\hat{R}_{\mathrm{s}}(\frac{1}{\hat{n}}\hat{R}_{\mathrm{s}}^\top U_{\mathrm{s}}'U_{\mathrm{s}}'^\top\hat{R}_{\mathrm{s}}+(\hat{\gamma}_{\mathrm{s}}+\beta_{\mathrm{s}})I)^{-1}\psi\|\right)\pm o(1)$$

$$=\|\frac{1}{\hat{n}}\hat{R}_{\mathrm{s}}^\top U_{\mathrm{s}}'U_{\mathrm{s}}'^\top\hat{R}_{\mathrm{s}}(\frac{1}{\hat{n}}\hat{R}_{\mathrm{s}}^\top U_{\mathrm{s}}'U'^\top\hat{R}_{\mathrm{s}}+(\hat{\gamma}_{\mathrm{s}}+\beta_{\mathrm{s}})I)^{-1}\psi\|^2\pm o(1) \quad \text{by Corollary A.10.}$$

$\square$

### A.7. Proof of Theorem 3.8

Given that $\|\frac{1}{\sqrt{\hat{n}}}\hat{y}\|=O(1)$ by **Boundedness**, and that $\|I-P_{\mathrm{w}}\|_{\mathrm{op}}=\frac{\beta_{\mathrm{w}}}{\lambda_{\min}(\frac{1}{\hat{n}}\hat{R}_{\mathrm{w}}^\top U_{\mathrm{w}}'U_{\mathrm{w}}'^\top\hat{R}_{\mathrm{w}})+\beta_{\mathrm{w}}}\leq 1$, we have $\|(I-P_{\mathrm{w}})\frac{1}{\sqrt{\hat{n}}}\hat{y}\|=O(1)$. Then, by Lemma A.12, the weak model's error on $\hat{\mathcal{D}}$ can be bounded as $\left\|(I-P_{\mathrm{w}})\frac{1}{\sqrt{\hat{n}}}\hat{y}\right\|+o(1)=O(1)$. Recalling the expression of **PredGap** derived in Equation 5 and applying Lemmas A.12 and A.13, we obtain:

$$\textbf{PredGap}=\|P_{\mathrm{s}}(I-P_{\mathrm{w}})\frac{1}{\sqrt{\hat{n}}}\hat{y}\|^2\pm o(1).$$

# B. Additional Analysis

## B.1. Additional Lemmas

**Lemma B.1.** *By Diminishing population covariance on $\mathcal{V}^{\perp}$ and Boundedness, we have*

$$\mathbb{E}[\boldsymbol{U}''\boldsymbol{U}''^{\top}\boldsymbol{r}y] = o(\sqrt{\gamma + \delta}).$$

*Proof.*

$$\mathbb{E}[\boldsymbol{U}''\boldsymbol{U}''^{\top}\boldsymbol{r}y] = \lim_{n\to\infty}\frac{1}{n}\sum_{i=1}^{n}\boldsymbol{U}''\boldsymbol{U}''^{\top}\boldsymbol{r}_i y_i = \lim_{n\to\infty}\frac{1}{\sqrt{n}}\boldsymbol{U}''\boldsymbol{U}''^{\top}\boldsymbol{R}\frac{1}{\sqrt{n}}\boldsymbol{y} \leq \lim_{n\to\infty}\|\frac{1}{\sqrt{n}}\boldsymbol{U}''\boldsymbol{U}''^{\top}\boldsymbol{R}\|_{\mathrm{op}}\|\frac{1}{\sqrt{n}}\boldsymbol{y}\|$$

$$= \lim_{n\to\infty}\sqrt{\|\frac{1}{n}\boldsymbol{U}''\boldsymbol{U}''^{\top}\boldsymbol{R}\boldsymbol{R}^{\top}\boldsymbol{U}''\boldsymbol{U}''^{\top}\|_{\mathrm{op}}}\sqrt{\frac{1}{n}\sum_{i=1}^{n}y_i^2} = \sqrt{\|\boldsymbol{\Sigma}''\|_{\mathrm{op}}}\sqrt{\mathbb{E}[y^2]} = o(\sqrt{\gamma + \delta}).$$

$\square$

**Lemma B.2.** *By Boundedness, we have*

$$\mathbb{E}[\boldsymbol{U}'\boldsymbol{U}'^{\top}\boldsymbol{r}y] = O(1).$$

*Proof.* The proof follows the same approach as that of Lemma B.1. This conclusion can also be derived by bounding $\mathbb{E}[\boldsymbol{U}'\boldsymbol{U}'^{\top}\boldsymbol{r}y]$ in terms of its empirical counterpart using **Concentration on $\mathcal{V}$**, and then applying **Boundedness** $\square$

## B.2. When $\mathrm{Err}_{\mathrm{w2s}} \approx \mathbf{PredGap} + \mathrm{Err}_{\mathrm{sc}}$

**Theorem B.3.** *Suppose that, in addition to Assumption 3.7, the conditions $\beta_s + \hat{\gamma}_s = o(\lambda_{min,\,\neq\,0}(\boldsymbol{\Sigma}(\boldsymbol{\Pi}_{\mathcal{V}_s}h_s)) = \Theta(1))$ and $\lambda_{min,\,\neq\,0}(\boldsymbol{\Sigma}(\boldsymbol{\Pi}_{\mathcal{V}_s}h_s)) = \Theta(\lambda_{\max}(\boldsymbol{\Sigma}(\boldsymbol{\Pi}_{\mathcal{V}_s}h_s)))$ hold. Then, w.h.p., we have:*

$$Err_{w2s} = \mathbf{PredGap} + Err_{sc} \pm o(1).$$

*Proof.* First, decompose $\mathrm{Err}_{\mathrm{w2s}}$ as follows

$$
\begin{aligned}
\mathrm{Err}_{\mathrm{w2s}} &= \mathbb{E}[(\boldsymbol{r}_{\mathrm{s}}^{\top}\boldsymbol{w}_{\mathrm{w2s}} - y)^2]\\
&= \mathbb{E}[(\boldsymbol{r}_{\mathrm{s}}^{\top}\boldsymbol{w}_{\mathrm{w2s}} - \boldsymbol{r}_{\mathrm{s}}^{\top}\boldsymbol{w}_{\mathrm{sc}} + \boldsymbol{r}_{\mathrm{s}}^{\top}\boldsymbol{w}_{\mathrm{sc}} - y)^2]\\
&= \mathbb{E}[(\boldsymbol{r}_{\mathrm{s}}^{\top}\boldsymbol{w}_{\mathrm{w2s}} - \boldsymbol{r}_{\mathrm{s}}^{\top}\boldsymbol{w}_{\mathrm{sc}})^2 + (\boldsymbol{r}_{\mathrm{s}}^{\top}\boldsymbol{w}_{\mathrm{sc}} - y)^2 + 2(\boldsymbol{w}_{\mathrm{w2s}}^{\top}\boldsymbol{r}_{\mathrm{s}} - \boldsymbol{w}_{\mathrm{sc}}^{\top}\boldsymbol{r}_{\mathrm{s}})(\boldsymbol{r}_{\mathrm{s}}^{\top}\boldsymbol{w}_{\mathrm{sc}} - y)]\\
&= \mathbf{PredGap} + \mathrm{Err}_{\mathrm{sc}} + 2\mathbb{E}[(\boldsymbol{w}_{\mathrm{w2s}}^{\top}\boldsymbol{r}_{\mathrm{s}} - \boldsymbol{w}_{\mathrm{sc}}^{\top}\boldsymbol{r}_{\mathrm{s}})(\boldsymbol{r}_{\mathrm{s}}^{\top}\boldsymbol{w}_{\mathrm{sc}} - y)]\\
&= \mathbf{PredGap} + \mathrm{Err}_{\mathrm{sc}} + 2(\boldsymbol{w}_{\mathrm{w2s}} - \boldsymbol{w}_{\mathrm{sc}})^{\top}(\boldsymbol{\Sigma}_{\mathrm{s}}\boldsymbol{w}_{\mathrm{sc}} - \mathbb{E}[\boldsymbol{r}_{\mathrm{s}}y]),
\end{aligned}
\tag{8}
$$

Thus, to prove the theorem, it suffices to show $|(\boldsymbol{w}_{\mathrm{w2s}} - \boldsymbol{w}_{\mathrm{sc}})^{\top}(\boldsymbol{\Sigma}_{\mathrm{s}}\boldsymbol{w}_{\mathrm{sc}} - \mathbb{E}[\boldsymbol{r}_{\mathrm{s}}y])| = o(1)$. We decompose $(\boldsymbol{w}_{\mathrm{w2s}} - \boldsymbol{w}_{\mathrm{sc}})^{\top}(\boldsymbol{\Sigma}_{\mathrm{s}}\boldsymbol{w}_{\mathrm{sc}} - \mathbb{E}[\boldsymbol{r}_{\mathrm{s}}y])$:

$$
\begin{aligned}
&(\boldsymbol{w}_{\mathrm{w2s}} - \boldsymbol{w}_{\mathrm{sc}})^{\top}(\boldsymbol{\Sigma}_{\mathrm{s}}\boldsymbol{w}_{\mathrm{sc}} - \mathbb{E}[\boldsymbol{r}_{\mathrm{s}}y])\\
=&(\boldsymbol{w}_{\mathrm{w2s}} - \boldsymbol{w}_{\mathrm{sc}})^{\top}(\boldsymbol{\Sigma}_{\mathrm{s}}'\boldsymbol{w}_{\mathrm{sc}} + \boldsymbol{\Sigma}_{\mathrm{s}}''\boldsymbol{w}_{\mathrm{sc}} - \boldsymbol{U}_{\mathrm{s}}'\boldsymbol{U}_{\mathrm{s}}'^{\top}\mathbb{E}[\boldsymbol{r}_{\mathrm{s}}y] - \boldsymbol{U}_{\mathrm{s}}''\boldsymbol{U}_{\mathrm{s}}''^{\top}\mathbb{E}[\boldsymbol{r}_{\mathrm{s}}y])\\
=&(\boldsymbol{w}_{\mathrm{w2s}} - \boldsymbol{w}_{\mathrm{sc}})^{\top}(\boldsymbol{\Sigma}_{\mathrm{s}}'\boldsymbol{w}_{\mathrm{sc}} - \boldsymbol{U}_{\mathrm{s}}'\boldsymbol{U}_{\mathrm{s}}'^{\top}\mathbb{E}[\boldsymbol{r}_{\mathrm{s}}y]) + (\boldsymbol{w}_{\mathrm{w2s}} - \boldsymbol{w}_{\mathrm{sc}})^{\top}\boldsymbol{\Sigma}_{\mathrm{s}}''\boldsymbol{w}_{\mathrm{sc}} - (\boldsymbol{w}_{\mathrm{w2s}} - \boldsymbol{w}_{\mathrm{sc}})^{\top}\boldsymbol{U}_{\mathrm{s}}''\boldsymbol{U}_{\mathrm{s}}''^{\top}\mathbb{E}[\boldsymbol{r}_{\mathrm{s}}y]
\end{aligned}
\tag{9}
$$

$\boldsymbol{w}_{\mathrm{w2s}} - \boldsymbol{w}_{\mathrm{sc}}$ can be approximated as:

$$
\begin{aligned}
\boldsymbol{w}_{\mathrm{w2s}} - \boldsymbol{w}_{\mathrm{sc}} =&\frac{1}{\sqrt{\hat{n}}}\hat{\boldsymbol{R}}_{\mathrm{s}}(\frac{1}{\hat{n}}\hat{\boldsymbol{R}}_{\mathrm{s}}^{\top}\hat{\boldsymbol{R}}_{\mathrm{s}} + \beta_{\mathrm{s}}\boldsymbol{I})^{-1}\frac{1}{\sqrt{\hat{n}}}(\hat{\boldsymbol{R}}_{\mathrm{w}}^{\top}\boldsymbol{w}_{\mathrm{w}}) - \frac{1}{\sqrt{\hat{n}}}\hat{\boldsymbol{R}}_{\mathrm{s}}(\frac{1}{\hat{n}}\hat{\boldsymbol{R}}_{\mathrm{s}}^{\top}\hat{\boldsymbol{R}}_{\mathrm{s}} + \beta_{\mathrm{s}}\boldsymbol{I})^{-1}\frac{1}{\sqrt{\hat{n}}}\hat{\boldsymbol{y}}\\
=&\frac{1}{\sqrt{\hat{n}}}\hat{\boldsymbol{R}}_{\mathrm{s}}(\frac{1}{\hat{n}}\hat{\boldsymbol{R}}_{\mathrm{s}}^{\top}\hat{\boldsymbol{R}}_{\mathrm{s}} + \beta_{\mathrm{s}}\boldsymbol{I})^{-1}(\frac{1}{\sqrt{\hat{n}}}\hat{\boldsymbol{R}}_{\mathrm{w}}^{\top}\boldsymbol{w}_{\mathrm{w}} - \frac{1}{\sqrt{\hat{n}}}\hat{\boldsymbol{y}})\\
=&\frac{1}{\sqrt{\hat{n}}}\hat{\boldsymbol{R}}_{\mathrm{s}}\left(\frac{1}{\hat{n}}\hat{\boldsymbol{K}}_{\mathrm{s}}' + (\hat{\gamma}_{\mathrm{s}} + \beta_{\mathrm{s}})\boldsymbol{I}\right)^{-1}(\frac{1}{\sqrt{\hat{n}}}\hat{\boldsymbol{R}}_{\mathrm{w}}^{\top}\boldsymbol{w}_{\mathrm{w}} - \frac{1}{\sqrt{\hat{n}}}\hat{\boldsymbol{y}}) + o(1) \quad \text{by Lemma A.5 and that other terms are } O(1)
\end{aligned}
\tag{10}
$$

where $\hat{K}'_{\mathrm{s}} = \hat{R}_{\mathrm{s}}^\top U'_{\mathrm{s}} U'^\top_{\mathrm{s}} \hat{R}_{\mathrm{s}}$ is shorthand for $\hat{K}(\Pi_{\mathcal{V}_{\mathrm{s}}} h_{\mathrm{s}})$. Then, by Lemma A.11 and **Boundedness**, we obtain:

$$\|(w_{\mathrm{w2s}} - w_{\mathrm{sc}})^\top U''_{\mathrm{s}}\| = O\Big(\frac{\sqrt{o(\gamma_{\mathrm{s}}^2 + \delta_{\mathrm{s}}^2) + \hat{\gamma}_{\mathrm{s}}}}{\hat{\gamma}_{\mathrm{s}} + \beta_{\mathrm{s}}}\Big). \tag{11}$$

We also have the following bound:

$$
\begin{aligned}
& \|U''^\top_{\mathrm{s}} \frac{1}{\sqrt{\hat{n}}} \hat{R}_{\mathrm{s}}(\frac{1}{\hat{n}} \hat{R}_{\mathrm{s}}^\top \hat{R}_{\mathrm{s}} + \beta_{\mathrm{s}} I)^{-1} \frac{1}{\sqrt{\hat{n}}} \hat{y}\|_{\mathrm{op}} \\
= & \|U''^\top_{\mathrm{s}} \frac{1}{\sqrt{\hat{n}}} \hat{R}_{\mathrm{s}}(\frac{1}{\hat{n}} \hat{R}_{\mathrm{s}}^\top U'_{\mathrm{s}} U'^\top_{\mathrm{s}} \hat{R}_{\mathrm{s}} + (\hat{\gamma}_{\mathrm{s}} + \beta_{\mathrm{s}}) I)^{-1} \frac{1}{\sqrt{\hat{n}}} \hat{y}\|_{\mathrm{op}} + o(\|U''^\top_{\mathrm{s}} \frac{1}{\sqrt{\hat{n}}} \hat{R}_{\mathrm{s}}\|_{\mathrm{op}}) \quad \text{by } \textbf{Boundedness} \text{ and Lemma A.5} \\
= & O\Big(\frac{\sqrt{o(\gamma_{\mathrm{s}}^2 + \delta_{\mathrm{s}}^2) + \hat{\gamma}_{\mathrm{s}}}}{\hat{\gamma}_{\mathrm{s}} + \beta_{\mathrm{s}}}\Big) \quad \text{by Lemma A.11 and } \textbf{Boundedness}
\end{aligned}
\tag{12}
$$

Combining **Diminishing population covariance on** $\mathcal{V}^\perp$ and Equations 11 and 12, the second term in Equation 9 can be bounded as:

$$
\begin{aligned}
|(w_{\mathrm{w2s}} - w_{\mathrm{sc}})^\top \Sigma''_{\mathrm{s}} w_{\mathrm{sc}}| = & |(w_{\mathrm{w2s}} - w_{\mathrm{sc}})^\top U''_{\mathrm{s}} \Lambda''_{\mathrm{s}} U''^\top_{\mathrm{s}} w_{\mathrm{sc}}| \\
= & o\left(\frac{(o(\gamma_{\mathrm{s}}^2 + \delta_{\mathrm{s}}^2) + \hat{\gamma}_{\mathrm{s}})(\gamma_{\mathrm{s}} + \delta_{\mathrm{s}})}{(\hat{\gamma}_{\mathrm{s}} + \beta_{\mathrm{s}})^2}\right) = o(1).
\end{aligned}
\tag{13}
$$

The third term in Equation 9 can be bounded as:

$$
\begin{aligned}
|(w_{\mathrm{w2s}} - w_{\mathrm{sc}})^\top U''_{\mathrm{s}} U''^\top_{\mathrm{s}} \mathbb{E}[r_{\mathrm{s}} y]| \leq & \|(w_{\mathrm{w2s}} - w_{\mathrm{sc}})^\top U''_{\mathrm{s}}\| \|U''^\top_{\mathrm{s}} \mathbb{E}[r_{\mathrm{s}} y]\| \\
= & O\Big(\frac{\sqrt{o(\gamma_{\mathrm{s}}^2 + \delta_{\mathrm{s}}^2) + \hat{\gamma}_{\mathrm{s}}}}{\hat{\gamma}_{\mathrm{s}} + \beta_{\mathrm{s}}}\Big) o(\sqrt{\gamma_{\mathrm{s}} + \delta_{\mathrm{s}}}) \quad \text{by Equation 11 and Lemma B.1} \\
= & o(1).
\end{aligned}
\tag{14}
$$

Now, it remains to bound the first term in Equation 9. We start with approximating $\Sigma'_{\mathrm{s}} w_{\mathrm{sc}} - U'_{\mathrm{s}} U'^\top_{\mathrm{s}} \mathbb{E}[r_{\mathrm{s}} y]$:

$$
\begin{aligned}
& \Sigma'_{\mathrm{s}} w_{\mathrm{sc}} - U'_{\mathrm{s}} U'^\top_{\mathrm{s}} \mathbb{E}[r_{\mathrm{s}} y] \\
= & U'_{\mathrm{s}} \Lambda'_{\mathrm{s}} U'^\top_{\mathrm{s}} \frac{1}{\sqrt{\hat{n}}} \hat{R}_{\mathrm{s}}(\frac{1}{\hat{n}} \hat{R}_{\mathrm{s}}^\top \hat{R}_{\mathrm{s}} + \beta_{\mathrm{s}} I)^{-1} \frac{1}{\sqrt{\hat{n}}} \hat{y} - U'_{\mathrm{s}} U'^\top_{\mathrm{s}} \mathbb{E}[r_{\mathrm{s}} y] \\
= & U'_{\mathrm{s}} \Lambda'_{\mathrm{s}} U'^\top_{\mathrm{s}} \frac{1}{\sqrt{\hat{n}}} \hat{R}_{\mathrm{s}} \left(\frac{1}{\hat{n}} \hat{R}_{\mathrm{s}}^\top U'_{\mathrm{s}} U'^\top_{\mathrm{s}} \hat{R}_{\mathrm{s}} + (\hat{\gamma}_{\mathrm{s}} + \beta_{\mathrm{s}}) I\right)^{-1} \frac{1}{\sqrt{\hat{n}}} \hat{y} - U'_{\mathrm{s}} U'^\top_{\mathrm{s}} \mathbb{E}[r_{\mathrm{s}} y] + o(1) \\
& \quad \text{by Lemma A.5 and } \textbf{Boundedness} \\
= & U_{\mathrm{s}} U'^\top_{\mathrm{s}} \hat{\Sigma} U'_{\mathrm{s}} U'^\top_{\mathrm{s}} \frac{1}{\sqrt{\hat{n}}} \hat{R}_{\mathrm{s}} \left(\frac{1}{\hat{n}} \hat{R}_{\mathrm{s}}^\top U'_{\mathrm{s}} U'^\top_{\mathrm{s}} \hat{R}_{\mathrm{s}} + (\hat{\gamma}_{\mathrm{s}} + \beta_{\mathrm{s}}) I\right)^{-1} \frac{1}{\sqrt{\hat{n}}} \hat{y} - U'_{\mathrm{s}} U'^\top_{\mathrm{s}} \mathbb{E}[r_{\mathrm{s}} y] + o(1) \\
& \quad \text{by } \textbf{Concentration on } \mathcal{V} \text{ and } \textbf{Boundedness} \\
= & U_{\mathrm{s}} U'^\top_{\mathrm{s}} \hat{\Sigma} U'_{\mathrm{s}} U'^\top_{\mathrm{s}} \frac{1}{\sqrt{\hat{n}}} \hat{R}_{\mathrm{s}} \left(\frac{1}{\hat{n}} \hat{R}_{\mathrm{s}}^\top U'_{\mathrm{s}} U'^\top_{\mathrm{s}} \hat{R}_{\mathrm{s}} + (\hat{\gamma}_{\mathrm{s}} + \beta_{\mathrm{s}}) I\right)^{-1} \frac{1}{\sqrt{\hat{n}}} \hat{y} - U_{\mathrm{s}} U'_{\mathrm{s}} \frac{1}{\hat{n}} \hat{R}_{\mathrm{s}} \hat{y} + o(1) \quad \text{by } \textbf{Concentration on } \mathcal{V}.
\end{aligned}
\tag{15}
$$

Due to the two additional assumptions in the statement of the theorem, along with **Concentration on** $\mathcal{V}$ and **Boundedness**,

the RHSs of both equation 10 and equation 15 are $O(1)$. Combining equation 10 and equation 15, we obtain:

$$(\boldsymbol{w}_{\text{w2s}} - \boldsymbol{w}_{\text{sc}})^\top (\boldsymbol{\Sigma}_s' \boldsymbol{w}_{\text{sc}} - \boldsymbol{U}_s \boldsymbol{U}_s'^\top \mathbb{E}[\boldsymbol{r}_s y])$$

$$= (\frac{1}{\sqrt{\hat{n}}} \hat{\boldsymbol{R}}_{\text{w}}^\top \boldsymbol{w}_{\text{w}} - \frac{1}{\sqrt{\hat{n}}} \hat{\boldsymbol{y}})^\top \left( \frac{1}{\hat{n}} \hat{\boldsymbol{K}}_s' + (\hat{\gamma}_s + \beta_s)\boldsymbol{I} \right)^{-1}$$

$$\times \frac{1}{\sqrt{\hat{n}}} \hat{\boldsymbol{R}}_s^\top \left( \boldsymbol{U}_s \boldsymbol{U}_s'^\top \hat{\boldsymbol{\Sigma}} \boldsymbol{U}_s' \boldsymbol{U}_s'^\top \left( \boldsymbol{U}_s \boldsymbol{U}_s'^\top \hat{\boldsymbol{\Sigma}} \boldsymbol{U}_s' \boldsymbol{U}_s'^\top + (\hat{\gamma}_s + \beta_s)\boldsymbol{I} \right)^{-1} - \boldsymbol{U}_s' \boldsymbol{U}_s'^\top \right) \frac{1}{\sqrt{\hat{n}}} \hat{\boldsymbol{R}}_s \frac{1}{\sqrt{\hat{n}}} \hat{\boldsymbol{y}} + o(1)$$

$$= (\frac{1}{\sqrt{\hat{n}}} \hat{\boldsymbol{R}}_{\text{w}}^\top \boldsymbol{w}_{\text{w}} - \frac{1}{\sqrt{\hat{n}}} \hat{\boldsymbol{y}})^\top \left( \frac{1}{\hat{n}} \hat{\boldsymbol{K}}_s' + (\hat{\gamma}_s + \beta_s)\boldsymbol{I} \right)^{-1}$$

$$\times \left( \frac{1}{\hat{n}} \hat{\boldsymbol{K}}_s' \left( \frac{1}{\hat{n}} \hat{\boldsymbol{K}}_s' + (\hat{\gamma}_s + \beta_s)\boldsymbol{I} \right)^{-1} \frac{1}{\hat{n}} \hat{\boldsymbol{K}}_s' - \frac{1}{\hat{n}} \hat{\boldsymbol{K}}_s' \right) \frac{1}{\sqrt{\hat{n}}} \hat{\boldsymbol{y}} + o(1) \quad \text{by Lemma A.2}$$

$$= (\frac{1}{\sqrt{\hat{n}}} \hat{\boldsymbol{R}}_{\text{w}}^\top \boldsymbol{w}_{\text{w}} - \frac{1}{\sqrt{\hat{n}}} \hat{\boldsymbol{y}})^\top (\boldsymbol{P}_s \boldsymbol{P}_s - \boldsymbol{P}_s)^\top \frac{1}{\sqrt{\hat{n}}} \hat{\boldsymbol{y}} + o(1). \tag{16}$$

$\boldsymbol{P}_s \boldsymbol{P}_s - \boldsymbol{P}_s$'s eigenvalues are given by: $(\frac{\lambda_i(\frac{1}{\hat{n}} \hat{\boldsymbol{K}}_s')}{\lambda_i(\frac{1}{\hat{n}} \hat{\boldsymbol{K}}_s') + (\hat{\gamma}_s + \beta_s)})^2 - \frac{\lambda_i(\frac{1}{\hat{n}} \hat{\boldsymbol{K}}_s')}{\lambda_i(\frac{1}{\hat{n}} \hat{\boldsymbol{K}}_s') + (\hat{\gamma}_s + \beta_s)} = -(\frac{\lambda_i(\frac{1}{\hat{n}} \hat{\boldsymbol{K}}_s')}{\lambda_i(\frac{1}{\hat{n}} \hat{\boldsymbol{K}}_s') + (\hat{\gamma}_s + \beta_s)})(\frac{\hat{\gamma}_s + \beta_s}{\lambda_i(\frac{1}{\hat{n}} \hat{\boldsymbol{K}}_s') + (\hat{\gamma}_s + \beta_s)})$.
since $\frac{1}{\hat{n}} \hat{\boldsymbol{K}}_s'$ and $\hat{\boldsymbol{\Sigma}}_s$ share non-zero eigenvalues, we analyze the relation between $\beta_s + \hat{\gamma}_s$ and $\hat{\boldsymbol{\Sigma}}_s'$'s non-zero eigenvalues. By **Concentration on $\mathcal{V}$** and Weyl's Theorem

$$|\lambda_{\min, \neq 0}(\hat{\boldsymbol{\Sigma}}_s') - \lambda_{\min, \neq 0}(\boldsymbol{\Sigma}_s')| = o(\gamma_s^2 + \delta_s^2 + \lambda_{\min, \neq 0}(\boldsymbol{\Sigma}_s'))$$

Combining this with $\beta_s + \hat{\gamma}_s = o(\lambda_{\min, \neq 0}(\boldsymbol{\Sigma}_s'))$, we conclude:

$$\beta_s + \hat{\gamma}_s = o(\lambda_{\min, \neq 0}(\hat{\boldsymbol{\Sigma}}_s')). \tag{17}$$

Using Equation 17, we then obtain $\|\boldsymbol{P}_s \boldsymbol{P}_s - \boldsymbol{P}_s\|_{\text{op}} = o(1)$. By Lemma A.12, the term $(\frac{1}{\sqrt{\hat{n}}} \hat{\boldsymbol{R}}_{\text{w}}^\top \boldsymbol{w}_{\text{w}} - \frac{1}{\sqrt{\hat{n}}} \hat{\boldsymbol{y}})$ can be bounded by $\left\| (\boldsymbol{I} - \boldsymbol{P}_{\text{w}}) \frac{1}{\sqrt{\hat{n}}} \hat{\boldsymbol{y}} \right\| + o(1) = O(1)$, and $\|\frac{1}{\sqrt{\hat{n}}} \hat{\boldsymbol{y}}\| = O(1)$ by **Boundedness**. Combining all these results, the RHS of Equation 16 is $o(1)$. Therefore, $|(\boldsymbol{w}_{\text{w2s}} - \boldsymbol{w}_{\text{sc}})^\top (\boldsymbol{\Sigma}_s \boldsymbol{w}_{\text{sc}} - \mathbb{E}[\boldsymbol{r}_s y])| = o(1)$, which completes the proof. □

### B.3. Proof of results in Section 4

B.3.1. PROOF OF THEOREM 4.1

First, we present the following lemma, which provides a sufficient condition under which any labeling can be fitted by the W2S model.

**Lemma B.4** (Condition for overfitting arbitrary labels). *As long as $\delta_s = o(\hat{\gamma}_s)$ and $\delta_s \leq \beta_s = o(\hat{\gamma}_s)$, given any $f_w \circ h_w$ s.t. $\frac{1}{\hat{n}} \sum_{i=1}^{\hat{n}} f_w(h_w(\hat{\boldsymbol{x}}_i))^2 = O(1)$, the weak-to-strong model can almost exactly overfit it, as indicated by an almost zero training error: $\frac{1}{\hat{n}} \sum_{i=1}^{\hat{n}} (f_{w2s}(h_s(\hat{\boldsymbol{x}}_i)) - f_w(h_w(\hat{\boldsymbol{x}}_i)))^2 = o(1)$, with high probability $1 - o(1)$.*

*Proof.* Let $\hat{\boldsymbol{T}} \in \mathbb{R}^{\hat{n}}$ denote the weak model's predictions on $\hat{\mathcal{D}}$. The following holds for all $\hat{\boldsymbol{T}}$ such that $\frac{1}{\hat{n}} |\hat{\boldsymbol{T}}|^2 = O(1)$. The training loss can be expressed as

$$\frac{1}{\hat{n}} \|\hat{\boldsymbol{R}}_s^\top \boldsymbol{w}_{\text{w2s}} - \hat{\boldsymbol{T}}\|^2 = \|\frac{1}{\sqrt{\hat{n}}} \hat{\boldsymbol{R}}_s^\top \frac{1}{\sqrt{\hat{n}}} \hat{\boldsymbol{R}}_s (\frac{1}{\hat{n}} \hat{\boldsymbol{R}}_s^\top \hat{\boldsymbol{R}}_s + \beta_s \boldsymbol{I})^{-1} \frac{1}{\sqrt{\hat{n}}} \hat{\boldsymbol{T}} - \frac{1}{\sqrt{\hat{n}}} \hat{\boldsymbol{T}}\|^2 \quad \text{by Equation 3}$$

$$= \|\left( \frac{1}{\sqrt{\hat{n}}} \hat{\boldsymbol{R}}_s^\top \frac{1}{\sqrt{\hat{n}}} \hat{\boldsymbol{R}}_s (\frac{1}{\hat{n}} \hat{\boldsymbol{R}}_s^\top \hat{\boldsymbol{R}}_s + \beta_s \boldsymbol{I})^{-1} - \boldsymbol{I} \right) \frac{1}{\sqrt{\hat{n}}} \hat{\boldsymbol{T}}\|^2$$

$$\leq \|\frac{1}{\sqrt{\hat{n}}} \hat{\boldsymbol{R}}_s^\top \frac{1}{\sqrt{\hat{n}}} \hat{\boldsymbol{R}}_s (\frac{1}{\hat{n}} \hat{\boldsymbol{R}}_s^\top \hat{\boldsymbol{R}}_s + \beta_s \boldsymbol{I})^{-1} - \boldsymbol{I}\|_{\text{op}}^2 \|\frac{1}{\sqrt{\hat{n}}} \hat{\boldsymbol{T}}\|^2$$

$$= \left( \frac{\beta_s}{\lambda_{\min}(\frac{1}{\hat{n}} \hat{\boldsymbol{R}}_s^\top \hat{\boldsymbol{R}}_s) + \beta_s} \right)^2 \|\frac{1}{\sqrt{\hat{n}}} \hat{\boldsymbol{T}}\|^2$$

$$= O\left( \left( \frac{\beta_s}{\lambda_{\min}(\frac{1}{\hat{n}} \hat{\boldsymbol{R}}_s^\top \hat{\boldsymbol{R}}_s) + \beta_s} \right)^2 \right) \quad \text{because we assume } \frac{1}{\hat{n}} \|\hat{\boldsymbol{T}}\|^2 = O(1). \tag{18}$$

By Lemma A.4 and Weyl's Theorem, we have

$$|\lambda_{\min}(\frac{1}{\hat{n}}\hat{R}_s^\top\hat{R}_s) - \lambda_{\min}(\frac{1}{\hat{n}}\hat{R}_s^\top U_s'U_s'^\top\hat{R}_s + \hat{\gamma}_s I)| \leq \left\|\frac{1}{\hat{n}}\hat{R}_s^\top\hat{R}_s - \left(\frac{1}{\hat{n}}\hat{R}_s^\top U_s'U_s'^\top\hat{R}_s + \hat{\gamma}_s I\right)\right\|_{\text{op}} = o(\gamma_s^2 + \delta_s^2)$$

$$\implies \lambda_{\min}(\frac{1}{\tilde{n}}\hat{R}_s^\top\hat{R}_s) \geq \lambda_{\min}(\frac{1}{\hat{n}}\hat{R}_s^\top U_s'U_s'^\top\hat{R}_s + \hat{\gamma}_s I) - o(\gamma_s^2 + \delta_s^2) \geq \hat{\gamma}_s - o(\gamma_s^2 + \delta_s^2). \tag{19}$$

Substituding Equation 19 into Equation 18 yields

$$\frac{1}{\hat{n}}\|\hat{R}_s^\top w_{\text{w2s}} - \hat{T}\|^2 = O\left(\left(\frac{\beta_s}{\hat{\gamma}_s - o(\gamma_s^2 + \delta_s^2) + \beta_s}\right)^2\right) = o(1) \qquad \text{because we assume } \beta_s = o(\hat{\gamma}_s) \text{ and } \delta_s = o(\hat{\gamma}_s),$$

which completes the proof. $\qquad\square$

The first statement in Theorem 4.1 can now be readily proved by invoking the above lemma.

For the second statement in Theorem 4.1, we first apply the triangle inequality, which gives $\sqrt{\text{Err}_{\text{w2s}}} \leq \sqrt{\textbf{PredGap}} + \sqrt{\text{Err}_{\text{sc}}}$. Given the assumption $\text{Err}_{\text{sc}} = o(1)$ and the fact that Theorem 3.8 implies $\textbf{PredGap} = O(1)$, we obtain $\text{Err}_{\text{w2s}} \leq \textbf{PredGap} + o(1)$. Furthermore, by our assumption combined with Theorem 3.8, we know $\textbf{PredGap} = \text{Err}_{\text{w}} - \Delta + o(1)$. Substituting this into the previous inequality yields $\text{Err}_{\text{w2s}} \leq \text{Err}_{\text{w}} - \Delta + o(1)$.

### B.3.2. PROOF OF COROLLARY 4.3

We begin by presenting the following general result regarding the test errors of the weak model and the strong ceiling model.

**Lemma B.5** (The weak model's error on the population). *If $|\mathbb{E}[y^2] - \frac{1}{\hat{n}}\sum_{i=1}^{\hat{n}}\hat{y}_i^2| = o(1)$ w.h.p., then the weak model's error on the population, $Err_w$, can be approximated as follows,*

$$Err_w = \|(I - P_w)\frac{1}{\sqrt{\hat{n}}}\hat{y}\|^2 \pm o(1).$$

*A similar conclusion holds for the strong ceiling's error $Err_{sc}$ as well: $Err_{sc} = \|(I - P_s)\frac{1}{\sqrt{\hat{n}}}\hat{y}\|^2 \pm o(1)$.*

*Proof.* We decompose the error as follows

$$\text{Err}_{\text{w}} = \mathbb{E}[(r_w^\top w_w - y)^2]$$
$$= w_w^\top\Sigma_w w_w - 2w_w^\top\mathbb{E}[r_w y] + \mathbb{E}[y^2]. \tag{20}$$

The first term can further be decomposed as:

$$w_w^\top\Sigma_w w_w = \|\sqrt{\Lambda_w}U_w^\top\frac{1}{\sqrt{\tilde{n}}}\tilde{R}_w(\frac{1}{\tilde{n}}\tilde{R}_w^\top\tilde{R}_w + \beta_w I)^{-1}\frac{1}{\sqrt{\tilde{n}}}\tilde{y}\|^2$$

$$= \|\sqrt{\Lambda_w}U_w^\top\frac{1}{\sqrt{\tilde{n}}}\tilde{R}_w(\frac{1}{\tilde{n}}\tilde{R}_w^\top U_w'U_w'^\top\tilde{R}_w + (\beta_w + \tilde{\gamma}_w)I)^{-1}\frac{1}{\sqrt{\tilde{n}}}\tilde{y}\|^2 \pm o(1) \quad \text{by Lemma A.4 and \textbf{Boundedness}}$$

$$= \|\begin{bmatrix}\sqrt{\Lambda_w'}U_w'^\top \\ \sqrt{\Lambda_w''}U_w''^\top\end{bmatrix}\frac{1}{\sqrt{\tilde{n}}}\tilde{R}_w(\frac{1}{\tilde{n}}\tilde{R}_w^\top U_w'U_w'^\top\tilde{R}_w + (\beta_w + \tilde{\gamma}_w)I)^{-1}\frac{1}{\sqrt{\tilde{n}}}\tilde{y}\|^2 \pm o(1)$$

$$= \|\sqrt{\Lambda_w'}U_w'^\top\frac{1}{\sqrt{\tilde{n}}}\tilde{R}_w(\frac{1}{\tilde{n}}\tilde{R}_w^\top U_w'U_w'^\top\tilde{R}_w + (\beta_w + \tilde{\gamma}_w)I)^{-1}\frac{1}{\sqrt{\tilde{n}}}\tilde{y}\|^2$$

$$+ \|\sqrt{\Lambda_w''}U_w''^\top\frac{1}{\sqrt{\tilde{n}}}\tilde{R}_w(\frac{1}{\tilde{n}}\tilde{R}_w^\top U_w'U_w'^\top\tilde{R}_w + (\beta_w + \tilde{\gamma}_w)I)^{-1}\frac{1}{\sqrt{\tilde{n}}}\tilde{y}\|^2 \pm o(1) \tag{21}$$

We bound the second term in Equation 21:

$$
\begin{aligned}
&\|\sqrt{\boldsymbol{\Lambda}_{\mathrm{w}}''}\boldsymbol{U}_{\mathrm{w}}''^{\top}\frac{1}{\sqrt{\tilde{n}}}\tilde{\boldsymbol{R}}_{\mathrm{w}}(\frac{1}{\tilde{n}}\tilde{\boldsymbol{R}}_{\mathrm{w}}^{\top}\boldsymbol{U}_{\mathrm{w}}'\boldsymbol{U}_{\mathrm{w}}'^{\top}\tilde{\boldsymbol{R}}_{\mathrm{w}}+(\beta_{\mathrm{w}}+\tilde{\gamma}_{\mathrm{w}})\boldsymbol{I})^{-1}\frac{1}{\sqrt{\tilde{n}}}\tilde{\boldsymbol{y}}\|^2 \\
\leq&\|\boldsymbol{\Lambda}_{\mathrm{w}}''\|_{\mathrm{op}}\|\boldsymbol{U}_{\mathrm{w}}''^{\top}\frac{1}{\sqrt{\tilde{n}}}\tilde{\boldsymbol{R}}_{\mathrm{w}}\|_{\mathrm{op}}^2\|(\frac{1}{\tilde{n}}\tilde{\boldsymbol{R}}_{\mathrm{w}}^{\top}\boldsymbol{U}_{\mathrm{w}}'\boldsymbol{U}_{\mathrm{w}}'^{\top}\tilde{\boldsymbol{R}}_{\mathrm{w}}+(\beta_{\mathrm{w}}+\tilde{\gamma}_{\mathrm{w}})\boldsymbol{I})^{-1}\|_{\mathrm{op}}^2\|\frac{1}{\sqrt{\tilde{n}}}\tilde{\boldsymbol{y}}\|^2 \\
\leq&\frac{o(\gamma_{\mathrm{w}}+\delta_{\mathrm{w}})(o(\gamma_{\mathrm{w}}^2+\delta_{\mathrm{w}}^2)+\tilde{\gamma}_{\mathrm{w}})}{(\beta_{\mathrm{w}}+\tilde{\gamma}_{\mathrm{w}})^2} \quad \text{by \textbf{Diminishing population covariance on }}\mathcal{V}^{\perp}\text{, Lemma A.11 and \textbf{Boundedness}} \\
=&o(1).
\end{aligned}
\tag{22}
$$

Then, we approximate the first term in Equation 21:

$$
\begin{aligned}
&\|\sqrt{\boldsymbol{\Lambda}_{\mathrm{w}}'}\boldsymbol{U}_{\mathrm{w}}'^{\top}\frac{1}{\sqrt{\tilde{n}}}\tilde{\boldsymbol{R}}_{\mathrm{w}}(\frac{1}{\tilde{n}}\tilde{\boldsymbol{R}}_{\mathrm{w}}^{\top}\boldsymbol{U}_{\mathrm{w}}'\boldsymbol{U}_{\mathrm{w}}'^{\top}\tilde{\boldsymbol{R}}_{\mathrm{w}}+(\beta_{\mathrm{w}}+\tilde{\gamma}_{\mathrm{w}})\boldsymbol{I})^{-1}\frac{1}{\sqrt{\tilde{n}}}\tilde{\boldsymbol{y}}\|^2 \\
=&\|\sqrt{\boldsymbol{\Lambda}_{\mathrm{w}}'}(\frac{1}{\tilde{n}}\boldsymbol{U}_{\mathrm{w}}'^{\top}\tilde{\boldsymbol{R}}_{\mathrm{w}}\tilde{\boldsymbol{R}}_{\mathrm{w}}^{\top}\boldsymbol{U}_{\mathrm{w}}'+(\beta_{\mathrm{w}}+\tilde{\gamma}_{\mathrm{w}})\boldsymbol{I})^{-1}\boldsymbol{U}_{\mathrm{w}}'^{\top}\frac{1}{\sqrt{\tilde{n}}}\tilde{\boldsymbol{R}}_{\mathrm{w}}\frac{1}{\sqrt{\tilde{n}}}\tilde{\boldsymbol{y}}\|^2 \quad \text{by Lemma A.2} \\
=&\|\sqrt{\boldsymbol{\Lambda}_{\mathrm{w}}'}(\boldsymbol{\Lambda}_{\mathrm{w}}'+(\beta_{\mathrm{w}}+\tilde{\gamma}_{\mathrm{w}})\boldsymbol{I})^{-1}\boldsymbol{U}_{\mathrm{w}}'^{\top}\frac{1}{\sqrt{\tilde{n}}}\tilde{\boldsymbol{R}}_{\mathrm{w}}\frac{1}{\sqrt{\tilde{n}}}\tilde{\boldsymbol{y}}\|^2\pm o(1) \quad \text{by Lemma A.6 and \textbf{Boundedness}} \\
=&\|\sqrt{\boldsymbol{\Lambda}_{\mathrm{w}}'}(\frac{1}{\hat{n}}\boldsymbol{U}_{\mathrm{w}}'^{\top}\hat{\boldsymbol{R}}_{\mathrm{w}}\hat{\boldsymbol{R}}_{\mathrm{w}}^{\top}\boldsymbol{U}_{\mathrm{w}}'+(\beta_{\mathrm{w}}+\tilde{\gamma}_{\mathrm{w}})\boldsymbol{I})^{-1}\boldsymbol{U}_{\mathrm{w}}'^{\top}\frac{1}{\sqrt{\tilde{n}}}\tilde{\boldsymbol{R}}_{\mathrm{w}}\frac{1}{\sqrt{\tilde{n}}}\tilde{\boldsymbol{y}}\|^2\pm o(1) \quad \text{by Lemma A.6 and \textbf{Boundedness}} \\
=&\|\sqrt{\boldsymbol{\Lambda}_{\mathrm{w}}'}(\frac{1}{\hat{n}}\boldsymbol{U}_{\mathrm{w}}'^{\top}\hat{\boldsymbol{R}}_{\mathrm{w}}\hat{\boldsymbol{R}}_{\mathrm{w}}^{\top}\boldsymbol{U}_{\mathrm{w}}'+(\beta_{\mathrm{w}}+\tilde{\gamma}_{\mathrm{w}})\boldsymbol{I})^{-1}\boldsymbol{U}_{\mathrm{w}}'^{\top}\frac{1}{\sqrt{\hat{n}}}\hat{\boldsymbol{R}}_{\mathrm{w}}\frac{1}{\sqrt{\hat{n}}}\hat{\boldsymbol{y}}\|^2\pm o(1) \quad \text{by \textbf{Concentration on }}\mathcal{V}\text{ and \textbf{Boundedness}} \\
=&\|\sqrt{\boldsymbol{\Lambda}_{\mathrm{w}}'}\boldsymbol{U}_{\mathrm{w}}'^{\top}\frac{1}{\sqrt{\hat{n}}}\hat{\boldsymbol{R}}_{\mathrm{w}}(\frac{1}{\hat{n}}\hat{\boldsymbol{R}}_{\mathrm{w}}^{\top}\boldsymbol{U}_{\mathrm{w}}'\boldsymbol{U}_{\mathrm{w}}'^{\top}\hat{\boldsymbol{R}}_{\mathrm{w}}+(\beta_{\mathrm{w}}+\tilde{\gamma}_{\mathrm{w}})\boldsymbol{I})^{-1}\frac{1}{\sqrt{\hat{n}}}\hat{\boldsymbol{y}}\|^2\pm o(1) \quad \text{by Lemma A.2} \\
=&\|\frac{1}{\hat{n}}\hat{\boldsymbol{R}}_{\mathrm{w}}^{\top}\boldsymbol{U}_{\mathrm{w}}'\boldsymbol{U}_{\mathrm{w}}'^{\top}\hat{\boldsymbol{R}}_{\mathrm{w}}(\frac{1}{\hat{n}}\hat{\boldsymbol{R}}_{\mathrm{w}}^{\top}\boldsymbol{U}_{\mathrm{w}}'\boldsymbol{U}_{\mathrm{w}}'^{\top}\hat{\boldsymbol{R}}_{\mathrm{w}}+(\beta_{\mathrm{w}}+\tilde{\gamma}_{\mathrm{w}})\boldsymbol{I})^{-1}\frac{1}{\sqrt{\hat{n}}}\hat{\boldsymbol{y}}\|^2\pm o(1) \quad \text{by Corollary A.10 and \textbf{Boundedness}} \\
=&\|\boldsymbol{P}_{\mathrm{w}}\frac{1}{\sqrt{\hat{n}}}\hat{\boldsymbol{y}}\|^2\pm o(1).
\end{aligned}
\tag{23}
$$

Now, we approximate the second term in Equation 20:

$$\boldsymbol{w}_{\mathrm{w}}^{\top}\mathbb{E}[\boldsymbol{r}_{\mathrm{w}}y]$$

$$=(\frac{1}{\sqrt{\tilde{n}}}\tilde{\boldsymbol{R}}_{\mathrm{w}}(\frac{1}{\tilde{n}}\tilde{\boldsymbol{R}}_{\mathrm{w}}^{\top}\tilde{\boldsymbol{R}}_{\mathrm{w}}+\beta_{\mathrm{w}}\boldsymbol{I})^{-1}\frac{1}{\sqrt{\tilde{n}}}\tilde{\boldsymbol{y}})^{\top}\mathbb{E}[\boldsymbol{r}_{\mathrm{w}}y]$$

$$=(\frac{1}{\sqrt{\tilde{n}}}\tilde{\boldsymbol{R}}_{\mathrm{w}}(\frac{1}{\tilde{n}}\tilde{\boldsymbol{R}}_{\mathrm{w}}^{\top}\tilde{\boldsymbol{R}}_{\mathrm{w}}+\beta_{\mathrm{w}}\boldsymbol{I})^{-1}\frac{1}{\sqrt{\tilde{n}}}\tilde{\boldsymbol{y}})^{\top}\boldsymbol{U}_{\mathrm{w}}'\boldsymbol{U}_{\mathrm{w}}'^{\top}\mathbb{E}[\boldsymbol{r}_{\mathrm{w}}y]+(\frac{1}{\sqrt{\tilde{n}}}\tilde{\boldsymbol{R}}_{\mathrm{w}}(\frac{1}{\tilde{n}}\tilde{\boldsymbol{R}}_{\mathrm{w}}^{\top}\tilde{\boldsymbol{R}}_{\mathrm{w}}+\beta_{\mathrm{w}}\boldsymbol{I})^{-1}\frac{1}{\sqrt{\tilde{n}}}\tilde{\boldsymbol{y}})^{\top}\boldsymbol{U}_{\mathrm{w}}''\boldsymbol{U}_{\mathrm{w}}''^{\top}\mathbb{E}[\boldsymbol{r}_{\mathrm{w}}y]$$

$$=(\frac{1}{\sqrt{\tilde{n}}}\tilde{\boldsymbol{R}}_{\mathrm{w}}(\frac{1}{\tilde{n}}\tilde{\boldsymbol{R}}_{\mathrm{w}}^{\top}\tilde{\boldsymbol{R}}_{\mathrm{w}}+\beta_{\mathrm{w}}\boldsymbol{I})^{-1}\frac{1}{\sqrt{\tilde{n}}}\tilde{\boldsymbol{y}})^{\top}\boldsymbol{U}_{\mathrm{w}}'\boldsymbol{U}_{\mathrm{w}}'^{\top}\mathbb{E}[\boldsymbol{r}_{\mathrm{w}}y]\pm o(\frac{\sqrt{o(\gamma_{\mathrm{w}}^{2}+\delta_{\mathrm{w}}^{2})+\tilde{\gamma}_{\mathrm{w}}}}{\tilde{\gamma}_{\mathrm{w}}+\beta_{\mathrm{w}}}\sqrt{\gamma_{\mathrm{w}}+\delta_{\mathrm{w}}})$$

by **Boundedness**, Lemmas A.5, A.11 and B.1

$$=(\frac{1}{\sqrt{\tilde{n}}}\tilde{\boldsymbol{R}}_{\mathrm{w}}(\frac{1}{\tilde{n}}\tilde{\boldsymbol{R}}_{\mathrm{w}}^{\top}\tilde{\boldsymbol{R}}_{\mathrm{w}}+\beta_{\mathrm{w}}\boldsymbol{I})^{-1}\frac{1}{\sqrt{\tilde{n}}}\tilde{\boldsymbol{y}})^{\top}\boldsymbol{U}_{\mathrm{w}}'\boldsymbol{U}_{\mathrm{w}}'^{\top}\mathbb{E}[\boldsymbol{r}_{\mathrm{w}}y]\pm o(1)$$

$$=(\frac{1}{\sqrt{\tilde{n}}}\tilde{\boldsymbol{R}}_{\mathrm{w}}(\frac{1}{\tilde{n}}\tilde{\boldsymbol{R}}_{\mathrm{w}}^{\top}\boldsymbol{U}_{\mathrm{w}}'\boldsymbol{U}_{\mathrm{w}}'^{\top}\tilde{\boldsymbol{R}}_{\mathrm{w}}+(\beta_{\mathrm{w}}+\tilde{\gamma}_{\mathrm{w}})\boldsymbol{I})^{-1}\frac{1}{\sqrt{\tilde{n}}}\tilde{\boldsymbol{y}})^{\top}\boldsymbol{U}_{\mathrm{w}}'\boldsymbol{U}_{\mathrm{w}}'^{\top}\mathbb{E}[\boldsymbol{r}_{\mathrm{w}}y]\pm o(1)$$

by Lemma A.5, **Boundedness**, and Lemma B.2

$$=\frac{1}{\sqrt{\tilde{n}}}\tilde{\boldsymbol{y}}^{\top}(\frac{1}{\tilde{n}}\tilde{\boldsymbol{R}}_{\mathrm{w}}^{\top}\boldsymbol{U}_{\mathrm{w}}'\boldsymbol{U}_{\mathrm{w}}'^{\top}\tilde{\boldsymbol{R}}_{\mathrm{w}}+(\beta_{\mathrm{w}}+\tilde{\gamma}_{\mathrm{w}})\boldsymbol{I})^{-1}\frac{1}{\sqrt{\tilde{n}}}\tilde{\boldsymbol{R}}_{\mathrm{w}}^{\top}\boldsymbol{U}_{\mathrm{w}}'\boldsymbol{U}_{\mathrm{w}}'^{\top}\mathbb{E}[\boldsymbol{r}_{\mathrm{w}}y]\pm o(1)$$

$$=\frac{1}{\sqrt{\tilde{n}}}\tilde{\boldsymbol{y}}^{\top}\frac{1}{\sqrt{\tilde{n}}}\tilde{\boldsymbol{R}}_{\mathrm{w}}^{\top}\boldsymbol{U}_{\mathrm{w}}'(\frac{1}{\tilde{n}}\boldsymbol{U}_{\mathrm{w}}'^{\top}\tilde{\boldsymbol{R}}_{\mathrm{w}}\tilde{\boldsymbol{R}}_{\mathrm{w}}^{\top}\boldsymbol{U}_{\mathrm{w}}'+(\beta_{\mathrm{w}}+\tilde{\gamma}_{\mathrm{w}})\boldsymbol{I})^{-1}\boldsymbol{U}_{\mathrm{w}}'^{\top}\mathbb{E}[\boldsymbol{r}_{\mathrm{w}}y]\pm o(1)\quad\text{by Lemma A.2}$$

$$=\frac{1}{\sqrt{\tilde{n}}}\tilde{\boldsymbol{y}}^{\top}\frac{1}{\sqrt{\tilde{n}}}\tilde{\boldsymbol{R}}_{\mathrm{w}}^{\top}\boldsymbol{U}_{\mathrm{w}}'(\frac{1}{\tilde{n}}\boldsymbol{U}_{\mathrm{w}}'^{\top}\hat{\boldsymbol{R}}_{\mathrm{w}}\hat{\boldsymbol{R}}_{\mathrm{w}}^{\top}\boldsymbol{U}_{\mathrm{w}}'+(\beta_{\mathrm{w}}+\tilde{\gamma}_{\mathrm{w}})\boldsymbol{I})^{-1}\boldsymbol{U}_{\mathrm{w}}'^{\top}\mathbb{E}[\boldsymbol{r}_{\mathrm{w}}y]\pm o(1)$$

by Lemma A.6, Lemma B.2, and **Boundedness**

$$=\frac{1}{\hat{n}}\hat{\boldsymbol{y}}^{\top}\hat{\boldsymbol{R}}_{\mathrm{w}}^{\top}\boldsymbol{U}_{\mathrm{w}}'(\frac{1}{\tilde{n}}\boldsymbol{U}_{\mathrm{w}}'^{\top}\hat{\boldsymbol{R}}_{\mathrm{w}}\hat{\boldsymbol{R}}_{\mathrm{w}}^{\top}\boldsymbol{U}_{\mathrm{w}}'+(\beta_{\mathrm{w}}+\tilde{\gamma}_{\mathrm{w}})\boldsymbol{I})^{-1}\boldsymbol{U}_{\mathrm{w}}'^{\top}\frac{1}{\hat{n}}\hat{\boldsymbol{R}}_{\mathrm{w}}\hat{\boldsymbol{y}}\pm o(1)$$

by **Concentration on** $\mathcal{V}$, **Boundedness** and Lemma B.2

$$=\frac{1}{\sqrt{\hat{n}}}\hat{\boldsymbol{y}}^{\top}\boldsymbol{P}_{\mathrm{w}}\frac{1}{\sqrt{\hat{n}}}\hat{\boldsymbol{y}}\pm o(1)\quad\text{by Lemma A.2.}\tag{24}$$

Combining Equations 20, 21, 22, 23, 24, and the assumption about $\mathbb{E}[y^2]$ yields

$$\mathrm{Err}_{\mathrm{w}}=\|(\boldsymbol{I}-\boldsymbol{P}_{\mathrm{w}})\frac{1}{\sqrt{\hat{n}}}\hat{\boldsymbol{y}}\|^{2}\pm o(1).$$

The proof of the result concerning $\mathrm{Err}_{\mathrm{sc}}$ is similar. □

We show that the condition regarding $\mathbb{E}[y^2]$ is satisfied in Example 4.2. Specifically, $\sum_{i=1}^{\hat{n}}\hat{y}_i^2$ follows a $\chi^2(\hat{n})$ distribution, with a mean of $\hat{n}\mathbb{E}[y^2]$ and a variance of $2\hat{n}$. For simplicity, we demonstrate the following result using Chebyshev's inequality, while noting that tighter bounds could be achieved with tail bounds for $\chi^2$ variables or Lemma C.3. For any $k > 0$, we have: $\Pr\left(|\sum_{i=1}^{\hat{n}}\hat{y}_i^2 - \hat{n}\mathbb{E}[y^2]| \ge k\sqrt{2\hat{n}}\right) \le \frac{1}{k^2}$. Letting $k = \hat{n}^{1/4}$, we find that with probability $1 - O\left(\frac{1}{\sqrt{\hat{n}}}\right)$, $\left|\frac{1}{\hat{n}}\sum_{i=1}^{\hat{n}}\hat{y}_i^2 - \mathbb{E}[y^2]\right| = O\left(\frac{1}{\hat{n}^{1/4}}\right)$. Thus, Lemma B.5 applies to Example 4.2.

Now, based on Lemmas A.12, B.5, and Theorem 3.8, the key to computing the errors of all these models boils down to simply computing $\boldsymbol{P}_{\mathrm{w}}$ and $\boldsymbol{P}_{\mathrm{s}}$.

We first compute the kernels. For convenience, we use the shorthand notations $\hat{K}_{\mathrm{w}}$ and $\hat{K}_{\mathrm{s}}$ to represent $\hat{K}(\boldsymbol{\Pi}_{\mathcal{V}_{\mathrm{w}}}h_{\mathrm{w}})$ and $\hat{K}(\boldsymbol{\Pi}_{\mathcal{V}_{\mathrm{s}}}h_{\mathrm{s}})$, respectively. Since the representations in Example 4.2 are decomposable with respect to the subspace corresponding to the first coordinate, for both the weak and strong models, the principal kernels are rank one and can be expressed as $\hat{K}_{\mathrm{w}} = \boldsymbol{q}\boldsymbol{q}^{\top}$ and $\hat{K}_{\mathrm{s}} = \hat{\boldsymbol{y}}\hat{\boldsymbol{y}}^{\top}$, where $\hat{\boldsymbol{q}} := \sqrt{\eta}\hat{\boldsymbol{y}} + \sqrt{1-\eta}\hat{\boldsymbol{\zeta}}$. Then, for $\frac{1}{\hat{n}}\hat{K}_{\mathrm{w}}$, it has a single nonzero eigenvalue $\|\frac{1}{\sqrt{\hat{n}}}\hat{\boldsymbol{q}}\|^2$, with the corresponding eigenvector $\frac{1}{\|\frac{1}{\sqrt{\hat{n}}}\hat{\boldsymbol{q}}\|}\frac{1}{\sqrt{\hat{n}}}\hat{\boldsymbol{q}}$. Similarly, $\frac{1}{\hat{n}}\hat{K}_{\mathrm{s}}$ has a single eigenvalue $\|\frac{1}{\sqrt{\hat{n}}}\hat{\boldsymbol{y}}\|^2$, with the corresponding eigenvector $\frac{1}{\|\frac{1}{\sqrt{\hat{n}}}\hat{\boldsymbol{y}}\|}\frac{1}{\sqrt{\hat{n}}}\hat{\boldsymbol{y}}$.

Next, we present the following Lemma.

**Lemma B.6.** *We have the following:*

$$\|\frac{1}{\sqrt{\hat{n}}}\hat{\boldsymbol{y}}\|^2 = 1 \pm o(1), \ \ \|\frac{1}{\sqrt{\hat{n}}}\hat{\boldsymbol{\zeta}}\|^2 = 1 \pm o(1), \ \ |\frac{1}{\sqrt{\hat{n}}}\hat{\boldsymbol{\zeta}}^\top \frac{1}{\sqrt{\hat{n}}}\hat{\boldsymbol{y}}| = o(1), \ \ \|\frac{1}{\sqrt{\hat{n}}}\hat{\boldsymbol{q}}\|^2 = 1 \pm o(1)$$

*Proof.* The first two statements can be proved by leveraging classical results on the concentration of Gaussian matrices (see Lemma C.3 for details). The third statement follows as a special case of Lemma C.4. The last statement is implied by the previous three. $\qquad\square$

Recall that both the weak and strong models' representations in Example 4.2 are special cases of Example 3.5. Given that $\sigma^2 = o(\hat{n})$ and $\tilde{n} = \Theta(\hat{n})$, we have $\hat{\gamma}_\text{w}$, $\tilde{\gamma}_\text{w}$, $\hat{\gamma}_\text{s}$, and $\tilde{\gamma}_\text{s}$ all being $o(1)$, $\delta_\text{w} = \delta_\text{s} = 0$, and $\beta_\text{w} = o(1), \beta_\text{s} = o(1)$. Combining these with Lemma B.6, we derive:

$$\|\boldsymbol{P}_\text{w} - \frac{1}{\hat{n}}\hat{\boldsymbol{q}}\hat{\boldsymbol{q}}^\top\|_\text{op} = o(1), \|\boldsymbol{P}_\text{s} - \frac{1}{\hat{n}}\hat{\boldsymbol{y}}\hat{\boldsymbol{y}}^\top\|_\text{op} = o(1).$$

Now, leveraging Lemma B.6, we can derive all the errors using the expressions provided in Lemmas A.12, B.5, and Theorem 3.8.

### B.4. Proof of Corollary 5.2

Following Theorem 3.8, we bound the RHS as follows

$$\begin{aligned}
\textbf{PredGap} &= \|\boldsymbol{P}_\text{s}(\boldsymbol{I} - \boldsymbol{P}_\text{w})\boldsymbol{P}_\text{s}\frac{1}{\sqrt{\hat{n}}}\hat{\boldsymbol{y}} + \boldsymbol{P}_\text{s}(\boldsymbol{I} - \boldsymbol{P}_\text{w})(\boldsymbol{I} - \boldsymbol{P}_\text{s})\frac{1}{\sqrt{\hat{n}}}\hat{\boldsymbol{y}}\|^2 \pm o(1) \\
&\leq \left(\|\boldsymbol{P}_\text{s}(\boldsymbol{I} - \boldsymbol{P}_\text{w})\boldsymbol{P}_\text{s}\|_\text{op}\|\frac{1}{\sqrt{\hat{n}}}\hat{\boldsymbol{y}}\| + \|\boldsymbol{P}_\text{s}(\boldsymbol{I} - \boldsymbol{P}_\text{w})\|_\text{op}\|(\boldsymbol{I} - \boldsymbol{P}_\text{s})\frac{1}{\sqrt{\hat{n}}}\hat{\boldsymbol{y}}\|\right)^2 + o(1) && (25) \\
&\leq \left(\|\boldsymbol{P}_\text{s}(\boldsymbol{I} - \boldsymbol{P}_\text{w})\boldsymbol{P}_\text{s}\|_\text{op}\sqrt{C} + \|(\boldsymbol{I} - \boldsymbol{P}_\text{s})\frac{1}{\sqrt{\hat{n}}}\hat{\boldsymbol{y}}\|\right)^2 + o(1) \\
&= \left(\|\boldsymbol{P}_\text{s}(\boldsymbol{I} - \boldsymbol{P}_\text{w})\boldsymbol{P}_\text{s}\|_\text{op}\sqrt{C} + \sqrt{\text{Err}_\text{sc} + o(1)}\right)^2 + o(1) && \text{by Lemma B.5} \\
&= \left(\|\boldsymbol{P}_\text{s}(\boldsymbol{I} - \boldsymbol{P}_\text{w})\boldsymbol{P}_\text{s}\|_\text{op}\sqrt{C} + \sqrt{\text{Err}_\text{sc}}\right)^2 + o(1)
\end{aligned}$$

## C. Proof of Examples in Section 3.3

### C.1. Example 3.4

For convenience, let $q = \textbf{intdim}(\boldsymbol{\Sigma})$ and $\tau = \|\boldsymbol{\Sigma}\|_\text{op}$.

Firstly, we note that the conditions in the example imply a low intrinsic dimension. Here's why: since $\text{Tr}(\boldsymbol{\Sigma}) = \mathbb{E}|\boldsymbol{r}|^2 \leq B$, it follows that

$$\textbf{intdim}(\boldsymbol{\Sigma}) = \frac{\text{Tr}(\boldsymbol{\Sigma})}{\|\boldsymbol{\Sigma}\|_\text{op}} \leq \frac{B}{\tau} = O(B), \tag{26}$$

where the last step holds because $\tau = \|\boldsymbol{\Sigma}\|_\text{op} = \Theta(1)$. Given that $n^{1-c} = \omega(B\log(q))$, we then have $n^{1-c} = \omega(q\log(q))$, as mentioned in the remark.

Additionally, since $\textbf{intdim}(\boldsymbol{\Sigma}) \geq 1$, Equation 26 also implies

$$B \geq \tau \quad \text{and} \quad B = \Omega(1), \tag{27}$$

which we will use later.

Next, we introduce the following two lemmas, both of which rely on the matrix Bernstein inequality with intrinsic dimension, as stated in Theorem 7.3.1 of (Tropp et al., 2015).

**Lemma C.1.** *With a probability of at least* $1 - 8q \exp(\frac{-0.5\hat{n}^{1-c}}{B\tau + (B+\tau)/3}) = 1 - o(1)$, *the following holds*

$$\|\hat{\boldsymbol{\Sigma}} - \boldsymbol{\Sigma}\|_{\mathrm{op}} \leq \hat{n}^{-0.5c}.$$

*The same conclusion applies to* $\tilde{\boldsymbol{\Sigma}}$ *as well.*

*Proof.* We prove the result for $\hat{\boldsymbol{\Sigma}}$; the result for $\tilde{\boldsymbol{\Sigma}}$ can be proved in the same way. Define $\boldsymbol{S}_i = \frac{1}{\hat{n}}(\hat{\boldsymbol{r}}_i \hat{\boldsymbol{r}}_i^\top - \boldsymbol{\Sigma})$. The random matrices $\boldsymbol{S}_i$ are independent, identically distributed, and centered. Their norms are bounded as follows

$$\|\boldsymbol{S}_i\|_{\mathrm{op}} \leq \frac{1}{\hat{n}}(\|\hat{\boldsymbol{r}}_i \hat{\boldsymbol{r}}_i^\top\|_{\mathrm{op}} + \|\boldsymbol{\Sigma}\|_{\mathrm{op}}) \leq \frac{B+\tau}{\hat{n}} := L.$$

Then,

$$\mathbb{E}\boldsymbol{S}_i^2 = \frac{1}{\hat{n}^2}\mathbb{E}(\hat{\boldsymbol{r}}_i \hat{\boldsymbol{r}}_i^\top - \boldsymbol{\Sigma})^2 = \frac{1}{\hat{n}^2}\mathbb{E}(\|\hat{\boldsymbol{r}}_i\|^2 \hat{\boldsymbol{r}}_i \hat{\boldsymbol{r}}_i^\top - 2\boldsymbol{\Sigma}^2 + \boldsymbol{\Sigma}^2) \preccurlyeq \frac{1}{\hat{n}^2}\mathbb{E}(B\hat{\boldsymbol{r}}_i \hat{\boldsymbol{r}}_i^\top - \boldsymbol{\Sigma}^2) \preccurlyeq \frac{B}{\hat{n}^2}\boldsymbol{\Sigma}$$

Define $\boldsymbol{Z} = \sum_{i=1}^{\hat{n}} \boldsymbol{S}_i$. We have

$$\boldsymbol{0} \preccurlyeq \mathbb{E}\boldsymbol{Z}^2 = \sum_{i=1}^{\hat{n}} \mathbb{E}\boldsymbol{S}_i^2 \preccurlyeq \frac{B}{\hat{n}}\boldsymbol{\Sigma} := \boldsymbol{V}$$

$\boldsymbol{V}$'s norm can be expressed as follows:

$$\|\boldsymbol{V}\|_{\mathrm{op}} = \frac{B\|\boldsymbol{\Sigma}\|_{\mathrm{op}}}{\hat{n}} = \frac{B\tau}{\hat{n}} := v$$

Define $d = \mathbf{intdim}(\begin{bmatrix} \boldsymbol{V} & \boldsymbol{0} \\ \boldsymbol{0} & \boldsymbol{V} \end{bmatrix})$, which can be simplified as:

$$d = 2\frac{\mathrm{Tr}(\frac{B}{\hat{n}}\boldsymbol{\Sigma})}{\|\frac{B}{\hat{n}}\boldsymbol{\Sigma}\|_{\mathrm{op}}} = 2\mathbf{intdim}(\frac{B}{\hat{n}}\boldsymbol{\Sigma}) = 2\mathbf{intdim}(\boldsymbol{\Sigma}) = 2q.$$

Now we are ready to apply Theorem 7.3.1 of (Tropp et al., 2015). It leads to the conclusion that, for any $t \geq \sqrt{v} + L/3$,

$$\mathbb{P}\{\|\boldsymbol{Z}\|_{\mathrm{op}} \geq t\} \leq 4d \exp\left(\frac{-t^2/2}{v + Lt/3}\right)$$

$$= 8q \exp\left(\frac{-t^2/2}{\frac{B\tau}{\hat{n}} + \frac{B+\tau}{\hat{n}}t/3}\right)$$

$$= 8q \exp\left(\frac{-\hat{n}t^2/2}{B\tau + (B+\tau)t/3}\right) \tag{28}$$

By assumption:

$$n^{1-c} = \omega(B\log q)$$

$$\implies n^{1-c} = \omega(((\tau + 1/3)B + \tau/3)\log q) \qquad \text{because } \tau = O(1)$$

$$\implies \frac{\hat{n}^{1-c}}{(\tau + 1/3)B + \tau/3} = \omega(\log q)$$

$$\implies 0.5\frac{\hat{n}^{1-c}}{(\tau + 1/3)B + \tau/3} = \omega(\log q)$$

$$\implies \exp\left(\frac{0.5\hat{n}^{1-c}}{(\tau + 1/3)B + \tau/3}\right) = \omega(q)$$

$$\implies q \exp\left(\frac{-0.5\hat{n}^{1-c}}{(\tau + 1/3)B + \tau/3}\right) = o(1) \tag{29}$$

Therefore, we set the value of $t$ to $\hat{n}^{-0.5c} = o(1)$ in Equation 28. It is easy to verify that $\hat{n}^{-0.5c} \geq \sqrt{v} + L/3$. Substituting, we get:

$$\mathbb{P}\{\|\boldsymbol{Z}\|_{\mathrm{op}} \geq \hat{n}^{-0.5c}\} \leq 4d \exp\left(\frac{-t^2/2}{v + Lt/3}\right) \leq 8q \exp\left(\frac{-\hat{n}t^2/2}{B\tau + (B+\tau)t/3}\right) = \quad 8q \exp\left(\frac{-0.5\hat{n}^{1-c}}{B\tau + (B+\tau)\hat{n}^{-0.5c}/3}\right)$$

$$\leq 8q \exp\left(\frac{-0.5\hat{n}^{1-c}}{B\tau + (B+\tau)/3}\right) \qquad \text{because } \hat{n}^{-0.5c} \leq 1$$

$$= o(1) \qquad \text{by Equation 29.}$$

Since $\boldsymbol{Z} = \hat{\boldsymbol{\Sigma}} - \boldsymbol{\Sigma}$, restating the above, we have that with a probability of at least $1 - 8q \exp\left(\frac{-0.5\hat{n}^{1-c}}{B\tau + (B+\tau)/3}\right)$, the following holds

$$\|\hat{\boldsymbol{\Sigma}} - \boldsymbol{\Sigma}\|_{\mathrm{op}} \leq \hat{n}^{-0.5c}.$$

$\square$

**Lemma C.2.** *With a probability of at least* $1 - (q+4) \exp\left(\frac{-0.5\hat{n}^{1-c}}{4BC + \frac{2}{3}\sqrt{BC}}\right) = 1 - o(1)$, *the following holds*

$$\|\frac{1}{\hat{n}} \sum_{i=1}^{\hat{n}} \hat{\boldsymbol{r}}_i y_i - \mathbb{E}[\boldsymbol{r}y]\| \leq \hat{n}^{-0.5c}.$$

*The same conclusion applies to* $\frac{1}{\tilde{n}} \sum_{i=1}^{\tilde{n}} \tilde{\boldsymbol{r}}_i y_i$ *as well.*

*Proof.* We prove the result for $\frac{1}{\hat{n}} \sum_{i=1}^{\hat{n}} \hat{\boldsymbol{r}}_i y_i$; the result for $\frac{1}{\tilde{n}} \sum_{i=1}^{\tilde{n}} \tilde{\boldsymbol{r}}_i y_i$ can be proved in the same way. Define $\boldsymbol{S}_i = \frac{1}{\hat{n}} (\hat{\boldsymbol{r}}_i y - \mathbb{E}[\boldsymbol{r}y])$. The random matrices (vectors) $\boldsymbol{S}_i$ are independent, identically distributed, and centered. Their norms are bounded as follows

$$\|\boldsymbol{S}_i\| \leq \frac{1}{\hat{n}} (\|\hat{\boldsymbol{r}}_i y\| + \|\mathbb{E}[\boldsymbol{r}y]\|) \leq \frac{1}{\hat{n}} (\|\hat{\boldsymbol{r}}_i\| |y| + \mathbb{E}[\|\boldsymbol{r}\| |y|]) \leq \frac{2}{\hat{n}} \sqrt{BC} := L. \tag{30}$$

Define $\boldsymbol{Z} = \sum_{i=1}^{\hat{n}} \boldsymbol{S}_i$. We analyze the semidefinite upper bounds for the variances $\mathbb{E}\boldsymbol{Z}\boldsymbol{Z}^\top$ and $\mathbb{E}\boldsymbol{Z}^\top\boldsymbol{Z}$:

$$\mathbb{E}\boldsymbol{Z}\boldsymbol{Z}^\top = \sum_{i=1}^{\hat{n}} \mathbb{E}\boldsymbol{S}_i\boldsymbol{S}_i^\top$$

$$= \frac{1}{\hat{n}^2} (\mathbb{E}y_i^2 \hat{\boldsymbol{r}}_i \hat{\boldsymbol{r}}_i^\top - \mathbb{E}[\boldsymbol{r}y]\mathbb{E}[\boldsymbol{r}y]^\top)$$

$$\preccurlyeq \frac{1}{\hat{n}^2} \mathbb{E}y_i^2 \hat{\boldsymbol{r}}_i \hat{\boldsymbol{r}}_i^\top$$

$$\preccurlyeq \frac{C}{\hat{n}^2} \boldsymbol{\Sigma} := \boldsymbol{V}_1.$$

$$\mathbb{E}\boldsymbol{Z}^\top\boldsymbol{Z} = \sum_{i=1}^{\hat{n}} \mathbb{E}\boldsymbol{S}_i^\top\boldsymbol{S}_i$$

$$= \hat{n}\mathbb{E}\|\boldsymbol{S}_i\|^2$$

$$\leq \frac{4}{\hat{n}}BC := \boldsymbol{V}_2 \qquad \text{by Equation 30.}$$

Define $v = \max(\|\boldsymbol{V}_1\|_{\mathrm{op}}, \|\boldsymbol{V}_2\|_{\mathrm{op}})$. It can be simplified as follows

$$v = \max(\|\frac{C}{\hat{n}}\boldsymbol{\Sigma}\|_{\mathrm{op}}, \frac{4}{\hat{n}}BC)$$

$$= \frac{4}{\hat{n}}BC \qquad \text{because } B \geq \|\boldsymbol{\Sigma}\|_{\mathrm{op}} \text{ as in Equation 27.}$$

Define $d = \textbf{intdim}(\begin{bmatrix} V_1 & 0 \\ 0 & V_2 \end{bmatrix})$, which can be simplified as

$$
\begin{aligned}
d =&\textbf{intdim}(\begin{bmatrix} \frac{C}{\hat{n}}\boldsymbol{\Sigma} & 0 \\ 0 & \frac{4}{\hat{n}}BC \end{bmatrix}) \\
=&\frac{\text{Tr}(\frac{C}{\hat{n}}\boldsymbol{\Sigma}) + \frac{4}{\hat{n}}BC}{\max(\|\frac{C}{\hat{n}}\boldsymbol{\Sigma}\|_{\text{op}}, \frac{4}{\hat{n}}BC)} \\
=&\frac{\text{Tr}(\frac{C}{\hat{n}}\boldsymbol{\Sigma}) + \frac{4}{\hat{n}}BC}{\frac{4}{\hat{n}}BC} \\
=&\frac{\text{Tr}(\frac{C}{\hat{n}}\boldsymbol{\Sigma})}{\frac{4}{\hat{n}}BC} + 1 \\
\leq& q/4 + 1 \qquad \text{because } B \geq \tau \text{ as in Equation 27 and } \frac{\text{Tr}(\boldsymbol{\Sigma})}{\tau} = q .
\end{aligned}
$$

Applying Theorem 7.3.1 of (Tropp et al., 2015), we have that for any $t \geq \sqrt{v} + L/3$,

$$
\begin{aligned}
\mathbb{P}\{\|\boldsymbol{Z}\| \geq t\} \leq& 4d \exp\left(\frac{-t^2/2}{v + Lt/3}\right) \\
\leq& (q+4) \exp\left(\frac{-t^2/2}{\frac{4}{\hat{n}}BC + \frac{2\sqrt{BC}}{\hat{n}}t/3}\right).
\end{aligned} \tag{31}
$$

By assumption:

$$
\begin{aligned}
& n^{1-c} = \omega(B \log q) \\
\Longrightarrow\ & n^{1-c} = \omega(B \log(q+4)) \\
\Longrightarrow\ & n^{1-c} = \omega\left(\left(4BC + \frac{2\sqrt{BC}}{3}\right)\log(q+4)\right) \qquad \text{because } C = \Theta(1), \text{ and } B = \Omega(1) \text{ as in Equation 27} \\
\Longrightarrow\ & \frac{0.5n^{1-c}}{\left(4BC + \frac{2\sqrt{BC}}{3}\right)} = \omega(\log(q+4)) \\
\Longrightarrow\ & (q+4)\exp\left(\frac{-0.5n^{1-c}}{4BC + \frac{2\sqrt{BC}}{3}}\right) = o(1).
\end{aligned}
$$

Therefore, we set the value of $t$ to $\hat{n}^{-0.5c} = o(1)$ in Equation 31. It is easy to verify that $\hat{n}^{-0.5c} \geq \sqrt{v} + L/3$. Substituting, we get:

$$
\begin{aligned}
\mathbb{P}\{\|\boldsymbol{Z}\|_{\text{op}} \geq \hat{n}^{-0.5c}\} \leq& (q+4)\exp\left(\frac{-0.5n^{-c}}{\frac{4}{\hat{n}}BC + \frac{2\sqrt{BC}}{\hat{n}}\hat{n}^{-0.5c}/3}\right) \\
\leq& (q+4)\exp\left(\frac{-0.5n^{-c}}{\frac{4}{\hat{n}}BC + \frac{2\sqrt{BC}}{\hat{n}}/3}\right) \qquad \text{because } \hat{n}^{-0.5c} \leq 1 \\
=& (q+4)\exp\left(\frac{-0.5n^{1-c}}{4BC + 2\sqrt{BC}/3}\right) \\
=& o(1).
\end{aligned}
$$

$\square$

Now, we are ready to show that Example 3.4 satisfies Definition 3.3. We let $\mathcal{V}$ be the entire representation space. Then, $\mathcal{V}^\perp$ is the zero space $\boldsymbol{0}$. In this case, the conditions **Kernel-wise $\delta$-isotropy on $\mathcal{V}^\perp$**, **Small cross-sample inner-product on $\mathcal{V}^\perp$**, and **Diminishing population covariance on $\mathcal{V}^\perp$** trivially hold. Thus, we only need to prove that **Boundedness** and **Concentration on $\mathcal{V}$** hold.

We let $\delta = n^{-0.1c}$ and $\gamma = 0$. First, note that $\delta^2 = n^{-0.2c} \geq \hat{n}^{-0.2c}$. Then, by Lemma C.1, we obtain that $\|\hat{\boldsymbol{\Sigma}} - \boldsymbol{\Sigma}\|_{\text{op}} \leq \hat{n}^{-0.5c} = o(\hat{n}^{-0.2c}) = o(\delta^2) = o(\gamma^2 + \delta^2 + \rho)$ with probability $1 - o(1)$. Similarly, we can show that $\|\tilde{\boldsymbol{\Sigma}} - \boldsymbol{\Sigma}\|_{\text{op}} = o(\gamma^2 + \delta^2 + \rho)$ with probability $1 - o(1)$.

Next, since $\delta = n^{-0.1c} \geq \hat{n}^{-0.1c}$, applying Lemma C.2 gives us $\left| \frac{1}{\hat{n}} \sum_{i=1}^{\hat{n}} \hat{\boldsymbol{r}}_i y_i - \mathbb{E}[\boldsymbol{r}y] \right| \leq \hat{n}^{-0.5c} = o(\hat{n}^{-0.1c}) = o(\delta) = o(\gamma + \delta + \rho)$ with probability $1 - o(1)$. Similarly, the same conclusion can be shown for $\frac{1}{\tilde{n}} \tilde{\boldsymbol{r}}_i y_i$.

Note that there are only four events above, so the probability that all of them occur remains $1 - o(1)$. To now, we have proved **Concentration on** $\mathcal{V}$.

Finally, regarding **Boundedness**, $\|\boldsymbol{\Sigma}\|_{\text{op}} = \Theta(1)$ is directly given in the assumption. Keeping in mind that $\mathcal{V}$ is the entire space, the conditions regarding covariance matrices are readily satisfied through the triangle inequality. For example: $\|\hat{\boldsymbol{\Sigma}}\|_{\text{op}} \leq \|\hat{\boldsymbol{\Sigma}} - \boldsymbol{\Sigma}\|_{\text{op}} + \|\boldsymbol{\Sigma}\|_{\text{op}} = o(1) + \Theta(1) = O(1)$. The other two conditions are directly implied by the boundedness of each $y$.

### C.2. Example 3.5

Originating from PCA (Johnstone, 2001), the spiked covariance model has been widely adopted in recent works to theoretically characterize key aspects across various topics (Ji et al., 2023; Nakada et al., 2023; Muthukumar et al., 2021; Pezeshki et al., 2022; Wu & Sahai, 2024). Furthermore, Example 3.5 also subsumes the sparse coding model as a special case, which has its roots in computer vision (Olshausen & Field, 1997; Foldiak, 2003; Olshausen & Field, 2004; Yang et al., 2009; Mairal et al., 2014; Papyan et al., 2017), has been used to model language data (Arora et al., 2018), and has been extensively employed in recent theoretical studies (Kalimeris et al., 2019; Allen-Zhu & Li, 2020; Wen & Li, 2021; Zou et al., 2021; Shen et al., 2022; Xue et al., 2023).

In the following proof, we start with a simple case where the data are Gaussian. We then extend the result to sub-Gaussian data by replacing the technical lemmas for Gaussian data with appropriate alternatives.

#### C.2.1. OVER-PARAMETERIZED GAUSSIAN DATA

Suppose that we have $\hat{\boldsymbol{R}} \in \mathbb{R}^{d \times \hat{n}}$, $\tilde{\boldsymbol{R}} \in \mathbb{R}^{d \times \tilde{n}}$ with $\hat{n} = \Theta(\tilde{n})$ and $d = \omega(\hat{n}^2)$ drawn from a high-dimensional $\boldsymbol{\Sigma}$-Gaussian ensemble with zero mean, where

$$\boldsymbol{\Sigma} = \begin{bmatrix} \boldsymbol{I}_k & \boldsymbol{0} \\ \boldsymbol{0} & \frac{\sigma^2}{d-k}\boldsymbol{I}_{d-k} \end{bmatrix} = \underbrace{\begin{bmatrix} \boldsymbol{I}_k = \boldsymbol{\Lambda}' & \boldsymbol{0} \\ \boldsymbol{0} & \boldsymbol{0} \end{bmatrix}}_{\boldsymbol{\Sigma}'} + \underbrace{\begin{bmatrix} \boldsymbol{0} & \boldsymbol{0} \\ \boldsymbol{0} & \frac{\sigma^2}{d-k}\boldsymbol{I}_{d-k} = \boldsymbol{\Lambda}'' \end{bmatrix}}_{\boldsymbol{\Sigma}''}, \quad \text{with } \sigma^2 = O(\hat{n}), \hat{n} = \omega(k^2). \tag{32}$$

Here the two data splits have comparable sizes, and the model is heavily over-parameterized. By splitting the matrix $\hat{\boldsymbol{R}} = \begin{bmatrix} \hat{\boldsymbol{F}} \\ \hat{\boldsymbol{A}} \end{bmatrix}$, where $\hat{\boldsymbol{F}} \in \mathbb{R}^{k \times \hat{n}}$ corresponds to the $k$ principal features (which form the space $\mathcal{V}$) and $\hat{\boldsymbol{A}} \in \mathbb{R}^{(d-k) \times \hat{n}}$ corresponds to the rest (which form the space $\mathcal{V}^\perp$), we can write the sample covariance matrix as

$$\hat{\boldsymbol{\Sigma}} = \frac{1}{\hat{n}} \hat{\boldsymbol{R}} \hat{\boldsymbol{R}}^\top = \frac{1}{\hat{n}} \begin{bmatrix} \hat{\boldsymbol{F}} \hat{\boldsymbol{F}}^\top & \hat{\boldsymbol{F}} \hat{\boldsymbol{A}}^\top \\ \hat{\boldsymbol{A}} \hat{\boldsymbol{F}}^\top & \hat{\boldsymbol{A}} \hat{\boldsymbol{A}}^\top \end{bmatrix}.$$

We note that $d - k = \omega(\hat{n}^2)$, and the corresponding labels have bounded mean and variance. The same decomposition applies to $\tilde{\boldsymbol{R}}$. Note that here $\boldsymbol{U}' = \begin{bmatrix} \boldsymbol{I}_k \\ \boldsymbol{0}_{(d-k) \times k} \end{bmatrix}$ and $\boldsymbol{U}'' = \begin{bmatrix} \boldsymbol{0}_{k \times (d-k)} \\ \boldsymbol{I}_{d-k} \end{bmatrix}$ allow us to define the projection matrices $\boldsymbol{U}' \boldsymbol{U}'^\top$ and $\boldsymbol{U}'' \boldsymbol{U}''^\top$ on $\mathcal{V}$ and $\mathcal{V}^\perp$ respectively.

In this section, we show that our assumptions hold in the above setting with $\delta = 0$ and $\hat{\gamma} = \sigma^2/\hat{n}, \tilde{\gamma} = \sigma^2/\tilde{n}$. We only prove for $\hat{\boldsymbol{R}}$ whenever the same proof can be easily applied to $\tilde{\boldsymbol{R}}$.

First, let us introduce the following Lemmas:

**Lemma C.3** (Restatement of Example 6.2 in (Wainwright, 2019)). *Let $\mathbf{X} \in \mathbb{R}^{d \times n}$ be a random matrix with i.i.d. entries drawn from $\mathcal{N}(0, 1)$ (that is a $\boldsymbol{\Sigma}$-Gaussian ensemble with $\boldsymbol{\Sigma} = \boldsymbol{I}_d$). Then with probability at least $1 - 2e^{-n\delta^2/2}$ for some*

$\delta > 0$, the following inequality holds:

$$\|\frac{1}{n}\boldsymbol{X}\boldsymbol{X}^T - \boldsymbol{I}_d\|_{\mathrm{op}} \le 2\left(\sqrt{\frac{d}{n}} + \delta\right) + \left(\sqrt{\frac{d}{n}} + \delta\right)^2.$$

**Lemma C.4.** *Consider two independently sampled Gaussian matrices where $\mathbf{A} \in \mathbb{R}^{d_1 \times n}$ has columns $\boldsymbol{a}_i \sim \mathcal{N}(0, \sigma_1^2 \boldsymbol{I}_{d_1})$ and $\mathbf{B} \in \mathbb{R}^{d_2 \times n}$ has columns $\boldsymbol{b}_i \sim \mathcal{N}(0, \sigma_2^2 \boldsymbol{I}_{d_2})$. Then for some $\frac{1}{d_1 d_2} > \delta > 0$ and constant $C$, with probability at least $1 - d_1 d_2 \delta$, we have*

$$\frac{1}{n}\|\boldsymbol{A}\boldsymbol{B}^\top\|_{\mathrm{op}} \le \frac{\sigma_1 \sigma_2}{n}\sqrt{Cd_1 d_2 n \log(\frac{2}{\delta})}.$$

*Proof.* Let $\boldsymbol{Q} = \boldsymbol{A}\boldsymbol{B}^T$. Then each entry of $\boldsymbol{Q}$ is an inner product $\boldsymbol{Q}_{ij} = \boldsymbol{a}_i \cdot \boldsymbol{b}_j$, where $\boldsymbol{a}_i \in \mathbb{R}^n$ is the $i$-th row of $\boldsymbol{A}$ and $\boldsymbol{b}_j \in \mathbb{R}^n$ is the $j$-th row of $\boldsymbol{B}$. Since each entry of $\boldsymbol{a}_i$ is $\mathcal{N}(0, \sigma_1^2)$ and each entry of $\boldsymbol{b}_j$ is $\mathcal{N}(0, \sigma_2^2)$, by Lemma 4 from (Shen et al., 2022), with probability at least $1 - \delta$ (taking $\frac{1}{d_1 d_2} > \delta > 0$), for some constant $C_{ij}$,

$$\boldsymbol{Q}_{ij}^2 = (\boldsymbol{a}_i \cdot \boldsymbol{b}_j)^2 \le C_{ij}\sigma_1^2\sigma_2^2 n \log(2/\delta').$$

We define $C = \max\{C_{ij} : 1 \le i \le d_1, 1 \le j \le d_2\}$. Now we bound the operator norm with

$$\begin{aligned}
\frac{1}{n}\|\boldsymbol{A}\boldsymbol{B}^\top\|_{\mathrm{op}} &\le \frac{1}{n}\|\boldsymbol{A}\boldsymbol{B}^\top\|_F = \frac{1}{n}\|\boldsymbol{Q}\|_F \\
&= \frac{1}{n}\sqrt{\sum_{1 \le i \le d_1, 1 \le j \le d_2} \boldsymbol{Q}_{ij}^2} \\
&\le \frac{1}{n}\sqrt{\sum_{1 \le i \le d_1, 1 \le j \le d_2} C_{ij}\sigma_1^2\sigma_2^2 n \log(2/\delta)} \\
&\le \frac{1}{n}\sqrt{Cd_1 d_2 \sigma_1^2\sigma_2^2 n \log(2/\delta)} = \frac{\sigma_1 \sigma_2}{n}\sqrt{Cd_1 d_2 n \log(2/\delta)}
\end{aligned}$$

with probability at least $1 - d_1 d_2 \delta$ since the inequality has to hold for each entry. $\square$

We now prove that the example satisfies the five aspects of the definition:

1. **Boundedness**:

   First, we have $\|\boldsymbol{\Sigma}\|_{\mathrm{op}} = 1 = O(1)$ from its definition, and

$$\|\hat{\boldsymbol{\Sigma}} - \boldsymbol{\Sigma}\|_{\mathrm{op}} \le \left\|\begin{bmatrix} \frac{1}{\hat{n}}\hat{\boldsymbol{F}}\hat{\boldsymbol{F}}^\top - \boldsymbol{I}_k & \boldsymbol{0} \\ \boldsymbol{0} & \frac{1}{\hat{n}}\hat{\boldsymbol{A}}\hat{\boldsymbol{A}}^\top - \frac{\sigma^2}{d-k}\boldsymbol{I}_{d-k} \end{bmatrix}\right\|_{\mathrm{op}} + \frac{1}{\hat{n}}\left\|\begin{bmatrix} \boldsymbol{0} & \hat{\boldsymbol{F}}\hat{\boldsymbol{A}}^\top \\ \hat{\boldsymbol{A}}\hat{\boldsymbol{F}}^\top & \boldsymbol{0} \end{bmatrix}\right\|_{\mathrm{op}}. \tag{33}$$

   By Lemma C.3, we take $\delta_1 = \hat{n}^{-1/4}$ and have that with probability at least $1 - 2e^{-\hat{n}\delta_1^2/2} = 1 - 2e^{-\sqrt{\hat{n}}/2} = 1 - o(1)$,

$$\|\frac{1}{\hat{n}}\hat{\boldsymbol{F}}\hat{\boldsymbol{F}}^\top - \boldsymbol{I}_k\|_{\mathrm{op}} \le 2\sqrt{\frac{k}{\hat{n}}} + \frac{2}{\hat{n}^{1/4}} + \left(\sqrt{\frac{k}{\hat{n}}} + \frac{1}{\hat{n}^{1/4}}\right)^2 = o(1) \quad \text{since } \hat{n} \gg k.$$

   As $\hat{\boldsymbol{A}} \in \mathbb{R}^{(d-k) \times \hat{n}}$ is sampled from $\frac{\sigma^2}{d-k}\boldsymbol{I}_{d-k}$, $\frac{\sqrt{d-k}}{\sigma}\hat{\boldsymbol{A}}$ is sampled from $\boldsymbol{I}_{d-k}$. With this scaling, similarly, Lemma C.3 implies that

$$\begin{aligned}
\left\|\frac{1}{\hat{n}}\left(\frac{\sqrt{d-k}}{\sigma}\hat{\boldsymbol{A}}\right)\left(\frac{\sqrt{d-k}}{\sigma}\hat{\boldsymbol{A}}\right)^\top - \boldsymbol{I}_{d-k}\right\|_{\mathrm{op}} &= \left\|\frac{d-k}{\hat{n}\sigma^2}\hat{\boldsymbol{A}}\hat{\boldsymbol{A}}^\top - \boldsymbol{I}_{d-k}\right\|_{\mathrm{op}} \\
&\le 2\sqrt{\frac{d-k}{\hat{n}}} + \frac{2}{\hat{n}^{1/4}} + \left(\sqrt{\frac{d-k}{\hat{n}}} + \frac{1}{\hat{n}^{1/4}}\right)^2
\end{aligned}$$

$$\iff \left\|\frac{1}{\hat{n}}\hat{A}\hat{A}^\top - \frac{\sigma^2}{d-k}I_{d-k}\right\|_{\text{op}} \le \frac{\sigma^2}{d-k}\left[2\sqrt{\frac{d-k}{\hat{n}}} + \frac{2}{\hat{n}^{1/4}} + \left(\sqrt{\frac{d-k}{\hat{n}}} + \frac{1}{\hat{n}^{1/4}}\right)^2\right] = O(1) \quad \text{as } \sigma^2 = O(\hat{n}).$$

We have bounded the first term on the right side of Eq. 33 and have that $\|\frac{1}{\sqrt{\hat{n}}}\hat{F}\|_{\text{op}}$ and $\|\frac{1}{\sqrt{\hat{n}}}\hat{A}\|_{\text{op}}$ are $O(1)$. It follows that

$$\frac{1}{\hat{n}}\left\|\begin{bmatrix} \mathbf{0} & \hat{F}\hat{A}^\top \\ \hat{A}\hat{F}^\top & \mathbf{0} \end{bmatrix}\right\|_{\text{op}} = \frac{1}{\hat{n}}\|\hat{F}\hat{A}^\top\|_{\text{op}} = O(1) \implies \|\hat{\Sigma} - \Sigma\|_{\text{op}} = O(1).$$

Hence, $\|\hat{\Sigma}\|_{\text{op}} = O(1)$ directly follows from $\|\Sigma\|_{\text{op}} = O(1)$.

Now we consider $\frac{1}{\hat{n}}\|\hat{y}\|^2 = \frac{1}{\hat{n}}\sum_{i=0}^{\hat{n}}\hat{y}_i^2$, where $\hat{y}_i$ represents the $i$-th entry of the vector. Since the label has bounded population variance $O(1)$, the i.i.d assumption implies

$$\text{Var}\left(\frac{1}{\hat{n}}\sum_{i=0}^{\hat{n}}\hat{y}_i^2\right) = \frac{1}{\hat{n}^2}\sum_{i=0}^{\hat{n}}\text{Var}(\hat{y}_i^2) = \frac{1}{\hat{n}^2}\sum_{i=0}^{\hat{n}}O(1) = O\left(\frac{1}{\hat{n}}\right).$$

Then by Chebyshev's inequality, for any $\epsilon > 0$ and some constant $C_1$, we let $z = \frac{1}{\hat{n}}\|\hat{y}\|^2$ for simplicity and then have

$$P\left(|z - \mathbb{E}[z]| > \epsilon\right) \le \frac{\text{Var}(z)}{\epsilon^2} \le \frac{C_1}{\hat{n}\epsilon^2}.$$

We take $\epsilon = \hat{n}^{-1/4}$. Then with probability at least $1 - \frac{C_1}{\sqrt{\hat{n}}} = 1 - o(1)$,

$$\left|\frac{1}{\hat{n}}\|\hat{y}\|^2 - \text{Var}(\boldsymbol{y}_i)\right| = o(1) \implies \frac{1}{\hat{n}}\|\hat{y}\|^2 = O(1) \quad \text{since the variance of the label is bounded.}$$

2. **Concentration on $\mathcal{V}$:**

With $\boldsymbol{U}' = \begin{bmatrix} \boldsymbol{I}_k \\ \mathbf{0}_{(d-k)\times k} \end{bmatrix}$ preserving only the first $k$ components, we have from above that with probability at least $1 - o(1)$,

$$\|\boldsymbol{U}'^\top\hat{\Sigma}\boldsymbol{U}' - \Lambda'\|_{\text{op}} = \|\frac{1}{\hat{n}}\hat{F}\hat{F}^\top - \boldsymbol{I}_k\|_{\text{op}} = o(1).$$

Now we consider

$$\|\frac{1}{\hat{n}}\boldsymbol{U}'^\top\hat{R}\hat{y} - \mathbb{E}[\boldsymbol{U}'^\top ry]\| = \|\frac{1}{\hat{n}}\hat{F}\hat{y} - \mathbb{E}[fy]\|,$$

where $\boldsymbol{f} = \boldsymbol{U}'r$. We define a new random variable $\boldsymbol{z} = \boldsymbol{f}y$ and its sample mean $\hat{\boldsymbol{Z}} = \frac{1}{\hat{n}}\hat{F}\hat{y} \in \mathbb{R}^k$. We first show that the variance of each entry of $\hat{\boldsymbol{Z}}$ is of magnitude $\sim \frac{1}{\hat{n}}$:

$$\text{Var}(\hat{\boldsymbol{Z}}_i) = \text{Var}\left(\sum_{j=1}^{\hat{n}}\frac{1}{\hat{n}}\hat{F}_{ij}\hat{y}_j\right) = \frac{1}{\hat{n}^2}\text{Var}\left(\sum_{j=1}^{\hat{n}}\hat{F}_{ij}\hat{y}_j\right) \quad \forall i = 1, \cdots, k.$$

For each term in the summation,

$$\text{Var}(\hat{F}_{ij}\hat{y}_j) = \mathbb{E}[(\hat{F}_{ij}\hat{y}_j)^2] - \mathbb{E}[\hat{F}_{ij}\hat{y}_j]^2 = O(1)$$

since $\hat{F}_{ij}$ and $\hat{y}_j$ are both bounded. By the i.i.d assumption,

$$\text{Var}(\hat{\boldsymbol{Z}}_i) = \frac{1}{\hat{n}^2}\sum_{j=1}^{\hat{n}}O(1) = O\left(\frac{1}{\hat{n}}\right).$$

By Chebyshev's inequality, for any $\epsilon > 0$ and some constant $C_2$,

$$P\left(\|\hat{\boldsymbol{Z}}_i - \mathbb{E}[\boldsymbol{z}_i]\| > \epsilon\right) \le \frac{\text{Var}(\hat{\boldsymbol{Z}}_i)}{\epsilon^2} \le \frac{C_2}{\hat{n}\epsilon^2}$$

$$P\left(\|\hat{Z}_i - \mathbb{E}[z_i]\| > \epsilon \quad \forall i = 1, \cdots, k\right) \leq \frac{kC_2}{\hat{n}\epsilon^2}$$

Similarly, by choosing $\epsilon = \hat{n}^{-1/4}$, the probability of large deviation decays rapidly as:

$$P\left(\|\hat{Z}_i - \mathbb{E}[z_i]\| > \frac{1}{\hat{n}^{1/4}} \quad \forall i = 1, \cdots, k\right) \leq \frac{kC_2}{\sqrt{\hat{n}}} = o(1) \quad \text{since } \hat{n} = \omega(k^2).$$

This statement implies that with probability at least $1 - o(1)$,

$$\|\hat{Z} - \mathbb{E}[z]\| = \|\frac{1}{\hat{n}}U'^\top \hat{R}\hat{y} - \mathbb{E}[U'^\top ry]\| \leq \sqrt{\frac{k}{\sqrt{\hat{n}}}} = o(1) = o(\gamma + \delta + \lambda_{\min}(\mathbf{\Lambda}'))$$

as we sum up the $k$ terms. This shows that our setting satisfies the second part of the definition.

3. **Kernel-wise $\delta$-isotropy on $\mathcal{V}^\perp$ :**

We define $\mathbf{Z} = \frac{\sqrt{d-k}}{\sigma}\hat{A} \in \mathbb{R}^{(d-k)\times\hat{n}}$, which has standard normal entries. With the scaling, we plug in $\mathbf{U''}$, $\hat{\gamma} = \sigma^2/\hat{n}$ and have

$$\|\frac{1}{\hat{n}}\hat{R}^\top U''U''^\top \hat{R} - \hat{\gamma}I\|_{\text{op}} = \|\frac{1}{\hat{n}}\hat{A}^\top \hat{A} - \frac{\sigma^2}{\hat{n}}I\|_{\text{op}} = \frac{\sigma^2}{\hat{n}}\|\frac{1}{d-k}Z^\top Z - I\|_{\text{op}}. \tag{34}$$

Now we apply Lemma C.3 and have that with probability at least $1 - 2e^{-\hat{n}\delta_2^2/2}$ for some $\delta_2 > 0$,

$$\frac{\sigma^2}{\hat{n}}\|\frac{1}{d-k}Z^\top Z - I\|_{\text{op}} \leq \frac{\sigma^2}{\hat{n}}\left[2\left(\sqrt{\frac{\hat{n}}{d-k}} + \delta_2\right) + \left(\sqrt{\frac{\hat{n}}{d-k}} + \delta_2\right)^2\right].$$

The rest follows similarly by taking $\delta_2 = \hat{n}^{-1/4}$.

4. **Small cross-sample inner-product on $\mathcal{V}^\perp$:**

By $\mathbf{U''} = \begin{bmatrix} \mathbf{0}_{k\times(d-k)} \\ \mathbf{I}_{d-k} \end{bmatrix}$ and Lemma C.4 with $\hat{A}^\top \in \mathbb{R}^{\hat{n}\times(d-k)}$ and $\tilde{A}^\top \in \mathbb{R}^{\tilde{n}\times(d-k)}$, each having $\mathcal{N}(0, \frac{\sigma^2}{d-k})$ entries, the target expression becomes

$$\|\frac{1}{\sqrt{\hat{n}}}\hat{R}^\top U''U''^\top \frac{1}{\sqrt{\tilde{n}}}\tilde{R}\|_{\text{op}} = \frac{1}{\sqrt{\hat{n}\tilde{n}}}\|\hat{A}^\top \tilde{A}\|_{\text{op}} \tag{35}$$

$$\leq \frac{1}{\sqrt{\hat{n}\tilde{n}}}\sqrt{C_4\hat{n}\tilde{n}\frac{\sigma^4}{(d-k)^2}(d-k)\log(2/\delta_3)}$$

$$= \sqrt{C_4\frac{\sigma^4}{d-k}\log(2/\delta_3)}$$

$$= \sigma^2\sqrt{C_4\log(2/\delta_3)}\sqrt{\frac{1}{d-k}}$$

for some constant $C_4$ and with probability at least $1 - \hat{n}\tilde{n}\delta_3$ for some $0 < \delta_3 < \frac{1}{\hat{n}\tilde{n}}$. We choose some $\delta_3 = o(\frac{1}{\hat{n}\tilde{n}})$ in this range and then have that with probability at least $1 - o(1)$, the previous bound can be expressed as:

$$\sigma^2\sqrt{C_4\log(2/\delta_3)}\sqrt{\frac{1}{d-k}} = \Theta\left(\sigma^2\sqrt{C_4\frac{\log(\hat{n}\tilde{n})}{d-k}}\right) = o(\frac{\sigma^2}{\max\{\hat{n},\tilde{n}\}}) = o(\gamma + \delta)$$

since $d - k = \omega(\hat{n}) = \omega(\tilde{n})$.

5. **Diminishing population covariance on $\mathcal{V}^\perp$:**

By definition, it is trivial to see that

$$\lambda_{max}(\mathbf{\Lambda''}) = \frac{\sigma^2}{d-k} = o(\frac{\sigma^2}{\max\{\hat{n},\tilde{n}\}}) = o(\gamma + \delta)$$

since $d - k = \omega(\hat{n}) = \omega(\tilde{n})$.

### C.2.2. FURTHER RELAXATION TO SUB-GAUSSIAN DATA

Now, we consider the more general sub-Gaussian setting outlined in Example 3.5. The population covariance is:

$$\boldsymbol{\Sigma} = \begin{bmatrix} \boldsymbol{I}_k & \boldsymbol{0} \\ \boldsymbol{0} & \frac{\sigma^2}{d-k}\boldsymbol{I}_{d-k} \end{bmatrix},$$

where the top left block has a corresponding sub-Gaussian parameter of $\Theta(1)$ and the rest has a parameter of $\Theta(\frac{\sigma^2}{d-k})$.

We adopt the following definitions from Chapter 2 of (Vershynin, 2018) for reference.

**Definition C.5.** A zero-mean random variable $X$ is sub-Gaussian if there is a positive parameter $K_g$ such that

$$\mathbb{E}[e^{X^2/K_g^2}] \leq 2.$$

**Definition C.6.** A zero-mean random variable $X$ is sub-exponential if there is a positive parameter $K_e$ such that

$$\mathbb{E}[e^{|X|/K_e}] \leq 2.$$

We can also define the following norms that give the sub-Gaussian or sub-exponential parameter:

$$\|X\|_{\psi_2} = \inf\{t > 0 : \mathbb{E}[e^{X^2/t^2}] \leq 2\} = K_g$$

$$\|X\|_{\psi_1} = \inf\{t > 0 : \mathbb{E}[e^{|X|/t}] \leq 2\} = K_e$$

*Remark.* There are many different characterizations for these two definitions, each with a different sub-Gaussian/sub-exponential parameter. A detailed summary can be found in Chapter 2 of (Vershynin, 2018). Notably, these parameters differ from each other only by at most a constant factor.

**Lemma C.7.** *(Extension of Lemma 4 (Shen et al., 2022) to sub-Gaussian)* *Consider high-dimensional independent sub-Gaussian vectors $\boldsymbol{z}_1, \boldsymbol{z}_2 \in \mathbb{R}^d$, whose i.i.d. entries have variances $\sigma_1^2$, $\sigma_2^2$ and sub-Gaussian parameters $\Theta(\sigma_1)$, $\Theta(\sigma_2)$ respectively. Then for $\delta > 0$ such that $\sqrt{\log(2/\delta)} > \sqrt{cd}$ for some constant c, there exists a constant C such that with probability at least $1 - \delta$,*

$$|\boldsymbol{z}_1 \cdot \boldsymbol{z}_2| \leq C\sigma_1\sigma_2\sqrt{d\log(2/\delta)}.$$

*Proof.* We consider the product $\boldsymbol{z}_1 \cdot \boldsymbol{z}_2 = \sum_{i=1}^d z_{1i}z_{2i} = \sum_{i=1}^d a_i$, where we define $a_i$ for simplicity. It is a well-known result that the product of two sub-Gaussian random variables is sub-exponential. More precisely,

$$\|a_i\|_{\psi_1} \leq \|\boldsymbol{z}_{1i}\|_{\psi_2}\|\boldsymbol{z}_{2i}\|_{\psi_2} = C\sigma_1\sigma_2.$$

By Bernstein's inequality for sub-exponential functions (see Theorem 3.8.1 (Vershynin, 2018)), this summation can be bounded as: for some constant $c > 0$,

$$P\left(\left|\sum_{i=1}^d a_i\right| \geq t\right) \leq 2\exp\left[-c\min\left\{\frac{t^2}{\sum_{i=1}^d \|a_i\|_{\psi_1}^2}, \frac{t}{\max_i \|a_i\|_{\psi_1}}\right\}\right]$$

$$\leq 2\exp\left[-c\min\left\{\frac{t^2}{dC^2\sigma_1^2\sigma_2^2}, \frac{t}{C\sigma_1\sigma_2}\right\}\right]$$

Let $t = \frac{C}{\sqrt{c}}\sigma_1\sigma_2\sqrt{d\log(2/\delta)}$ for some $\delta$ that satisfies the condition $\sqrt{\log(2/\delta)} > \sqrt{cd}$ (e.g. $\delta = 1/d^2$). The probability statement becomes:

$$P\left(\left|\sum_{i=1}^d a_i\right| \geq \frac{C}{\sqrt{c}}\sigma_1\sigma_2\sqrt{d\log(2/\delta)}\right) \leq 2\exp\left[-c\min\left\{\frac{\log(2/\delta)}{c}, \sqrt{\frac{d\log(2/\delta)}{c}}\right\}\right]$$

$$= 2\exp\left[-\min\left\{\log(2/\delta), \sqrt{cd\log(2/\delta)}\right\}\right].$$

Since our choice of $\delta$ ensures that the first quantity is smaller,

$$P\left(\left|\sum_{i=1}^{d} a_i\right| \geq \frac{C}{\sqrt{c}}\sigma_1\sigma_2\sqrt{d\log(2/\delta)}\right) \leq \delta$$

In other words, letting $C' = C/\sqrt{c}$, we have that with probability at least $1 - \delta$,

$$|\boldsymbol{z}_1 \cdot \boldsymbol{z}_2| \leq C'\sigma_1\sigma_2\sqrt{d\log(2/\delta)}.$$

$\square$

Now we are ready to show that our assumptions capture the setting in Section C.2.1 but with sub-Gaussian data. That is, we now allow the data to have possibly even lighter tail than that of Gaussian. The proof can be easily replicated, as Chebyshev's inequality still applies here and Lemmas C.3, C.4 find the following "sub-Gaussian" alternatives, namely Lemmas C.8, C.9:

**Lemma C.8.** *(Restatement of Theorem 6.5 in (Wainwright, 2019)) Let* $\mathbf{X} \in \mathbb{R}^{d\times n}$ *be a random sub-Gaussian matrix with parameter* $K_g$ *and population covariance* $\boldsymbol{I}_d$. *Then for all* $\delta \geq 0$, *there are universal constants* $C_1, C_2, C_3$ *such that*

$$\|\frac{1}{n}\boldsymbol{X}\boldsymbol{X}^T - \boldsymbol{I}_d\|_{\mathrm{op}} \leq K_g^2\left[C_1\left(\sqrt{\frac{d}{n}} + \frac{d}{n}\right) + \delta\right]$$

*with probability at least* $1 - C_2 e^{-C_3 n \min\{\delta,\delta^2\}}$.

**Lemma C.9.** *Consider two independently sampled row-wise sub-Gaussian matrices* $\mathbf{A} \in \mathbb{R}^{d_1\times n}$, $\mathbf{B} \in \mathbb{R}^{d_2\times n}$ *that have i.i.d. entries with variances* $\sigma_1^2, \sigma_2^2$ *respectively. Then for some* $\frac{1}{d_1 d_2} > \delta > 0$ *and constant C, with probability at least* $1 - d_1 d_2 \delta$, *we have*

$$\frac{1}{n}\|\boldsymbol{A}\boldsymbol{B}^\top\|_{\mathrm{op}} \leq \frac{\sigma_1\sigma_2}{n}\sqrt{Cd_1 d_2 n \log(\frac{2}{\delta})}.$$

*Proof.* The proof is the same as Lemma C.4 except that we now use Lemma C.7 to bound the squared value of each entry in the Frobenius norm. $\square$

With these alternative extended results, the proof in Section C.2.1 immediately generalizes to sub-Gaussian data. This extension potentially allows us to accommodate more realistic scenario and enhances the theoretical robustness of our assumptions. Sub-Gaussian distributions capture a wider class of data behaviors; for instance, the fact that bounded random variables are sub-Gaussian makes the theory more applicable to many real-world datasets, which naturally exhibit sub-Gaussian characteristics. In the following section, we show a general result that even more examples can be constructed.

### C.3. Proof of Theorem 3.6

The intuition behind this theorem is that adding high-dimensional sub-Gaussian entries to the given representation preserves decomposbility while slightly modifying the parameters. Due to the orthogonality of $\boldsymbol{M}$ and $\boldsymbol{M}^\perp$, we let $\boldsymbol{U} = \begin{bmatrix} \boldsymbol{M} & \boldsymbol{M}^\perp \end{bmatrix}$ and then $\alpha(\boldsymbol{x}) = \boldsymbol{U}\begin{bmatrix} h(\boldsymbol{x}) \\ \xi(\boldsymbol{x}) \end{bmatrix}$; naturally, the column space of $\boldsymbol{M}$ can be regarded as the subspace $\mathcal{V}$, and the column space of $\boldsymbol{M}^\perp$ is $\mathcal{V}^\perp$. Given that $h(\boldsymbol{x})$'s representations are $(\delta, 0, 0)$-decomposable w.r.t. $\mathbb{R}^d$, we now prove that the new representations are $(\delta, \frac{\sigma^2}{\hat{n}}, \frac{\sigma^2}{\tilde{n}})$-decomposable. Again we only present the proof for one data split whenever it can be replicated for the other.

For notation, we let $\gamma = \sigma^2 / \max\{\hat{n}, \tilde{n}\}$.

1. **Boundedness:** $\frac{1}{\hat{n}}\sum_{i=1}^{\hat{n}} \hat{y}_i^2 = O(1)$ follows from the previous proof using Chebyshev's inequality. For the population

covariance,

$$\|\boldsymbol{\Sigma}(\alpha)\|_{\text{op}} = \|\mathbb{E}_{\mathcal{D}_{\boldsymbol{x}}}[\alpha(\boldsymbol{x})\alpha(\boldsymbol{x})^\top]\|_{\text{op}} = \left\|\mathbb{E}_{\mathcal{D}_{\boldsymbol{x}}} \begin{bmatrix} h(\boldsymbol{x})h(\boldsymbol{x})^\top & h(\boldsymbol{x})\xi(\boldsymbol{x})^\top \\ \xi(\boldsymbol{x})h(\boldsymbol{x})^\top & \xi(\boldsymbol{x})\xi(\boldsymbol{x})^\top \end{bmatrix}\right\|_{op}$$

$$\leq \left\|\mathbb{E}_{\mathcal{D}_{\boldsymbol{x}}} \begin{bmatrix} h(\boldsymbol{x})h(\boldsymbol{x})^\top & \mathbf{0} \\ \mathbf{0} & \xi(\boldsymbol{x})\xi(\boldsymbol{x})^\top \end{bmatrix}\right\|_{op} + \left\|\mathbb{E}_{\mathcal{D}_{\boldsymbol{x}}} \begin{bmatrix} \mathbf{0} & h(\boldsymbol{x})\xi(\boldsymbol{x})^\top \\ \xi(\boldsymbol{x})h(\boldsymbol{x})^\top & \mathbf{0} \end{bmatrix}\right\|_{op} \tag{36}$$

We have that $\|\mathbb{E}_{\mathcal{D}_{\boldsymbol{x}}}[h(\boldsymbol{x})h(\boldsymbol{x})^\top]\|_{\text{op}} = \|\boldsymbol{\Sigma}(h)\|_{\text{op}} = O(1)$ by the $(\delta, 0, 0)$-decomposibility assumption on $h$'s representations. From the proof for sub-Gaussian data in Section C.2.1, $\|\mathbb{E}_{\mathcal{D}_{\boldsymbol{x}}}[\xi(\boldsymbol{x})\xi(\boldsymbol{x})^\top]\|_{\text{op}} = \|\boldsymbol{\Sigma}(\xi)\|_{\text{op}} = O(1)$. These bound the first term on the RHS of Equation 36.

By the definition of operator norm,

$$\|\mathbb{E}_{\mathcal{D}_{\boldsymbol{x}}}[h(\boldsymbol{x})\xi(\boldsymbol{x})^\top]\|_{\text{op}} = \sup_{\|\boldsymbol{u}\|=1} \sup_{\|\boldsymbol{v}\|=1} \boldsymbol{u}^T \mathbb{E}_{\mathcal{D}_{\boldsymbol{x}}}[h(\boldsymbol{x})\xi(\boldsymbol{x})^\top]\boldsymbol{v} = \sup_{\|\boldsymbol{u}\|=1} \sup_{\|\boldsymbol{v}\|=1} \mathbb{E}_{\mathcal{D}_{\boldsymbol{x}}}[(\boldsymbol{u}^T h(\boldsymbol{x}))(\boldsymbol{v}^T \xi(\boldsymbol{x}))]. \tag{37}$$

By Cauchy-Schwartz inequality, we can bound this expectation as:

$$\mathbb{E}_{\mathcal{D}_{\boldsymbol{x}}}[(\boldsymbol{u}^T h(\boldsymbol{x}))(\boldsymbol{v}^T \xi(\boldsymbol{x}))] \leq \sqrt{\mathbb{E}_{\mathcal{D}_{\boldsymbol{x}}}[(\boldsymbol{u}^T h(\boldsymbol{x}))^2]}\sqrt{\mathbb{E}_{\mathcal{D}_{\boldsymbol{x}}}[(\boldsymbol{v}^T \xi(\boldsymbol{x}))^2]}, \text{ where}$$

$$\mathbb{E}_{\mathcal{D}_{\boldsymbol{x}}}[(\boldsymbol{u}^T h(\boldsymbol{x}))^2] = \mathbb{E}_{\mathcal{D}_{\boldsymbol{x}}}[\boldsymbol{u}^T h(\boldsymbol{x})h(\boldsymbol{x})^\top \boldsymbol{u}] = \boldsymbol{u}^T \mathbb{E}_{\mathcal{D}_{\boldsymbol{x}}}[h(\boldsymbol{x})h(\boldsymbol{x})^\top]\boldsymbol{u} \leq \|\boldsymbol{u}\|^2 \|\boldsymbol{\Sigma}(h)\|_{\text{op}} = O(1),$$

$$\mathbb{E}_{\mathcal{D}_{\boldsymbol{x}}}[(\boldsymbol{v}^T \xi(\boldsymbol{x}))^2] = \mathbb{E}_{\mathcal{D}_{\boldsymbol{x}}}[\boldsymbol{v}^T \xi(\boldsymbol{x})\xi(\boldsymbol{x})^\top \boldsymbol{v}] = \boldsymbol{v}^T \mathbb{E}_{\mathcal{D}_{\boldsymbol{x}}}[\xi(\boldsymbol{x})\xi(\boldsymbol{x})^\top]\boldsymbol{v} \leq \|\boldsymbol{v}\|^2 \|\boldsymbol{\Sigma}(\xi)\|_{\text{op}} = O(1).$$

Combing these results, we have that Equation 37 $= \|\mathbb{E}_{\mathcal{D}_{\boldsymbol{x}}}[h(\boldsymbol{x})\xi(\boldsymbol{x})^\top]\|_{\text{op}} = O(1)$, bounding the second term in Equation 36. Hence, $\|\boldsymbol{\Sigma}(\alpha)\|_{\text{op}} = O(1)$.

Simiarly, we can prove for the empirical covariance:

$$\|\hat{\boldsymbol{\Sigma}}(\alpha)\|_{\text{op}} = \|\frac{1}{\hat{n}}\sum_{i=1}^{\hat{n}} \alpha(\hat{\boldsymbol{x}}_i)\alpha(\hat{\boldsymbol{x}}_i)^\top\|_{\text{op}} = \left\|\frac{1}{\hat{n}} \begin{bmatrix} \sum_{i=1}^{\hat{n}} h(\hat{\boldsymbol{x}}_i)h(\hat{\boldsymbol{x}}_i)^\top & \sum_{i=1}^{\hat{n}} h(\hat{\boldsymbol{x}}_i)\xi(\hat{\boldsymbol{x}}_i)^\top \\ \sum_{i=1}^{\hat{n}} \xi(\hat{\boldsymbol{x}}_i)h(\hat{\boldsymbol{x}}_i)^\top & \sum_{i=1}^{\hat{n}} \xi(\hat{\boldsymbol{x}}_i)\xi(\hat{\boldsymbol{x}}_i)^\top \end{bmatrix}\right\|_{op}$$

$$= \left\|\frac{1}{\hat{n}} \begin{bmatrix} \hat{\boldsymbol{H}}\hat{\boldsymbol{H}}^\top & \hat{\boldsymbol{H}}\hat{\boldsymbol{\Xi}}^\top \\ \hat{\boldsymbol{\Xi}}\hat{\boldsymbol{H}}^\top & \hat{\boldsymbol{\Xi}}\hat{\boldsymbol{\Xi}}^\top \end{bmatrix}\right\|_{op},$$

where the $i$-th column of $\hat{\boldsymbol{\Xi}}$ is $\xi(\hat{\boldsymbol{x}}_i)$ and the $i$-th column of $\boldsymbol{H}$ is $h(\hat{\boldsymbol{x}}_i)$.

The rest is straightforward: the assumption on $h$ and the existing proof for sub-Gaussian data imply $\|\frac{1}{\hat{n}}\hat{\boldsymbol{H}}\hat{\boldsymbol{H}}^\top\|_{\text{op}} = O(1)$ and $\|\frac{1}{\hat{n}}\hat{\boldsymbol{\Xi}}\hat{\boldsymbol{\Xi}}^\top\|_{\text{op}} = O(1)$. Hence, $\|\frac{1}{\sqrt{\hat{n}}}\hat{\boldsymbol{H}}\|_{\text{op}}$ and $\|\frac{1}{\sqrt{\hat{n}}}\hat{\boldsymbol{\Xi}}\|_{\text{op}}$ are $O(1)$, and we have $\|\frac{1}{\hat{n}}\hat{\boldsymbol{H}}\hat{\boldsymbol{\Xi}}^\top\|_{\text{op}}$ is also O(1). These together bound the empirical covariance.

2. **Concentration on $\mathcal{V}$**: Since $\mathcal{V}$ corresponds to the representation space of $h(\boldsymbol{x})$, this condition is automatically satisfied by the $(\delta, 0, 0)$-decomposibility assumption on $h$.

3. **Kernel-wise $\delta$-isotropy on $\mathcal{V}^\perp$**: In this setting, since $\mathcal{V}^\perp$ corresponds to the column space of $\boldsymbol{M}^\perp$ (the high-dimensional sub-Gaussian part), we have

$$\|\frac{1}{\hat{n}}\hat{\boldsymbol{K}}(\boldsymbol{\Pi}_{\mathcal{V}^\perp}\alpha) - \frac{\sigma^2}{\hat{n}}\boldsymbol{I}\|_{\text{op}} = \|\frac{1}{\hat{n}}\hat{\boldsymbol{K}}(\xi) - \frac{\sigma^2}{\hat{n}}\boldsymbol{I}\|_{\text{op}}$$

By definition of the kernel matrix, $\hat{\boldsymbol{K}}(\xi) = [\xi(\hat{\boldsymbol{x}}_i)^\top \xi(\hat{\boldsymbol{x}}_j)]_{1\leq i,j\leq \hat{n}} = \hat{\boldsymbol{\Xi}}^\top \hat{\boldsymbol{\Xi}}$ with $\hat{\boldsymbol{\Xi}}$ defined above. Then the equation is essentially in the same form of Equation 34, so the previous proof applies here.

4. **Small cross-sample inner product on $\mathcal{V}^\perp$**: Similar to 3, we have

$$\|\frac{1}{\sqrt{\hat{n}\tilde{n}}}[(\boldsymbol{\Pi}_{\mathcal{V}^\perp}\alpha(\hat{\boldsymbol{x}}_i))^\top \boldsymbol{\Pi}_{\mathcal{V}^\perp}\alpha(\tilde{\boldsymbol{x}}_j)]_{1\leq i\leq \hat{n}, 1\leq j\leq \tilde{n}}\|_{\text{op}} = \|\frac{1}{\sqrt{\hat{n}\tilde{n}}}[\xi(\hat{\boldsymbol{x}}_i)^\top \xi(\tilde{\boldsymbol{x}}_j)]_{1\leq i\leq \hat{n}, 1\leq j\leq \tilde{n}}\|_{\text{op}} = \frac{1}{\sqrt{\hat{n}\tilde{n}}}\|\hat{\boldsymbol{\Xi}}^T \tilde{\boldsymbol{\Xi}}\|_{\text{op}},$$

where $\tilde{\boldsymbol{\Xi}}$ is defined in the same manner. Then the proof after Equation 35 for sub-Gaussian data applies.

5. **Diminishing population covariance on $\mathcal{V}^\perp$**: This refers covariance matrix of the sub-Gaussian part, and we simply have:

$$\|\boldsymbol{\Sigma}(\boldsymbol{\Pi}_{\mathcal{V}^\perp}h)\|_{\text{op}} = \|\boldsymbol{\Sigma}(\xi)\|_{\text{op}} = \|\mathbb{E}_{\mathcal{D}_{\boldsymbol{x}}}[\xi(\boldsymbol{x})\xi(\boldsymbol{x})^\top]\|_{\text{op}} = \frac{\sigma^2}{m} = o(\delta + \gamma) \quad \text{as } m = \omega(\hat{n}) = \omega(\tilde{n})$$

# D. Additional Experimental Details

## D.1. Training details

### D.1.1. MOLECULAR PREDICTION.

Our experiment is built on the GitHub codebase provided by (Fabian et al., 2020). The strong model, MolBERT, can be downloaded using the link provided on their GitHub repository. For the weak models, we train small transformers using their pipeline with a batch size of 256. For finetuning, we use SGD to train a linear model on representations with the following settings: batch size = 1024, learning rate = 0.001, weight decay = 0.1, and epochs = 2000 when using representations from the strong model; and batch size = 1024, learning rate = 0.01, weight decay = 0, and epochs = 2000 when using representations from the weak models.

### D.1.2. NLP TASKS WITH EMBEDDING MODELS.

We use `nvidia/NV-Embed-v2`, ranked first on the leaderboard of the Massive Text Embedding Benchmark (MTEB) (Muennighoff et al., 2022), as the strong model. We consider the following 22 embedding models as the weak model:

```
avsolatorio/GIST-Embedding-v0
Alibaba-NLP/gte-base-en-v1.5
jxm/cde-small-v1
thenlper/gte-base
infgrad/stella-base-en-v2
BAAI/bge-base-en-v1.5
thenlper/gte-small
intfloat/e5-base-v2
abhinand/MedEmbed-small-v0.1
nomic-ai/nomic-embed-text-v1
sentence-transformers/facebook-dpr-question_encoder-single-nq-base
sentence-transformers/paraphrase-MiniLM-L3-v2
sentence-transformers/average_word_embeddings_glove.840B.300d
sentence-transformers/roberta-base-nli-mean-tokens
sentence-transformers/all-mpnet-base-v1
sentence-transformers/bert-base-wikipedia-sections-mean-tokens
sentence-transformers/sentence-t5-base
Snowflake/snowflake-arctic-embed-s
TaylorAI/gte-tiny
jinaai/jina-embeddings-v2-small-en
sentence-transformers/gtr-t5-base
dumyy/sft-bge-small
```

During fine-tuning, we train a linear classifier on representations using the Adam optimizer (Kingma, 2014) with the following settings: batch size = 200, learning rate = 0.01, weight decay = 0.00001, and epochs = 200.

### D.1.3. NLP TASKS WITH END-TO-END FINETUEND LLMS.

We largely reuse the GitHub codebase provided by (Burns et al., 2023). We use `Qwen/Qwen-7B` as the strong model. We consider the following 28 LLMs as the weak model:

```
bigscience/bloom-560m
bigscience/bloomz-560m
bigscience/mt0-base
baidu/ernie-code-560m
bigscience/mt0-small
google/umt5-small
google/umt5-base
```

Table 2: Average Spearman correlation with hyperparameters selected on half of the models and evaluated on the rest.

| Justice | Commonsense |
|---|---|
| $0.885_{\pm 0.16}$ | $0.67_{\pm 0.20}$ |

```
google/mt5-base
facebook/xglm-564M
MBZUAI/LaMini-T5-61M
MBZUAI/LaMini-Flan-T5-77M
MBZUAI/LaMini-GPT-124M
MBZUAI/LaMini-Neo-125M
MBZUAI/LaMini-T5-223M
apple/OpenELM-270M
apple/OpenELM-450M
EleutherAI/pythia-160m
MBZUAI/LaMini-Flan-T5-248M
MBZUAI/LaMini-GPT-774M
cerebras/Cerebras-GPT-111M
google-t5/t5-small
facebook/opt-125m
Qwen/Qwen2.5-0.5B
distilbert/distilgpt2
EleutherAI/gpt-neo-125m
gpt2
google/mt5-small
EleutherAI/pythia-70m
```

We finetune all the models using the pipeline provided in the codebase, which employs the Adam optimizer with a batch size of 32 and trains for a single epoch. The learning rate is set to 5e-5 for weak models and 1e-5 for the strong model, following the default configuration in the codebase, which applies smaller learning rates for larger models.

### D.2. Details and discussions on hyperparameters

In Exp. I, we set $\alpha_w = \alpha_s = 0.1$ and $\beta_w = \beta_s = 0.1$ for all datasets. In Exp. II, we set $\alpha_w = 0.001$, $\alpha_s = 0.05$, $\lambda_w = 0.0001$, and $\lambda_s = 0.01$ for both datasets. In Exp. III, we tune the hyperparameters for each dataset, reporting the best result. Specifically, we set $\alpha_w = \alpha_s$ and vary them within the range $\{0.02, 0.05\}$, and vary $\beta_w$ and $\beta_s$ independently within the range $\{0.2, 0.5, 0.8, 1.0, 2, 4, 8\}$.

**Effect of hyperparameters.** We vary the hyperparameters to evaluate their impact on performance. In the setting of Exp. II, we vary $\alpha_w$ and $\alpha_s$ within the range $0.001, 0.01, 0.05$ and $\beta_w$ and $\beta_s$ within the range $0.0001, 0.001, 0.01$. The results are visualized in Figure 6. In the setting of Exp. III, we vary the hyperparameters while keeping $\alpha_w = \alpha_s$ as described in the previous paragraph, with results visualized in Figure 7. Although certain hyperparameter configurations may lead to lower correlation, a non-trivial positive correlation is observed in most cases. Interestingly, in Exp. III, which is seemingly the most 'challenging setting', the results are highly robust to changes in hyperparameters, with the worst-case correlation remaining around 0.6 across all three datasets.

**Cross-model hyperparameter transfer.** We note that, although each model could technically require different hyperparameters, in experiments we let all weak models share hyperparameters for simplicity and still achieve strong results, suggesting that our approach is not very sensitive to hyperparameters. Further, we present a new experiment demonstrating that hyperparameters selected using one group of models (i.e., as a validation set) generalize to other models. We randomly split the weak models into two groups, select hyperparameters based on one group, and evaluate them on the other. We repeat this 20 times and report the results in Table 2. Correlation remains high with low standard deviation, indicating that hyperparameters selected using a few models can reliably generalize to new ones. Additionally, we note that a small number of labeled data should suffice for hyperparameters tuning, as they are only used to measure test performance and not to compute our metric.

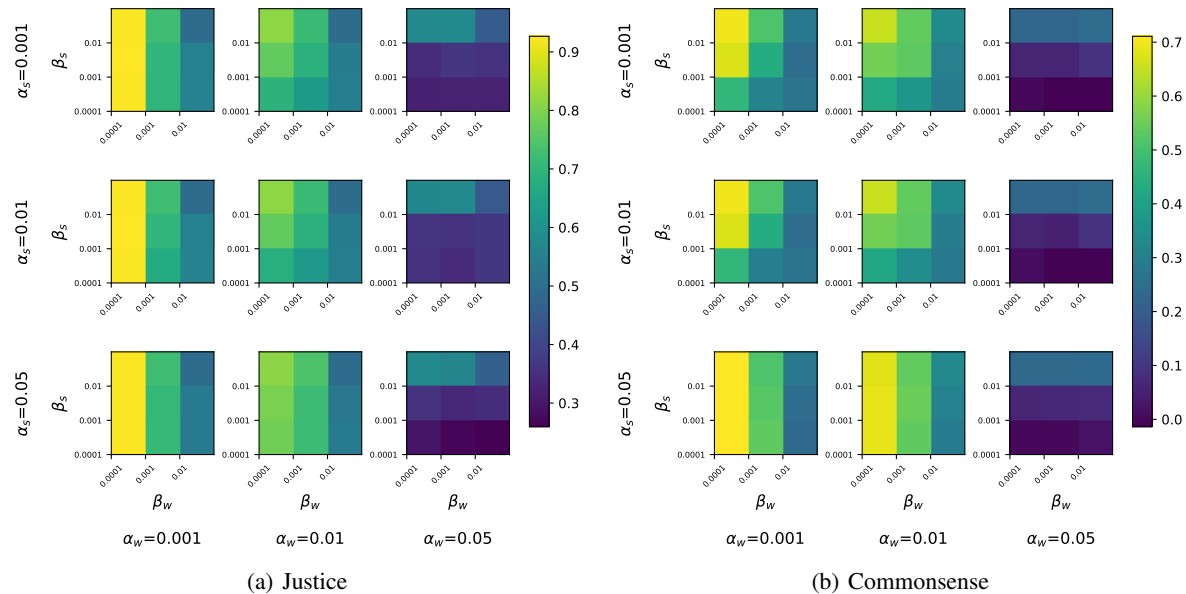

(a) Justice

(b) Commonsense

Figure 6: Effect of hyperparameters in Exp. II. Colors indicate Spearman correlation.

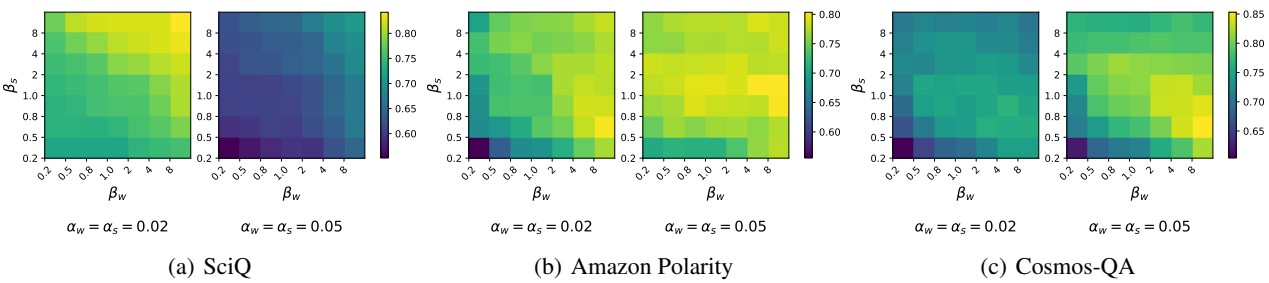

(a) SciQ

(b) Amazon Polarity

(c) Cosmos-QA

Figure 7: Effect of hyperparameters in Exp. III. Colors indicate Spearman correlation.

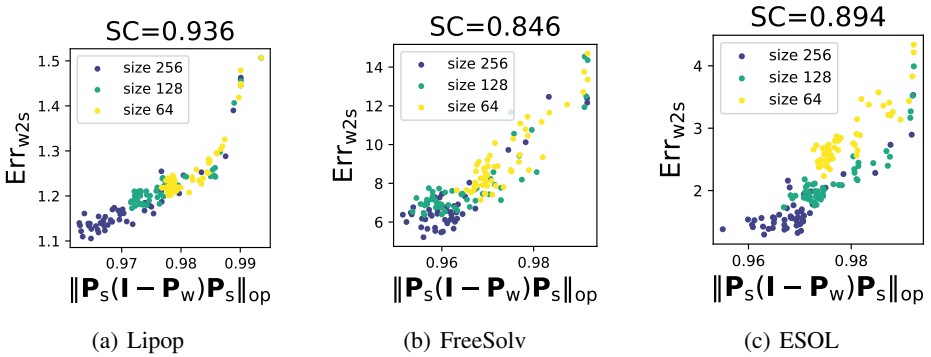

(a) Lipop

(b) FreeSolv

(c) ESOL

Figure 8: Results for $\|P_s(I - P_w)P_s\|_{\mathrm{op}}$ in Exp. I.

### D.3. Results for $\|P_s(I - P_w)P_s\|_{\mathrm{op}}$

Results for $\|P_s(I - P_w)P_s\|_{\mathrm{op}}$ are presented in Figures 8, 9, and 10. We observe a strong correlation between $\mathrm{Err}_{w2s}$ and $\|P_s(I - P_w)P_s\|_{\mathrm{op}}$ across the settings. These correlations are similar to those achieved using $\|P_s(I - P_w)\|_{\mathrm{op}}$, indicating

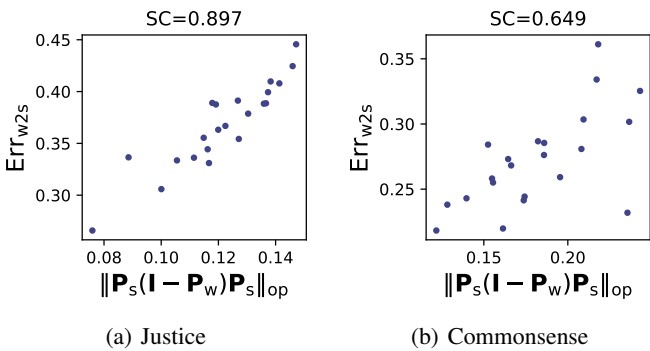

Figure 9: Results for $\|\boldsymbol{P}_{\mathrm{s}}(\boldsymbol{I} - \boldsymbol{P}_{\mathrm{w}})\boldsymbol{P}_{\mathrm{s}}\|_{\mathrm{op}}$ in Exp. II.

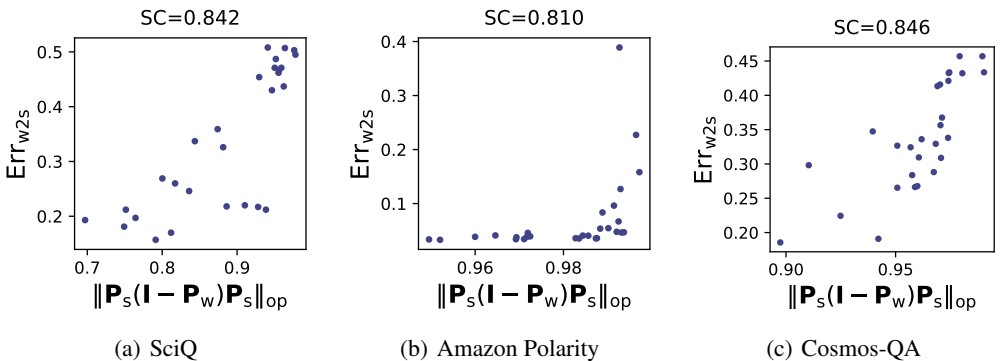

Figure 10: Results for $\|\boldsymbol{P}_{\mathrm{s}}(\boldsymbol{I} - \boldsymbol{P}_{\mathrm{w}})\boldsymbol{P}_{\mathrm{s}}\|_{\mathrm{op}}$ in Exp. III.

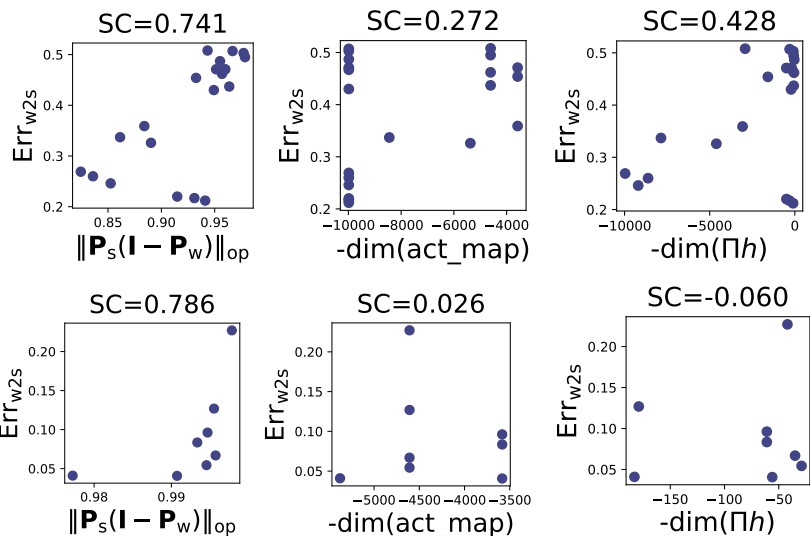

Figure 11: The top panel shows results on SciQ for models with sizes $\leq 10000$, while the bottom panel shows results on Amazon Polarity for models with sizes $\leq 8000$. The patterns observed here are consistent with those discussed in Figure 5 in the main paper.

that the two metrics are similarly informative for W2SG in practice, despite being theoretically derived in different ways.

### D.4. Comparison with model size and effective dimension

Figure 11 compares our metric with the activation map dimension and the dimension of approximated principal representations for smaller models on SciQ and Amazon Polarity. The results are consistent with those presented in Figure 5 in the main paper.

## E. Discussion

Using activation maps as representations in Exp. III is a simple heuristic that yields promising results. However, more principled methods for defining and extracting representations from LLMs, such as those through NTK (Malladi et al., 2023) or representation engineering (Zou et al., 2023), could be explored. Future research could leverage these approaches to improve results and uncover new applications. For instance, (Zou et al., 2023) introduces a method for extracting specific concept directions in representations, such as honesty and power-seeking. This could enable computing our metric based on topic-specific representations, allowing predictions of W2SG for general tasks within specific topical domains.

