# OpenReview forum: "Representations Shape Weak-to-Strong Generalization: Theoretical Insights and Empirical Predictions"
_ICML.cc/2025/Conference — ICML 2025 poster_

### Official Review · Reviewer_BNKN · 2025-03-13

**Overall Recommendation:** 4

**Summary:**

This paper studies weak to strong generalization, where a strong model is fine-tuned on a task using data labeled by a weaker supervisor model (it is known that perhaps surprisingly the strong model can outperform its weak supervisor). Specifically, the paper introduces estimators, depending only on internal representations of the strong and weak models on the input data distribution for the fine-tuning task, that correlate with the test set error Err w2s of the weakly supervised strong model.

**Claims And Evidence:**

Theoretical claims are precisely formulated with definitions and assumptions made clear, and proofs are included in the appendix (although I didn't read them). Experimental evidence supports the main claims.

One comment:
- Cor. 5.1-2: the *variance* of the labels does appear in these corollaries, so some discussion of the precise sense in which they are labeled agnostic would be helpful.

**Essential References Not Discussed:**

Not that I'm aware of.

**Experimental Designs Or Analyses:**

- hyperparameters $\alpha, \beta$: I wish there were additional discussion on this topic. How are these tuned? On what split of the data set? To maximize what (i.e. correlation with Err w2s)? I also wonder if there are some characteristics that would allow for choosing good $\alpha$ without hyperparameter tuning (it's effectively a cut off on eigenvalues, there are lots of heuristics for that). In the appendix, I see that for the embedding model experiments, low $\alpha_w, \beta_w$ and higher $\alpha_s, \beta_s$ do better, but for the LLM experiments e.g. higher $\beta_w$ does better. There's also notable dataset dependence in the LLM experiments.
- it would be interesting to see an experiment explicitly targeting the label agnostic property of this estimator of Err w2s. For example, does computing $\lvert P_s (I-P_w)\rvert$ on text from the common crawl correlate with Err w2s in the LLM experiments?

**Methods And Evaluation Criteria:**

Yes.

**Other Comments Or Suggestions:**

Overall, I felt that the balance of the main body of the paper leaned very heavily towards theory, leaving very little space for discussion of the experiments. Especially since some of the theoretical content is (useful, but not essential for the flow) examples, differing some of it to the appendix to allow for a more detailed experiment section (including, for example, more discussion about hyper parameter tuning as I asked about above) would improve the paper (in my view).

**Other Strengths And Weaknesses:**

- "As AI systems become increasingly capable of performing complex tasks beyond human comprehension" do you have an example of an an AI system performing a task that's "beyond human comprehension"? I get that it's a first sentence, and motivational introductions like this are very common today. I just want to know if we are verifiably in the "beyond human comprehension" regime (and impressive task completion in the sense of very few humans could complete the task or a human couldn't complete the task nearly as quickly, etc. don't count, because in those cases, at least some humans could at least comprehend what the system is doing).

**Questions For Authors:**

None beyond the above.

**Relation To Broader Scientific Literature:**

The key contributions of this paper are motivated by human-AI alignment research. They may have broader implications for distillation fine-tuning in machine learning.

As the introduction and related work outline, results of a similar flavor (decomposing Err w2s using one of various triangle inequalities and showing an easier-to-estimate term that emerges correlates with Err w2s). As I understand that the novelty of this paper lies in:
- label agnostic-ness
- estimates based on model representations

From looking at the references mentioned in related work, I feel that this paper has a relatively comprehensive experiment suite.

**Theoretical Claims:**

I did not read proofs, but I did read definitions, assumptions, statements, and claims carefully. I have a handful of mostly minor comments on the math.

- L110, top left: might also be worth mentioning using L2 regression as the objective function, as opposed to e.g. cross-entropy.
- L163, right: "w.r.t. a subspace V" in the italicized definition is strange because the subspace gets defined explicitly as part of the criterion. To me the following would make more sense: "representations of $h$ are $(\delta, \hat{gamma}, \tilde{gamma})$-decomposable for some $\delta$ ..." and then V gets defined as part of the existence criteria.
- L175 Left: I don't agree with the well concentrated characterization. This condition seems to be about the empirical quantities $\tilde{\Sigma}, \hat{\Sigma}$ closely approximating the expectations. There's also some Einstein notation on L180-181, unless that's introduced a summation symbol is warranted.
- Conditions d and e seem closely related: I.e. assuming (e), and that $hat{\mathcal{D}}, \tilde{\mathcal{D}}$ are both IID samples from the population $\mathcal{D}$, and some relationship between operator, norms of covariate and kernel, that the estimate of the full population statistic one gets from estimating with $hat{\mathcal{D}}, \tilde{\mathcal{D}}$ is also small? Okay, some work to be done here, but my point is if possible rearranging to: (e) first, and something should be said about (d) being "(e) but with hats and tildes" would be easier to grasp.
- Thm 3.6: this is interesting, but as a reader, it's not clear to me how important it is for the flow of the paper. Some commentary on what the theorem says in natural language would be helpful.

---

> ### Author Rebuttal · Authors · 2025-04-01
>
> We thank the reviewer for their positive feedback on our theoretical analysis, extensive experiments, novelty and broader impact.
>
> > Cor 5.1-2: why label agnostic
>
> As noted in L306 right, once we factor the label-dependent term out of the operator norm (becoming the label variance C), it can be treated as constant for a fixed dataset. When varying the models, the only factor that changes on RHS of 5.1 is $||P_s(I-P_w)||$, independent of labels. Thus the trend of RHS can be predicted w/o labels. Same for 5.2
>
> > L110,163
>
> Thanks for the suggestions. We’ll incorporate them in the revision.
>
> > L175 Left: L180 typos
>
> This is a typo. We didn’t intend to use Einstein notation. We’ll add the missing summation after $\frac{1}{n}$.
>
> > “Conditions d and e closely related”
>
> One possible way to make (d) and (e) more “unified” is to rephrase e as a variant of the cross-sample statement but with the size of one sample taken to infinity. We’ll note this in the revision.
>
> > Thm 3.6
>
> We’ll include the following discussion. Thm 3.6 highlights the generality of Def 3.3, showing more examples can be constructed. Any representation composed of a part satisfying Def 3.3 with δ=0 (eg, from a very low-rank distribution) and a high-dimensional sub-Gaussian will, as a whole, satisfy Def 3.3. Eg, Example 3.4 can be extended by concatenating sub-Gaussian.
>
> > discussion on hyperparameters
>
> (1) How we tuned the hps: See Sec D.2 for detailed hp values. We select the hps that maximizes correlation with test Err_w2s. Our metric is computed on the w2s split, consistent with theory.
>
> (2) **Cross-model hp transfer:** We note that, although each model could technically require different hps, in experiments we let all weak models share hps for simplicity and still achieve strong results, suggesting that our approach is not very sensitive to hps. Further, we present **a new experiment** demonstrating that hps selected using one group of models (ie, as a validation set) generalize to other models. We randomly split the weak models into two groups, select hps based on one group, and evaluate them on the other. We repeat this 20 times and report the results on 5 datasets https://anonymous.4open.science/r/icml2025figures/table.png. Correlation remains high with low std, indicating that hps selected using a few models can reliably generalize to new ones. Additionally, we note that a small number of labeled data should suffice for hp tuning, as they are only used to measure test performance and not to compute our metric.
>
> (3) **intuitions for hp selection**
>
> - **For β**: β captures the effect of regularization and should be set higher when stronger regularization is used. This could explain why the optimal β in Exp III is much higher than in II: in III finetuning for one epoch only (following prior work) introduces very early stopping and thus strong regularization.
>
> - **For α**:  The choice of α depends on the underlying dimensional structure. While the relationship can be complex, one intuition is that larger models tend to require a higher α. For small models whose dimensionality is relatively low compared to the sample size, most components can concentrate well. There may be few or no non-principal ones, so a small α suffices to filter them out. In contrast, larger models have high-dimensional yet low-rank representations (Huh et al 2021), where only a few top components concentrate with finite data. There are more non-principal components, and moreover, their magnitudes can appear inflated in the finite sample due to MP law—necessitating a larger α. These intuitions align with the reviewer’s observation from Fig 6 Exp II that the strong model requires a larger α than the weak ones
>
> > “explicitly targeting the label agnostic property; common crawl”
>
> We note that all experiments demonstrate the label-agnostic property. In Figs 2-4, our label-agnostic estimator (x-axis) strongly correlates with Err_w2s (y-axis). For the second half of the comment, if the reviewer was asking whether our metric computed on Common Crawl (CC) could predict performance on arbitrary tasks, this is not feasible: label-agnostic≠task-agnostic. Predicting performance on task A still requires unlabeled data from A. It is not reasonable to expect that CC could be used to indicate performance on any tasks. If the question was instead about predicting performance on CC itself, the bottleneck is evaluation: since CC lacks labels, we can’t compute W2SG performance as ground truth to compare our metric against.
>
> > beyond human comprehension regime
>
> We will revise the sentence to reflect: (1) in the future AI may surpass humans in certain tasks; (2) even today, AI can outperform *average* humans on certain tasks—yet W2SG shows that *average* humans can still help improve AI in those tasks. In particular, our results indicate which humans (or weaker LLMs) can best teach a strong AI —and interestingly the answer is not necessarily the strongest human or the strongest weak LLM.

---

### Official Review · Reviewer_Yk6a · 2025-03-13

**Overall Recommendation:** 3

**Summary:**

This work provides a theoretical analysis of how a strong model can surpass its weak supervisor by studying the structure of their representations. They key insight (beyond prior analyses) is that even when a strong model perfectly fits the weak model’s predictions at train time, its surpasses its weak supervisor due to its better “principal representations,” which govern generalization. They quantify this, and use it to estimate the error of weak-to-strong models in a few empirical settings.

**Claims And Evidence:**

In their experiments, the authors argue that their representation-based metric captures weak-to-strong error beyond model size, i.e., it predicts weak-to-strong error in a more fine-grained manner than model size. Besides model size, it would be valuable to also consider grouping according to the error of the weak supervisor. For a given error-level of the weak supervisor, does the representation-based metric predict weak-to-strong error?

This would be a more convincing experiment that illustrates that the relative representation structures of the weak teacher and strong student matter rather than just the quality of the student model. Without controlling for weak supervisor quality, it’s hard to know whether this is a confounder that causes a high correlation between weak-to-strong error and their representation-based metric.

**Essential References Not Discussed:**

N/A

**Experimental Designs Or Analyses:**

See "Claims and Evidence"

**Methods And Evaluation Criteria:**

See "Claims and Evidence"

**Other Comments Or Suggestions:**

N/A

**Other Strengths And Weaknesses:**

The writing throughout the first part of Section 3 could be more clear. Lots of notations and intermediate results are presented without a lot of guiding intuitions towards the final claims.

**Questions For Authors:**

N/A

**Relation To Broader Scientific Literature:**

The paper studies a topic of recent interest: the ability of a strong model to exceed the performance of its weak supervisor when trained on labels produced by this weak supervisor. There are several existing theoretical analyses studying the same phenomenon. This paper specifically studies representation structures, and yields novel insights on benign overfitting.

**Theoretical Claims:**

I read through the theoretical claims presented in the paper and they are clear and convincing. I did not check the proofs provided in the supplementary material.

---

> ### Author Rebuttal · Authors · 2025-04-01
>
> We thank the reviewer for finding our theoretical claims clear and convincing, and for acknowledging the novel insights of our paper.
>
> > “...relative representation structures of the weak teacher and strong student matter…Without controlling for weak supervisor quality, it’s hard to know whether this is a confounder …”
>
> (a) **A new experiment**. We note there are cases where the weak teacher’s performance cannot indicate W2S performance. We conducted a new experiment, where we control the weak supervisor’s error to lie within a narrow range, and observe that **the weak supervisor’s error correlates poorly with the W2S error, while our metric shows a strong correlation**.
>
> Since in our main experiments in the paper, the weak supervisors’ errors span a relatively large range, it is difficult to find a sufficient number of weak supervisors with similar errors. Therefore, we explicitly construct weak supervisors with similar error levels in the following way: We take a single checkpoint of a weak supervisor from Experiment I (the one with hidden size 256 pretrained for 5 epochs), and generate 20 modified versions of it by randomly masking out 100 coordinates of its features each time. We then run the weak-to-strong pipeline for each of these 20 weak supervisors. We compare the correlation between $Err_w$ and $Err_{w2s}$, and between $| P_s(I - P_w) |{\mathrm{op}}$ and $Err_{w2s}$ across all three datasets. As shown in the table below, the weak supervisor’s error correlates poorly with the weak-to-strong error, while our metric maintains a strong correlation. This further demonstrates that the detailed relationship between the weak teacher and the strong student plays an important role in weak-to-strong generalization—beyond what can be explained by the weak supervisor’s error alone.
>
> |  | Lipop | FreeSolv | ESOL |
> |----|----|----|----|
> | Err_w  | 0.24  | 0.29 | 0.13   | |
> | $\|\| P_s(I-P_w) \|\|_{op}$  |  **0.62**    | **0.65** | **0.61** | |
>
> (b) We illustrate our intuition with a simple example. Suppose a downstream task consists of 40% advanced linear algebra and 60% advanced calculus. We have two weak pretrained models: model A specializes in basic linear algebra, and model B in basic calculus. Assume fine-tuning mainly builds on existing knowledge rather than learning from scratch. Then, after fine-tuning, model A would likely achieve ~40% performance and model B ~60%, reflecting their alignment with the task. Now consider a strong student pretrained only on linear algebra. According to our main theory, model A—being aligned with linear algebra—should be a better supervisor for this student leading to better W2SG performance, even if its own performance is lower. Our proposed metric captures this alignment and should correlate with W2SG performance, whereas weak supervisor performance alone does not. This highlights a case where our metric offers meaningful insight beyond what weak model performance alone can explain.
>
> (c) **Practical perspective**: Measuring the weak supervisor’s error requires access to labels. In contrast, our metric is label-agnostic. The fact that it achieves such a high correlation while using less information is already impressive.
>
> > “The writing throughout the first part of Section 3 could be more clear. Lots of notations and intermediate results are presented without a lot of guiding intuitions towards the final claims.”
>
> Thanks for the suggestion. Due to space constraints and the large number of results, we focused on conveying the most important messages and were unable to include more detailed explanations. We will add more intuitive explanations in the revised version.
>
> If there are no further concerns, and given the reviewer’s largely positive feedback, we would greatly appreciate it if the reviewer would consider raising the score.

---

### Official Review · Reviewer_wiCj · 2025-03-16

**Overall Recommendation:** 3

**Summary:**

This paper provides a theoretical analysis for weak-to-strong generalization (W2SG) from a representation-based perspective. In particular, the authors consider finetuning over fixed representations with mild structural assumptions.
- It is shown that the overlap between the principal subspace of the strong (student) model's representation and the orthogonal complement of the weak (teacher) model's representation is a key quantity that governs W2SG.
- The theoretical framework is then leveraged to explain benign overfitting in W2SG -- errors that do not align with the strong model’s principal subspace do not affect W2SG.
- The overlap between the two subspaces -- the principal subspace of the strong model's representation and the orthogonal complement of the weak model's representation -- provides a metric that theoretically predicts the W2SG performance. In practice, this metric demonstrates a strong correlation with the W2SG performance across various datasets and architectures.

**Claims And Evidence:**

The main claims made in the paper are well-supported by analysis and experiments. The theoretical analysis is mostly reasonable. However, some statements seem to have minor issues and are in lack of sufficient explanations (see "Theoretical Claims"). The empirical evidences are convincing.

**Essential References Not Discussed:**

To my knowledge, the paper discussed the essential references in the field.

**Experimental Designs Or Analyses:**

I review the experiments in the main text (Sec 5) and some of the details in Appendix E. The experimental setup is reasonable, sufficiently detailed, and well-organized. The empirical results align with the theoretical claims and provide convincing evidence for the proposed metric.

**Methods And Evaluation Criteria:**

Yes, the proposed metric is well motivated by the analysis.

**Other Comments Or Suggestions:**

Comments are raised in previous sections.

**Other Strengths And Weaknesses:**

Strengths and weaknesses are discussed in previous sections.

**Questions For Authors:**

Major questions are raised in previous sections.

**Relation To Broader Scientific Literature:**

This paper provides a theoretical analysis for W2SG from a representation-based perspective. Toward the three contributions of this work:
- The analysis-inspired metric is novel, intuitive, and well-motivated.
- The explanation on benign overfitting in W2SG intuitive and insightful. But I feel the difference from the analysis in (Wu & Sahai, 2024) is not well explained, especially after stating the results on benign overfitting in W2SG. It seems that both benign overfitting analysis share the same intuition, just using different ensemble models. If more sophisticated mechanisms are involved, it would be helpful to remark on the difference between the two analyses.
- The empirical verification of the correlation between the proposed metric and W2SG performance is extensive and convincing.

**Theoretical Claims:**

I couldn't verify all the proofs in the appendix, but by quickly going through Appendix A and B, I think the main theoretical results in the paper seem reasonable. However, I feel that some statements are not accurately made or well organized.
- While the assumptions in Definition 3.3 are relatively mild from the analysis perspective, they are nevertheless dense and not quite well-motivated. For example, only the notion of "kernel-wise isotropy" in Def 3.2 is explained, with the explanation focusing on its necessity for the analysis, but not its intuition or practical implications.
- Some assumptions in Def 3.3 looks counterintuitive. For example, in "(b) Concentration on $\mathcal{V}$", the concentration of correlation with labels, $\|\|\frac{1}{n} \Pi_V h(x_i) y_i - \mathbb{E}[\Pi_V h(x) y]\|\|$, should accurately be $\|\|\frac{1}{n} \sum_{i=1}^n \Pi_V h(x_i) y_i - \mathbb{E}[\Pi_V h(x) y]\|\|$?

---

> ### Author Rebuttal · Authors · 2025-04-01
>
> We thank the reviewer for finding our claims well-supported, our explanations insightful and intuitive, the theory and experiments extensive and convincing, and the proposed metric novel. We respond to the comments below.
>
> > the distinction between Sec 4 and Wu & Sahai (2024)
>
> The main differences are twofold:
>
> (1) Wu & Sahai (2024) show that benign overfitting **can happen**, but they do not extract **general insights about when and how it occurs**. In contrast, we identify a single key quantity driving benign overfitting in W2SG—namely, $||P_s(I - P_w)\frac{1}{\sqrt{n}} y||$ in Thm 4.1—which characterizes how much of the label aligns with the intersection between what is missed by the weak model's principal kernel and captured by the strong model’s principal kernel. When this quantity is small, the strong model can avoid repeating the weak model’s mistake, regardless of the extent of overfitting, thereby achieving error mitigation. This very mechanism is not revealed in Wu & Sahai.
>
> (2) Our Thm 4.1 is stated in a very general setting, whereas Wu & Sahai focus on a highly specific distribution with detailed assumptions, making it more of a toy example than a realistic scenario. E.g., there is no evidence that neural network representations follow exactly their assumed bi-level ensemble structure  and that labels depend 1-sparsely on representations. In contrast, our assumptions (discussed on page 4) cover a wide range of realistic cases, supported by literature suggesting that neural network representations often exhibit such properties.
>
> > further explanation of assumptions in Def 3.3; and intuition or practical implications of condition (c)
>
> Due to space limitations, we only provided explanations for kernel-wise isotropy and small cross-sample inner products on $\mathcal{V}^\perp$, as we believe these two are the most involved, while the others are relatively natural and self-explanatory. Here, we provide further explanation for all the items, which we’ll include in the revised version.
>
> - (a) is a basic condition that ensures reasonable magnitudes of representations and labels.
> - (b) states that representations are well-concentrated in the subspace $\mathcal{V}$, both in terms of their covariance and their correlation with labels. This is why the representations on $\mathcal{V}$ are referred to as the principal representations—they are the part where the empirical distribution closely aligns with the underlying population distribution.
>
> - (c) implies that kernels constructed using only the components in $\mathcal{V}^\perp$ exhibit a certain level of uniformity across all directions, with the degree of this uniformity controlled by $\delta$. In the paper, we discuss two extreme cases—one with very small $\delta$ and one with very large $\delta$—to aid understanding. Importantly, this assumption is not made solely for analytical convenience; it is also general and applicable to realistic settings. For example, high-dimensional sub-Gaussian noise satisfies this condition with a small $\delta$—a scenario highly relevant to deep neural networks with large internal dimensions—since these vectors tend to be orthogonal to each other in the high-dimensional limit. More concrete instances can be found in Examples 3.4 and 3.5 and Thm 3.6, with their significance and relevance discussed in the right column of page 4. Thus, (c) is a key condition that allows us to capture all these diverse scenarios. It is not just analytically useful, but also practically relevant to real-world scenarios.
>
> - (d) holds either when representations on $\mathcal{V}^\perp$ are nearly orthogonal across samples or when their magnitudes are small.
>
> - (e) means that the representations on $\mathcal{V}^\perp$ have small magnitudes in the population.
>
> > Typos in Def 3.3 (b)
>
> As the reviewer correctly pointed out, the expression $|| \frac{1}{\tilde{n}} \Pi_{\mathcal{V}} h(\tilde{x_i}) \tilde{y_i} - \mathbb{E}[ \Pi_{\mathcal{V}} h(x) y ]||$ contains a typo—we forgot to include the summation after $\frac{1}{\tilde{n}} $. We will fix it in the revised version.
>
> If there are no other concerns, and given that the reviewer’s feedback is largely positive, we would greatly appreciate it if the reviewer could consider raising the score.

---

> > ### Comment · Reviewer_wiCj · 2025-04-02
> >
> > I appreciate the authors’ responses to my questions. I think this paper provides some valuable insights, but the presentation of some theoretical results could be improved. Overall, I will maintain my current evaluation.

---

### Decision · Program_Chairs · 2025-05-01

**Decision:**

Accept (poster)

**Comment:**

This paper presents an extensive theoretical analysis of weak-to-strong generalization from a representational perspective, supported by some empirical validation. While the reviewers and I generally found the work useful, the presentation is often unclear, including both issues in the communication of the theoretical results and their assumptions, and the details of the empirical experiments. Improving the clarity of the presentation would help to communicate the value of the perspective offered.